# Understanding and Improving Adversarial Collaborative Filtering for Robust Recommendation

**Kaike Zhang**[1,2], **Qi Cao**[1]*, **Yunfan Wu**[1,2], **Fei Sun**[1], **Huawei Shen**[1], **Xueqi Cheng**[1]

[1] CAS Key Laboratory of AI Safety, Institute of Computing Technology,
Chinese Academy of Sciences
[2] University of Chinese Academy of Sciences
Beijing, China
{zhangkaike21s, caoqi, wuyunfan19b, sunfei, shenhuawei, cxq}@ict.ac.cn

## Abstract

Adversarial Collaborative Filtering (ACF), which typically applies adversarial perturbations at user and item embeddings through adversarial training, is widely recognized as an effective strategy for enhancing the robustness of Collaborative Filtering (CF) recommender systems against poisoning attacks. Besides, numerous studies have empirically shown that ACF can also improve recommendation performance compared to traditional CF. Despite these empirical successes, the theoretical understanding of ACF's effectiveness in terms of both performance and robustness remains unclear. To bridge this gap, in this paper, we first theoretically show that ACF can achieve a lower recommendation error compared to traditional CF with the same training epochs in both clean and poisoned data contexts. Furthermore, by establishing bounds for reductions in recommendation error during ACF's optimization process, we find that applying personalized magnitudes of perturbation for different users based on their embedding scales can further improve ACF's effectiveness. Building on these theoretical understandings, we propose **P**erson**a**lized **M**agnitude **A**dversarial **C**ollaborative **F**iltering (**PamaCF**). Extensive experiments demonstrate that PamaCF effectively defends against various types of poisoning attacks while significantly enhancing recommendation performance.

## 1 Introduction

Collaborative Filtering (CF) is widely recognized as a powerful tool for providing personalized recommendations [1, 2, 3] across various domains [4, 5]. However, the inherent openness of recommender systems allows attackers to inject fake users into the training data, aiming to manipulate recommendations, also known as poisoning attacks [6, 7]. Such manipulations can skew the distribution of item exposure, degrading the overall quality of the recommender system, thus harming the user experience and hindering the long-term development of the recommender system [8].

Existing methods for defending against poisoning attacks in CF can be categorized into two types [8]: (1) detecting and mitigating the influence of fake users [9, 10, 11, 12, 13, 14], and (2) developing robust models via adversarial training, also known as Adversarial Collaborative Filtering (ACF) [15, 16, 17, 18, 19, 20]. The first strategy focuses on detecting and removing fake users from the dataset before training [9, 10, 11, 14] or mitigating their impact during the training phase [12, 13]. These methods often rely on predefined assumptions about attacks [9, 12] or require labeled data related to attacks [10, 11, 12, 13]. Consequently, deviations from predefined attack patterns may lead to misclassification, failing to resist attacks while potentially harming genuine users' experience [13].

---

*Corresponding author.

38th Conference on Neural Information Processing Systems (NeurIPS 2024).

In contrast, ACF provides a more general defense paradigm without prior knowledge [15, 16, 17, 18, 19, 20]. Poisoning attacks in recommender systems mainly affect the learned embeddings of users and items, i.e., the system's parameters [6, 7]. Predominant ACF methods, particularly those aligned with Adversarial Personalized Ranking (APR) framework [15], heuristically incorporate adversarial perturbations at the parameter level during the training phase to mitigate these attacks [15, 17, 19, 20]. This approach employs a "min-max" paradigm, designed to minimize the recommendation error while contending with parameter perturbations aimed at maximizing this error within a specified magnitude [15], thus enhancing the robustness of CF.

It is interesting to note that adversarial training in the Computer Vision (CV) domain [21, 22, 23] has been observed to degrade model performance on clean samples [24, 25]. Several studies have also theoretically demonstrated a trade-off between robustness against evasion attacks and the performance of adversarial training in CV [26]. In contrast, ACF in recommender systems has been shown in numerous studies not only to enhance the robustness against poisoning attacks [8, 13, 18] but also to improve recommendation performance [15, 20, 27]. Despite the empirical evidence highlighting ACF's advantages, it still lacks a comprehensive theoretical understanding, which limits the ability to fully exploit the benefits and potential of ACF. To bridge this gap, in this paper, we propose the following research questions for further investigation:

    i. *Why does ACF enhance both robustness and performance compared to traditional CF?*

   ii. *How can we further improve ACF?*

To answer these questions, we delve into a theoretical analysis of a simplified CF scenario. This analysis confirms that ACF can achieve a lower recommendation error at the same training epoch in both clean and poisoned data contexts, showing better performance and robustness compared to traditional CF. To investigate potential improvements to ACF, we establish upper and lower bounds for reductions in recommendation error during ACF's optimization process. Our findings indicate that (1) Users have varying constraints for perturbation magnitudes, i.e., different maximum perturbation magnitudes; (2) Within these constraints, applying personalized perturbation magnitudes as much as possible for each user can increase the error reduction bounds, further improving ACF's effectiveness.

Extending our theoretical results to practical CF scenarios, we establish a positive correlation between users' maximum perturbation magnitudes and their embedding scales. Building on these theoretical understandings, we introduce **P**ersonal**a**lized **M**agnitude **A**dversarial **C**ollaborative **F**iltering (**PamaCF**). PamaCF dynamically and personally assigns perturbation magnitudes based on users' embedding scales. Extensive experiments confirm that PamaCF outperforms baselines in both performance and robustness. Notably, PamaCF increases the average recommendation performance of the backbone model by 13.84% and reduces the average success ratio of attacks by 44.92% compared to the best baseline defense method. The main contributions of our study are summarized as follows:

- We provide theoretical evidence that ACF can achieve better performance and robustness compared to traditional CF in both clean and poisoned data contexts.

- We further identify upper and lower bounds of reduction in recommendation error for ACF during its optimization and demonstrate that applying personalized magnitudes of perturbation for each user can further improve ACF.

- Based on the above theoretical understandings, we propose Personalized Magnitude Adversarial Collaborative Filtering (PamaCF), with extensive experiments confirming that PamaCF further improves both performance and robustness compared to state-of-the-art defense methods.

The code of our experiments is available at `https://github.com/Kaike-Zhang/PamaCF`.

## 2 Preliminary

**Collaborative Filtering** (CF) methods are widely employed in recommender systems. Following [1], we define a set of users $\mathcal{U} = \{u\}$ and a set of items $\mathcal{V} = \{v\}$. Using data from user-item interactions, our objective is to learn latent embeddings $\boldsymbol{U} = [\boldsymbol{u} \in \mathbb{R}^d]_{u \in \mathcal{U}}$ for users and $\boldsymbol{V} = [\boldsymbol{v} \in \mathbb{R}^d]_{v \in \mathcal{V}}$ for items. Then, we employ a preference function $f : \mathbb{R}^d \times \mathbb{R}^d \to \mathbb{R}$, which predicts user-item preference scores, denoted as $\hat{r}_{u,v} = f(\boldsymbol{u}, \boldsymbol{v})$.

**Adversarial Collaborative Filtering** (ACF) is acknowledged as an effective approach for enhancing both the performance and the robustness of CF recommender systems in the face of poisoning attacks. ACF methods, particularly within the framework of Adversarial Personalized Ranking (APR) [15], integrate adversarial perturbations at the parameter level (i.e., the latent embeddings $U$ and $V$) during the training phase. Let $\mathcal{L}(\Theta)$ denote the loss function of the CF recommender system, where $\Theta = (U, V)$ represents the recommender system's parameters. ACF methods apply perturbations $\Delta$ directly to the parameters as:

$$\mathcal{L}_{\mathrm{ACF}}(\Theta) = \mathcal{L}(\Theta) + \lambda \mathcal{L}(\Theta + \Delta^{\mathrm{adv}}),$$
$$\text{where} \quad \Delta^{\mathrm{adv}} = \arg \max_{\Delta, \|\Delta\| \leq \epsilon} \mathcal{L}(\Theta + \Delta), \tag{1}$$

where $\epsilon > 0$ defines the maximum magnitude of perturbations, and $\lambda$ is the adversarial training weight. Due to constraints on space, a detailed discussion of related works is provided in Appendix A.

## 3 Theoretical Understanding of ACF

In this section, we provide a theoretical analysis of why ACF achieves superior performance and robustness compared to traditional CF from the perspective of recommendation error. Then, we explore mechanisms to further improve ACF's effectiveness based on its error reduction bounds. For clarity and simplicity, we initially focus on a Gaussian Single-item Recommender System, aligning with the frameworks presented in [18, 28]. It's important to **note that** the insights and analytical frameworks developed here are also applicable to more practical scenarios, as discussed in Section 4.

**Definition 1** (Gaussian Recommender System). *Given a rating set $\mathcal{R} = \{r_1, r_2, \ldots, r_n\}$ corresponding to $n$ users, where each rating $r$ is randomly selected from $\{\pm 1\}$, an average embedding vector $\bar{u} \in \mathbb{R}^d$, and $\sigma > 0$, the Gaussian Recommender System initializes each user's embedding $u$ from the normal distribution $\mathcal{N}(r\bar{u}, \sigma^2 I)$. The item embedding $v$ is initialized as the average vector derived from these users: $v = \frac{1}{n} \sum_{i=1}^{n} r_i u_i$. Then, a preference function $f : \mathbb{R}^d \times \mathbb{R}^d \to \{\pm 1\}$ is employed to predict user preferences: $f(u, v) = \mathrm{sgn}(\langle v, u \rangle)$, where $\mathrm{sgn}(\cdot)$ denotes the sign function, returning 1 if $\langle u, v \rangle > 0$ and -1 otherwise.*

Based on Definition 1, we obtain $\mathcal{I} = \{(u_1, r_1), \ldots, (u_n, r_n)\}$, where $u$ represents the system-learned user embedding. With continued training, both each user embedding $u$ and item embedding $v$ are iteratively updated. Let $u_{(t)}$ and $v_{(t)}$ denote user and item embeddings at the $t^{\mathrm{th}}$ epoch, respectively. For analytical simplicity and without loss of generality, we define the standard loss function $\mathcal{L}(\Theta)$ (as traditional CF) used in the Gaussian Recommender System as follows [18]:

$$\mathcal{L}(\Theta_{(t)}) = - \sum_{(u,r) \in \mathcal{I}} \left[ r \cdot \langle u_{(t)}, v_{(t)} \rangle \right], \tag{2}$$

where the model parameters $\Theta_{(t)} = \left( v_{(t)}, \left[ u_{1,(t)}, u_{2,(t)}, \ldots, u_{n,(t)} \right] \right)$. To integrate ACF into the Gaussian Recommender System, we introduce the adversarial loss [15], $\mathcal{L}_{\mathrm{adv}}(\Theta)$, defined as:

$$\mathcal{L}_{\mathrm{adv}}(\Theta_{(t)}) = \mathcal{L}(\Theta_{(t)}) - \lambda \sum_{(u,r) \in \mathcal{I}} \left[ r \cdot \langle u_{(t)} + \Delta_u, v_{(t)} + \Delta_v \rangle \right], \tag{3}$$

where $\lambda$ is the adversarial training weight. The perturbations $\Delta_u$ and $\Delta_v$ are applied to the user and item embeddings, respectively, as computed based on Equation 1.

### 3.1 Why Does Adversarial Collaborative Filtering Benefit Recommender Systems?

To analyze the performance and robustness of traditional CF and ACF within the Gaussian Recommender System, we evaluate them from the perspective of recommendation error during the training process. For each user, both performance and robustness are reflected by the user's recommendation error. Specifically, attacks—whether item promotion attacks [6, 29] or performance damage attacks [8]—inevitably increase the user's recommendation error. Meanwhile, a smaller recommendation error means a higher recommendation performance. For a given user $u$, the initial item embedding $v_{(0)}$ in the Gaussian Recommender System can be approximately modeled[2] as a sample from $\mathcal{N}(\bar{u}, \frac{\sigma^2}{n-1} I)$. Here, we provide the definition of recommendation error for user $u$.

---

[2]The precise form is $\mathcal{N}(\bar{u}, \frac{(n-1)\sigma^2}{n^2} I)$, but we make this approximation for the sake of clarity and brevity. The approximation does not impact the subsequent theoretical results.

**Definition 2** (Recommendation Error). *Given a Gaussian Recommender System $f_{(t)}$ that has been trained for $t$ epochs, the recommendation error for the user $\boldsymbol{u}$ with rating $r$ at the $t^{\text{th}}$ epoch is defined as the probability that the system's prediction does not align with the user's actual rating, as:*

$$\mathbb{P}_{\boldsymbol{v}_{(0)} \sim \mathcal{N}(\bar{\boldsymbol{u}}, \frac{\sigma^2}{n-1}I)} \left[ f_{(t)}(\boldsymbol{u}, \boldsymbol{v}) \neq r \mid (\boldsymbol{u}, r) \right] := \mathbb{E}_{\boldsymbol{v}_{(0)} \sim \mathcal{N}(\bar{\boldsymbol{u}}, \frac{\sigma^2}{n-1}I)} \left[ \mathbb{I} \left( f_{(t)}(\boldsymbol{u}, \boldsymbol{v}) \neq r \mid (\boldsymbol{u}, r), \boldsymbol{v}_{(0)} \right) \right],$$

*where $\mathbb{I}(\cdot)$ is an indicator function that returns 1 if the condition is true and 0 otherwise.*

Based on the framework of ACF [15, 20], which includes $t$ epochs of pre-training with standard loss before adversarial training, we derive a theorem that identifies the difference in recommendation error between standard and adversarial loss at the $(t+1)^{\text{th}}$ epoch. To distinguish between the recommendation error of traditional CF and ACF, we define $\mathbb{P}_{\boldsymbol{v}_{(0)} \sim \mathcal{N}(\bar{\boldsymbol{u}}, \frac{\sigma^2}{n-1}I)} \left[ f_{(t+1)}(\boldsymbol{u}, \boldsymbol{v}) \neq r \mid (\boldsymbol{u}, r) \right]$ as the recommendation error following standard training (Equation 2) at the $(t+1)^{\text{th}}$ epoch, and $\mathbb{P}^{\text{adv}}_{\boldsymbol{v}_{(0)} \sim \mathcal{N}(\bar{\boldsymbol{u}}, \frac{\sigma^2}{n-1}I)} \left[ f_{(t+1)}(\boldsymbol{u}, \boldsymbol{v}) \neq r \mid (\boldsymbol{u}, r) \right]$ as the recommendation error following adversarial training (Equation 3) at the $(t+1)^{\text{th}}$ epoch.

**Theorem 1.** *Consider a Gaussian Recommender System $f_{(t)}$, pre-trained for $t$ epochs using the standard loss function (Equation 2). Given a learning rate $\eta$, an adversarial training weight $\lambda$, and a perturbation magnitude $\epsilon$, when $\epsilon < \frac{\min(\|\boldsymbol{u}_{(t)}\|, \|\bar{\boldsymbol{u}}\|)}{\eta\lambda}$, and $\|\bar{\boldsymbol{u}}\| \gg \sigma^3$, the recommendation error for a user $\boldsymbol{u}$ with rating $r$ at the $(t+1)^{\text{th}}$ epoch follows that:*

$$\mathbb{P}_{\boldsymbol{v}_{(0)} \sim \mathcal{N}(\bar{\boldsymbol{u}}, \frac{\sigma^2}{n-1}I)} \left[ f_{(t+1)}(\boldsymbol{u}, \boldsymbol{v}) \neq r \mid (\boldsymbol{u}, r) \right] > \mathbb{P}^{\text{adv}}_{\boldsymbol{v}_{(0)} \sim \mathcal{N}(\bar{\boldsymbol{u}}, \frac{\sigma^2}{n-1}I)} \left[ f_{(t+1)}(\boldsymbol{u}, \boldsymbol{v}) \neq r \mid (\boldsymbol{u}, r) \right].$$

For the proof, please refer to Appendix D.1.1. After the same epochs of pre-training, ACF at the next epoch achieves a lower recommendation error compared to traditional CF, thereby benefiting recommendation performance.

Next, our analysis extends to contexts where the recommender system is subject to poisoning attacks. These attacks involve injecting fake users into the system's training dataset to manipulate item exposure. We examine a Gaussian Recommender System with $\mathcal{I} = \{(\boldsymbol{u}_1, r_1), \ldots, (\boldsymbol{u}_n, r_n)\}$, where each tuple $(\boldsymbol{u}, r) \in \mathbb{R}^d \times \{\pm 1\}$ represents the learned embedding and the rating of a genuine user. A *poisoning attack* on this system injects a *poisoning user set*, $\mathcal{I}' = \{(\boldsymbol{u}'_1, r'_1), (\boldsymbol{u}'_2, r'_2), \ldots, (\boldsymbol{u}'_{n'}, r'_{n'})\}$, with each tuple $(\boldsymbol{u}', r') \in \mathbb{R}^d \times \{\pm 1\}$ representing a fake user crafted by attackers[4]. The *poisoned item embedding* $\boldsymbol{v}'$ is reinitialized to include both genuine and malicious contributions:

$$\boldsymbol{v}' = \frac{1}{n+n'} \left( \sum_{(\boldsymbol{u}, r) \in \mathcal{I}} r\boldsymbol{u} + \sum_{(\boldsymbol{u}', r') \in \mathcal{I}'} r'\boldsymbol{u}' \right),$$

where $n$ and $n'$ represent the number of genuine and fake users, respectively.

To evaluate the impact of these attacks, we introduce a formal definition of recommendation error in poisoned data.

**Definition 3** ($\alpha$-Poisoned Recommendation Error). *Given a boundary $\alpha > 0$, and a set of fake users injected by attackers within this boundary, i.e., $\mathcal{I}' \subseteq \mathcal{P}(\boldsymbol{u}', \alpha) = \{(\boldsymbol{u}', r') \mid (\boldsymbol{u}', r') \in \mathbb{R}^d \times \{\pm 1\} \wedge \|\boldsymbol{u}'\|_\infty \leq \alpha\}$, the $\alpha$-poisoned recommendation error for the genuine user $\boldsymbol{u}$ with rating $r$ at the $t^{\text{th}}$ epoch is defined as the probability:*

$$\mathbb{P}_{\boldsymbol{v}_{(0)} \sim \mathcal{N}(\bar{\boldsymbol{u}}, \frac{\sigma^2}{n-1}I)} \left[ f_{(t),\alpha}(\boldsymbol{u}, \boldsymbol{v}') \neq r \mid (\boldsymbol{u}, r) \right] := \mathbb{E}_{\boldsymbol{v}_{(0)} \sim \mathcal{N}(\bar{\boldsymbol{u}}, \frac{\sigma^2}{n-1}I)} \left[ \mathbb{I} \left( f_{(t),\alpha}(\boldsymbol{u}, \boldsymbol{v}') \neq r \mid (\boldsymbol{u}, r), \boldsymbol{v}_{(0)} \right) \right],$$

*where $f_{(t),\alpha}$ represents the Gaussian Recommender System under the $\alpha$-poisoned condition, and $\mathbb{I}(\cdot)$ is an indicator function that returns 1 if the condition is true and 0 otherwise.*

For simplicity, we continue using the distribution of $\boldsymbol{v}_{(0)}$ from the definition. This allows us to further analyze the $\alpha$-poisoned recommendation error based on the distribution of $\boldsymbol{v}'_{(0)} = \frac{n}{n+n'}\boldsymbol{v}_{(0)} + \frac{1}{n+n'} \sum_{(\boldsymbol{u}', r') \in \mathcal{I}'} r'\boldsymbol{u}'$.

Then we extend Theorem 1 to $\alpha$-Poisoned Recommendation Error in the following theorem:

---

[3]Unless otherwise specified, $\|\cdot\|$ denotes the L2 norm $\|\cdot\|_2$ in this paper.

[4]To make poisoning attacks effective in single-item recommendation scenarios, attackers can directly inject users' initialized embeddings, which is equivalent to constructing interactions for different items in multi-item scenarios.

**Theorem 2.** *Consider a poisoned Gaussian Recommender System $f_{(t),\alpha}$, pre-trained for $t$ epochs using the standard loss function (Equation 2). Given a learning rate $\eta$, an adversarial training weight $\lambda$, and a perturbation magnitude $\epsilon$, when $\epsilon < \frac{\min(\|\boldsymbol{u}_{(t)}\|, \|\bar{\boldsymbol{u}}\|)}{\eta\lambda}$, and $\|\bar{\boldsymbol{u}}\| \gg \sigma$, the $\boldsymbol{\alpha}$-poisoned recommendation error for a genuine user $\boldsymbol{u}$ with rating $r$ at the $(t+1)^{\text{th}}$ epoch follows that:*

$$\mathbb{P}_{\boldsymbol{v}_{(0)} \sim \mathcal{N}(\bar{\boldsymbol{u}}, \frac{\sigma^2}{n-1} I)} \left[ f_{(t+1),\alpha}(\boldsymbol{u}, \boldsymbol{v}') \neq r \mid (\boldsymbol{u}, r) \right] > \mathbb{P}^{\text{adv}}_{\boldsymbol{v}_{(0)} \sim \mathcal{N}(\bar{\boldsymbol{u}}, \frac{\sigma^2}{n-1} I)} \left[ f_{(t+1),\alpha}(\boldsymbol{u}, \boldsymbol{v}') \neq r \mid (\boldsymbol{u}, r) \right].$$

For the proof, please refer to Appendix D.1.2. Combining Theorem 1 and Theorem 2, we find that adversarial training, i.e., ACF, lowers recommendation errors compared to traditional CF in both clean and poisoned data contexts. Accordingly, ACF achieves better performance and robustness.

## 3.2 How to Further Enhance Adversarial Collaborative Filtering

To explore mechanisms to further improve the effectiveness of ACF, we subsequently derive upper and lower bounds on the reduction of recommendation error between any two consecutive epochs after $t$ epochs of pre-training.

**Theorem 3.** *Consider a Gaussian Recommender System $f_{(t)}$ which has been pre-trained for $t$ epochs using standard loss (Equation 2) and subsequently trained on adversarial loss (Equation 3). For the $(t+k+1)^{\text{th}}$ epoch, let the reduction in recommendation error of user $\boldsymbol{u}$ with rating $r$ relative to the $(t+k)^{\text{th}}$ epoch from adversarial loss be denoted by:*

$$\Delta^{\text{adv}}_{(t+k+1)} \mathbb{P}^{\text{adv}}_{\boldsymbol{v}_{(0)} \sim \mathcal{N}(\bar{\boldsymbol{u}}, \frac{\sigma^2}{n-1} I)}[f(\boldsymbol{u}, \boldsymbol{v}) \neq r \mid (\boldsymbol{u}, r)] =$$

$$\mathbb{P}^{\text{adv}}_{\boldsymbol{v}_{(0)} \sim \mathcal{N}(\bar{\boldsymbol{u}}, \frac{\sigma^2}{n-1} I)} \left[ f_{(t+k)}(\boldsymbol{u}, \boldsymbol{v}) \neq r \mid (\boldsymbol{u}, r) \right] - \mathbb{P}^{\text{adv}}_{\boldsymbol{v}_{(0)} \sim \mathcal{N}(\bar{\boldsymbol{u}}, \frac{\sigma^2}{n-1} I)} \left[ f_{(t+k+1)}(\boldsymbol{u}, \boldsymbol{v}) \neq r \mid (\boldsymbol{u}, r) \right].$$

*Given a learning rate $\eta$, an adversarial training weight $\lambda$, and a perturbation magnitude $\epsilon$, when $\epsilon < \frac{\min(\|\boldsymbol{u}_{(t+k)}\|, \|\bar{\boldsymbol{u}}\|)}{\eta\lambda}$, and $\|\bar{\boldsymbol{u}}\| \gg \sigma$, it follows that:*

$$\Delta^{\text{adv}}_{(t+k+1)} \mathbb{P}^{\text{adv}}_{\boldsymbol{v}_{(0)} \sim \mathcal{N}(\bar{\boldsymbol{u}}, \frac{\sigma^2}{n-1} I)}[f(\boldsymbol{u}, \boldsymbol{v}) \neq r \mid (\boldsymbol{u}, r)] \geq$$

$$\Phi\left( \frac{\sqrt{n-1}}{\sigma} \left( \|\bar{\boldsymbol{u}}\| + \eta(\|\bar{\boldsymbol{u}}\|^2 + \frac{d\sigma^2}{n-1})\Psi(\boldsymbol{u}, t+k) \right) \right) - \Phi\left( \frac{\sqrt{n-1}}{\sigma} \|\bar{\boldsymbol{u}}\| \right),$$

$$\Delta^{\text{adv}}_{(t+k+1)} \mathbb{P}^{\text{adv}}_{\boldsymbol{v}_{(0)} \sim \mathcal{N}(\bar{\boldsymbol{u}}, \frac{\sigma^2}{n-1} I)}[f(\boldsymbol{u}, \boldsymbol{v}) \neq r \mid (\boldsymbol{u}, r)] \leq 2\Phi\left( \frac{\sqrt{n-1}\eta}{2\sigma}(\|\bar{\boldsymbol{u}}\|^2 + \frac{d\sigma^2}{n-1})\Psi(\boldsymbol{u}, t+k) \right) - 1,$$

*where $d$ is the embedding dimension, and $\Phi(\cdot)$ denotes the cumulative distribution function (CDF) of the standard Gaussian distribution, and $\Psi(\boldsymbol{u}, t+k)$ is defined as:*

$$\Psi(\boldsymbol{u}, t+k) = (1+\lambda)\gamma^{\boldsymbol{u}}_{(t+k)} \frac{C_{t+k}}{\|\boldsymbol{u}_{(t+k)}\|}, \quad \text{where} \quad \gamma^{\boldsymbol{u}}_{(t+k)} = \left( 1 - \frac{\eta\lambda\epsilon}{\|\boldsymbol{u}_{(t+k)}\|} \right)^{-1}, \tag{4}$$

*where $C_{t+k}$ is a constant at the $(t+k)^{\text{th}}$ epoch.*

For the proof, please refer to Appendix D.2.1. In light of Theorem 3, given a learning rate $\eta$ and an adversarial training weight $\lambda$, we can establish the following: (1) When the conditions, i.e., $\epsilon < \frac{\min(\|\boldsymbol{u}_{(t+k)}\|, \|\bar{\boldsymbol{u}}\|)}{\eta\lambda}$ and $\|\bar{\boldsymbol{u}}\| \gg \sigma$, are satisfied, the error reduction for ACF can be both upper and lower bounded. (2) Increasing the perturbation magnitude $\epsilon$ under the above conditions can further improve these bounds, thus benefiting ACF's effectiveness.

Then, similarly, we extend Theorem 3 to the $\boldsymbol{\alpha}$-poisoned context.

**Theorem 4.** *Consider a poisoned Gaussian Recommender System $f_{(t),\alpha}$ which has been pre-trained for $t$ epochs using standard loss (Equation 2) and subsequently trained on adversarial loss (Equation 3). For the $(t+k+1)^{\text{th}}$ epoch, let the reduction in $\boldsymbol{\alpha}$-poisoned recommendation error of a genuine user $\boldsymbol{u}$ with rating $r$ relative to the $(t+k)^{\text{th}}$ epoch from adversarial loss be denoted by $\Delta^{\text{adv}}_{(t+k+1)} \mathbb{P}^{\text{adv}}_{\boldsymbol{v}_{(0)} \sim \mathcal{N}(\bar{\boldsymbol{u}}, \frac{\sigma^2}{n-1} I)}[f_\alpha(\boldsymbol{u}, \boldsymbol{v}') \neq r \mid (\boldsymbol{u}, r)]$. Let $\beta = \frac{n'}{n}\sqrt{d}\alpha + \|\bar{\boldsymbol{u}}\|$ and $\tau = 2nn'\alpha\|\bar{\boldsymbol{u}}\|_0$, where $d$ is the embedding dimension, and given a learning rate $\eta$, an adversarial training weight $\lambda$,*

*and a perturbation magnitude $\epsilon$, when $\epsilon < \frac{\min(\|\boldsymbol{u}_{(t+k)}\|, \|\bar{\boldsymbol{u}}\|)}{\eta\lambda}$, and $\|\bar{\boldsymbol{u}}\| \gg \sigma$, it follows that:*

$$\Delta^{\text{adv}}_{(t+k+1)}\mathbb{P}^{\text{adv}}_{\boldsymbol{v}_{(0)}\sim\mathcal{N}(\bar{\boldsymbol{u}},\frac{\sigma^2}{n-1}I)}[f_\alpha(\boldsymbol{u},\boldsymbol{v}') \neq r \mid (\boldsymbol{u},r)] \quad >$$

$$\Phi\left(\frac{\sqrt{n-1}}{\sigma}\left(\beta + \eta\left(\frac{n^2\|\bar{\boldsymbol{u}}\|^2 - \tau}{n(n+n')} + \frac{nd\sigma^2}{(n-1)(n+n')}\right)\Psi(\boldsymbol{u},t+k)\right)\right) - \Phi\left(\frac{\sqrt{n-1}}{\sigma}\beta\right),$$

$$\Delta^{\text{adv}}_{(t+k+1)}\mathbb{P}^{\text{adv}}_{\boldsymbol{v}_{(0)}\sim\mathcal{N}(\bar{\boldsymbol{u}},\frac{\sigma^2}{n-1}I)}[f_\alpha(\boldsymbol{u},\boldsymbol{v}') \neq r \mid (\boldsymbol{u},r)] \quad \leq$$

$$2\Phi\left(\frac{\sqrt{n-1}\eta}{2\sigma}\left(\frac{n^2\|\bar{\boldsymbol{u}}\|^2 + (n')^2 d\alpha^2 + \tau}{n(n+n')} + \frac{nd\sigma^2}{(n-1)(n+n')}\right)\Psi(\boldsymbol{u},t+k)\right) - 1,$$

*where $\Phi()$ denotes the cumulative distribution function (CDF) of the standard Gaussian distribution, $n'$ is the number of fake users, and $\Psi(\cdot)$ is defined in Equation 4.*

For the proof, please refer to Appendix D.2.2. From Theorem 4, we understand that increasing $\Psi(\boldsymbol{u},t+k)$ can further improve both the upper and lower bounds of error reduction, thereby mitigating the negative impact of poisons. Specifically, this involves the same mechanism as in the clean data context: increasing the perturbation magnitude $\epsilon$ within $\epsilon < \frac{\min(\|\boldsymbol{u}_{(t+k)}\|, \|\bar{\boldsymbol{u}}\|)}{\eta\lambda}$.

In conclusion, the theorems in this section indicate that for each user $\boldsymbol{u}$, when the user's perturbation magnitude meets $\epsilon < \frac{\min(\|\boldsymbol{u}_{(t+k)}\|, \|\bar{\boldsymbol{u}}\|)}{\eta\lambda}$, we have the following: (1) ACF is theoretically shown to be more effective than traditional CF, and (2) Increasing the user's perturbation magnitude during training as much as possible can further improve both the performance and robustness of ACF. These theoretical understandings can further benefit exploring and fully unleashing the potential of ACF.

## 4 Methodology

To extend theoretical understandings from the simple CF scenario to more practical scenarios, such as multi-item recommendations with Bayesian Personalized Ranking (BPR) [31], which is a mainstream loss function used in CF recommendations, we first conduct a preliminary experiment shown in Figure 1. Using Matrix Factorization [2] on the Gowalla dataset [32], we observe results similar to those in Theorem 3 and Theorem 4: NDCG@20 for users improves within their maximum magnitudes, i.e., constraints, but significantly declines once these constraints are surpassed. Based on the theoretical understandings provided in Section 3, we derive the following corollary to identify the maximum perturbation magnitude for each user in practical CF scenarios.

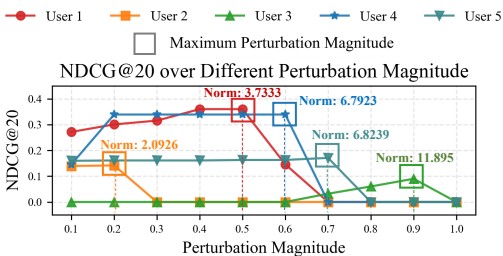

Figure 1: NDCG@20 across various perturbation magnitudes for five users (subject to Random Attacks [30]).

**Corollary 1.** *Given any dot-product-based loss function $\mathcal{L}(\Theta)$, within the framework of Adversarial Collaborative Filtering as defined in Equation 1, the maximum perturbation magnitude $\epsilon^{(\boldsymbol{u})}_{(t),\max}$ for user $\boldsymbol{u}$ at the $t^{\text{th}}$ epoch is positively related to $\|\boldsymbol{u}_{(t)}\|$.*

For the proof of Corollary 1, please refer to Appendix D.3. According to Corollary 1, we observe that for a user $\boldsymbol{u}$, the larger $\|\boldsymbol{u}\|$, the greater the maximum perturbation magnitude. Considering that maximum perturbation magnitudes will be affected by other factors in the actual training process, to ensure training stability, we decompose $\epsilon^{(\boldsymbol{u})}_{(t),\max}$ for a user $\boldsymbol{u}$ at epoch $t$ into two components: the uniform perturbation magnitude $\rho$, applicable to all users, and a user-specific perturbation coefficient $c(\boldsymbol{u},t)$, expressed as:

$$\epsilon^{(\boldsymbol{u})}_{(t),\max} = \rho \cdot c(\boldsymbol{u},t). \tag{5}$$

According to Corollary 1, $c(\boldsymbol{u},t)$ provides coefficients positively related to users' embedding scales. To avoid training instability caused by extreme scale values, we map $c(\boldsymbol{u},t)$ into the interval $(0,1)$,

defined by:

$$c(\boldsymbol{u}, t) = \text{sig}\left(\frac{\|\boldsymbol{u}_{(t)}\| - \overline{\|\boldsymbol{u}_{(t)}\|}}{\|\boldsymbol{u}_{(t)}\|}\right),$$

where $\overline{\|\boldsymbol{u}_{(t)}\|}$ represents the average norm of all user embeddings at epoch $t$, and $\text{sig}(\cdot)$ denotes the sigmoid function. Consequently, the loss function for our method, Personalized Magnitude Adversarial Collaborative Filtering (PamaCF), is defined as:

$$\begin{aligned} \mathcal{L}_{\text{PamaCF}}(\Theta) &= \mathcal{L}(\Theta) + \lambda \mathcal{L}(\Theta + \Delta^{\text{PamaCF}}), \\ \text{where} \quad \Delta^{\text{PamaCF}} &= \arg \max_{\Delta,\, \|\Delta_u\| \le \rho \cdot c(\boldsymbol{u}, t)} \mathcal{L}(\Theta + \Delta), \end{aligned} \tag{6}$$

where $\lambda$ is the weight of adversarial training, $\rho$ represents the uniform perturbation magnitude for all users, and $\Delta_u$ is the perturbation relative to user $u$. To maximize the perturbation magnitude for each user within $\rho c(\boldsymbol{u}, t)$, we use the perturbation along the gradient direction of the user's adversarial loss with a step length of $\rho c(\boldsymbol{u}, t)$ as $\Delta_u$. The specific algorithm process is detailed in Appendix B.

## 5 Experiments

In this section, we conduct extensive experiments to address the following research questions (**RQs**):

- **RQ1:** Can PamaCF further improve the performance and robustness of traditional ACF?
- **RQ2:** Why does PamaCF perform better than traditional ACF?
- **RQ3:** How do hyper-parameters affect PamaCF?

### 5.1 Experimental Setup

In this section, we briefly introduce the experimental settings. For detailed information, including dataset preprocessing, comprehensive baseline descriptions, and implementation details, please refer to Appendix C.1.

**Datasets**. We employ three common benchmarks: the *Gowalla* check-in dataset [32], the *Yelp2018* business dataset, and the *MIND* news recommendation dataset [33].

**Attack Methods**. We employ both heuristic (Random Attack [30], Bandwagon Attack [34]) and optimization-based (Rev Attack [7], DP Attack [6]) attack methods within a black-box context, where the attacker does not have access to the internal architecture or parameters of the target model.

**Defense Baselines**. We incorporate a variety of defense methods, including detection-based approaches (GraphRfi [12] and LLM4Dec [13]), adversarial collaborative filtering methods (APR [15] and SharpCF [20]), and a denoise-based strategy (StDenoise [35, 19]). In our study, we employ three common backbone recommendation models, Matrix Factorization (MF) [2], LightGCN [3], and NeurMF [36].

**Evaluation Metrics**. The primary metrics for assessing recommendation performance are the top-$k$ metrics: Recall@$k$ and NDCG@$k$, as documented in [3, 8, 37]. To quantify the success ratio of attacks, we utilize T-HR@$k$ and T-NDCG@$k$ to measure the performance of target items within the top-$k$ recommendations [7, 6, 13], as:

$$\text{T-HR@}k = \frac{1}{|\mathcal{T}|} \sum_{tar \in \mathcal{T}} \frac{\sum_{u \in \mathcal{U} \setminus \mathcal{U}_{tar}} \mathbb{I}\left(tar \in L_{u, 1:k}\right)}{|\mathcal{U} \setminus \mathcal{U}_{tar}|}, \tag{7}$$

where $\mathcal{T}$ is the set of target items, $\mathcal{U}_{tar}$ denotes the set of genuine users who have interacted with target items $tar$, $L_{u, 1:k}$ represents the top-$k$ list of recommendations for user $u$, and $\mathbb{I}(\cdot)$ is the indicator function that returns 1 if the condition is true. The T-NDCG@$k$ mirrors T-HR@$k$, serving as the target item-specific version of NDCG@$k$.

### 5.2 Performance Comparison (RQ1)

In this section, we answer **RQ1**. We focus on two key aspects: the recommendation performance and the robustness against poisoning attacks.

Table 1: Recommendation Performance

| Model | Clean (%) | | Random Attack (%) | | Bandwagon Attack (%) | | DP Attack (%) | | Rev Attack (%) | |
|---|---|---|---|---|---|---|---|---|---|---|
| (Dataset) | Recall@20 | NDCG@20 | Recall@20 | NDCG@20 | Recall@20 | NDCG@20 | Recall@20 | NDCG@20 | Recall@20 | NDCG@20 |
| **MF** (Gowalla) | 11.35 ± 0.09 | 7.16 ± 0.03 | 11.31 ± 0.08 | 7.20 ± 0.06 | 11.24 ± 0.08 | 7.11 ± 0.04 | 10.72 ± 0.11 | 8.17 ± 0.08 | 10.70 ± 0.09 | 8.19 ± 0.04 |
| +StDenoise | 10.48 ± 0.10 | 8.07 ± 0.10 | 10.46 ± 0.09 | 8.07 ± 0.07 | 10.41 ± 0.06 | 8.04 ± 0.02 | 10.53 ± 0.13 | 8.12 ± 0.09 | 10.57 ± 0.05 | 8.19 ± 0.04 |
| +GraphRfi | 10.43 ± 0.07 | 7.97 ± 0.03 | 10.34 ± 0.08 | 7.89 ± 0.06 | 10.30 ± 0.06 | 7.85 ± 0.06 | 10.40 ± 0.11 | 7.94 ± 0.08 | 10.50 ± 0.09 | 8.01 ± 0.07 |
| +APR | 13.06 ± 0.06 | 10.65 ± 0.06 | 12.93 ± 0.04 | 10.52 ± 0.01 | 12.90 ± 0.07 | 10.50 ± 0.03 | 12.95 ± 0.06 | 10.59 ± 0.06 | 13.13 ± 0.05 | 10.72 ± 0.06 |
| +SharpCF | 13.20 ± 0.07 | 10.02 ± 0.09 | 13.19 ± 0.08 | 10.03 ± 0.07 | 13.03 ± 0.06 | 9.89 ± 0.05 | 13.27 ± 0.14 | 10.08 ± 0.10 | 13.22 ± 0.09 | 10.10 ± 0.04 |
| +PamaCF | **13.48 ± 0.02** | **10.94 ± 0.05** | **13.37 ± 0.07** | **10.84 ± 0.03** | **13.35 ± 0.03** | **10.82 ± 0.02** | **13.44 ± 0.08** | **10.93 ± 0.04** | **13.61 ± 0.05** | **11.06 ± 0.08** |
| Gain | +2.10% ↑ | +2.76% ↑ | +1.40% ↑ | +3.04% ↑ | +2.53% ↑ | +3.07% ↑ | +1.31% ↑ | +3.23% ↑ | +2.96% ↑ | +3.15% ↑ |
| Gain w.r.t. MF | +18.75% ↑ | +52.84% ↑ | +18.27% ↑ | +50.64% ↑ | +18.83% ↑ | +52.29% ↑ | +25.39% ↑ | +33.76% ↑ | +27.18% ↑ | +35.05% ↑ |
| **MF** (Yelp2018) | 3.76 ± 0.03 | 2.97 ± 0.04 | 3.73 ± 0.02 | 2.93 ± 0.01 | 3.74 ± 0.04 | 2.95 ± 0.03 | 3.87 ± 0.04 | 3.03 ± 0.03 | 3.81 ± 0.04 | 3.03 ± 0.04 |
| +StDenoise | 3.41 ± 0.08 | 2.61 ± 0.09 | 3.29 ± 0.04 | 2.50 ± 0.03 | 3.32 ± 0.06 | 2.52 ± 0.05 | 3.38 ± 0.06 | 2.58 ± 0.06 | 3.38 ± 0.10 | 2.59 ± 0.10 |
| +GraphRfi | 3.73 ± 0.05 | 2.94 ± 0.03 | 3.66 ± 0.04 | 2.90 ± 0.03 | 3.64 ± 0.05 | 2.88 ± 0.03 | 3.76 ± 0.06 | 2.93 ± 0.05 | 3.72 ± 0.05 | 2.95 ± 0.04 |
| +APR | 4.09 ± 0.02 | 3.20 ± 0.02 | 4.04 ± 0.02 | 3.16 ± 0.02 | 4.08 ± 0.03 | 3.19 ± 0.03 | 4.01 ± 0.06 | 3.15 ± 0.04 | 4.06 ± 0.03 | 3.20 ± 0.02 |
| +SharpCF | 3.93 ± 0.04 | 3.11 ± 0.05 | 3.88 ± 0.01 | 3.06 ± 0.02 | 3.91 ± 0.05 | 3.08 ± 0.03 | 4.03 ± 0.03 | 3.16 ± 0.04 | 3.97 ± 0.05 | 3.16 ± 0.05 |
| +PamaCF | **4.18 ± 0.02** | **3.29 ± 0.02** | **4.13 ± 0.01** | **3.25 ± 0.01** | **4.19 ± 0.04** | **3.29 ± 0.03** | **4.25 ± 0.04** | **3.33 ± 0.04** | **4.27 ± 0.03** | **3.37 ± 0.03** |
| Gain | +2.20% ↑ | +2.75% ↑ | +2.33% ↑ | +2.91% ↑ | +2.70% ↑ | +3.01% ↑ | +5.30% ↑ | +5.24% ↑ | +5.04% ↑ | +5.22% ↑ |
| Gain w.r.t. MF | +11.22% ↑ | +10.63% ↑ | +10.72% ↑ | +10.84% ↑ | +11.91% ↑ | +11.60% ↑ | +9.88% ↑ | +9.84% ↑ | +11.91% ↑ | +11.36% ↑ |
| **MF** (MIND) | 1.20 ± 0.01 | 0.68 ± 0.00 | 1.19 ± 0.01 | 0.67 ± 0.01 | 1.19 ± 0.02 | 0.68 ± 0.00 | 1.20 ± 0.00 | 0.69 ± 0.01 | OOM | OOM |
| +StDenoise | 1.13 ± 0.01 | 0.63 ± 0.01 | 1.12 ± 0.01 | 0.63 ± 0.00 | 1.12 ± 0.01 | 0.63 ± 0.00 | 1.13 ± 0.01 | 0.64 ± 0.01 | OOM | OOM |
| +GraphRfi | 1.20 ± 0.01 | 0.67 ± 0.00 | 1.19 ± 0.01 | 0.67 ± 0.00 | 1.19 ± 0.01 | 0.67 ± 0.01 | 1.20 ± 0.02 | 0.67 ± 0.01 | OOM | OOM |
| +LLM4Dec | 1.20 ± 0.01 | 0.68 ± 0.00 | 1.19 ± 0.01 | 0.67 ± 0.01 | 1.19 ± 0.01 | 0.68 ± 0.00 | 1.19 ± 0.00 | 0.68 ± 0.00 | OOM | OOM |
| +APR | 1.22 ± 0.01 | 0.68 ± 0.01 | 1.26 ± 0.02 | 0.71 ± 0.01 | 1.21 ± 0.01 | 0.69 ± 0.00 | 1.21 ± 0.01 | 0.70 ± 0.01 | OOM | OOM |
| +PamaCF | **1.30 ± 0.01** | **0.73 ± 0.00** | **1.27 ± 0.01** | **0.72 ± 0.00** | **1.27 ± 0.01** | **0.72 ± 0.00** | **1.30 ± 0.01** | **0.74 ± 0.01** | OOM | OOM |
| Gain | +7.06% ↑ | +7.53% ↑ | +0.71% ↑ | +0.69% ↑ | +5.02% ↑ | +5.12% ↑ | +6.90% ↑ | +6.26% ↑ | - | - |
| Gain w.r.t. MF | +8.30% ↑ | +8.49% ↑ | +6.81% ↑ | +7.00% ↑ | +6.80% ↑ | +6.66% ↑ | +7.79% ↑ | +7.49% ↑ | - | - |

[1] The Rev attack method could not be executed on the dataset due to memory constraints, resulting in an out-of-memory error.

Table 2: Robustness against target items promotion

| Dataset | Model | Random Attack(%) | | Bandwagon Attack(%) | | DP Attack(%) | | Rev Attack(%) | |
|---|---|---|---|---|---|---|---|---|---|
| | | T-HR@50[1] | T-NDCG@50 | T-HR@50 | T-NDCG@50 | T-HR@50 | T-NDCG@50 | T-HR@50 | T-NDCG@50 |
| **Gowalla** | **MF** | 0.148 ± 0.030 | 0.036 ± 0.008 | 0.120 ± 0.027 | 0.029 ± 0.007 | 0.201 ± 0.020 | 0.051 ± 0.005 | 0.246 ± 0.097 | 0.061 ± 0.027 |
| | +StDenoise | 0.200 ± 0.049 | 0.050 ± 0.012 | 0.165 ± 0.034 | 0.038 ± 0.008 | 0.292 ± 0.034 | 0.074 ± 0.010 | 0.355 ± 0.126 | 0.084 ± 0.030 |
| | +GraphRfi | 0.159 ± 0.061 | 0.042 ± 0.015 | 0.154 ± 0.038 | 0.036 ± 0.009 | 0.174 ± 0.038 | 0.043 ± 0.009 | 0.206 ± 0.042 | 0.050 ± 0.010 |
| | +APR | 0.201 ± 0.091 | 0.054 ± 0.026 | 0.184 ± 0.067 | 0.047 ± 0.015 | 0.034 ± 0.021 | 0.006 ± 0.004 | 0.261 ± 0.063 | 0.067 ± 0.018 |
| | +SharpCF | 0.204 ± 0.037 | 0.049 ± 0.010 | 0.169 ± 0.031 | 0.041 ± 0.008 | 0.303 ± 0.024 | 0.077 ± 0.006 | 0.350 ± 0.111 | 0.087 ± 0.031 |
| | +PamaCF | **0.070 ± 0.028** | **0.017 ± 0.007** | **0.064 ± 0.026** | **0.015 ± 0.006** | **0.021 ± 0.011** | **0.004 ± 0.002** | **0.079 ± 0.039** | **0.019 ± 0.009** |
| | Gain[2] | +52.72% ↑ | +51.95% ↑ | +46.19% ↑ | +47.01% ↑ | +36.33% ↑ | +33.02% ↑ | +61.41% ↑ | +62.51% ↑ |
| **Yelp2018** | **MF** | 0.035 ± 0.007 | 0.010 ± 0.002 | 0.073 ± 0.032 | 0.020 ± 0.009 | 0.223 ± 0.040 | 0.049 ± 0.009 | 0.153 ± 0.025 | 0.040 ± 0.006 |
| | +StDenoise | 0.108 ± 0.038 | 0.027 ± 0.010 | 0.181 ± 0.046 | 0.043 ± 0.011 | 0.376 ± 0.198 | 0.077 ± 0.039 | 0.331 ± 0.145 | 0.075 ± 0.031 |
| | +GraphRfi | 0.032 ± 0.009 | 0.009 ± 0.003 | 0.058 ± 0.014 | 0.015 ± 0.003 | 0.200 ± 0.041 | 0.043 ± 0.010 | 0.129 ± 0.027 | 0.031 ± 0.007 |
| | +APR | 0.012 ± 0.007 | 0.004 ± 0.002 | 0.057 ± 0.047 | 0.013 ± 0.011 | 0.185 ± 0.038 | 0.040 ± 0.009 | 0.098 ± 0.048 | 0.022 ± 0.011 |
| | +SharpCF | 0.034 ± 0.007 | 0.010 ± 0.002 | 0.072 ± 0.029 | 0.019 ± 0.008 | 0.226 ± 0.041 | 0.050 ± 0.010 | 0.152 ± 0.025 | 0.040 ± 0.006 |
| | +PamaCF | **0.010 ± 0.006** | **0.004 ± 0.002** | **0.028 ± 0.022** | **0.007 ± 0.005** | **0.135 ± 0.033** | **0.027 ± 0.007** | **0.045 ± 0.021** | **0.010 ± 0.004** |
| | Gain | +14.29% ↑ | +18.22% ↑ | +50.33% ↑ | +45.73% ↑ | +27.41% ↑ | +30.62% ↑ | +54.24% ↑ | +53.25% ↑ |
| **MIND** | **MF** | 0.032 ± 0.007 | 0.010 ± 0.002 | 0.169 ± 0.017 | 0.055 ± 0.005 | 0.023 ± 0.013 | 0.005 ± 0.003 | OOM | OOM |
| | +StDenoise | 0.036 ± 0.006 | 0.013 ± 0.004 | 0.040 ± 0.006 | 0.020 ± 0.004 | 0.010 ± 0.003 | 0.002 ± 0.001 | OOM | OOM |
| | +GraphRfi | 0.031 ± 0.006 | 0.010 ± 0.002 | 0.189 ± 0.015 | 0.059 ± 0.005 | 0.020 ± 0.009 | 0.004 ± 0.002 | OOM | OOM |
| | +LLM4Dec | 0.020 ± 0.001 | **0.004 ± 0.000** | 0.083 ± 0.009 | 0.025 ± 0.003 | 0.019 ± 0.010 | 0.004 ± 0.002 | OOM | OOM |
| | +APR | 0.083 ± 0.013 | 0.035 ± 0.006 | 0.068 ± 0.005 | 0.023 ± 0.002 | 0.008 ± 0.007 | 0.002 ± 0.001 | OOM | OOM |
| | +PamaCF | **0.012 ± 0.002** | **0.005 ± 0.001** | **0.016 ± 0.002** | **0.006 ± 0.001** | **0.000 ± 0.000** | **0.000 ± 0.000** | OOM | OOM |
| | Gain | +39.80% ↑ | - | +60.15% ↑ | +72.10% ↑ | +95.02% ↑ | +94.32% ↑ | - | - |

[1] Target Item Hit Ratio (Equation 7); T-HR@50 and T-NDCG@50 of all target items on clean datasets are 0.000.
[2] The relative percentage increase of PamaCF's metrics to the best value of other baselines' metrics, i.e., $(\min(\text{T-HR}_{\text{Beslines}}) - \text{T-HR}_{\text{PamaCF}}) / \min(\text{T-HR}_{\text{Beslines}})$. Notably, only **three decimal places** are presented due to space limitations, though the actual ranking and calculations utilize the **full precision** of the data.

**Recommendation Performance**. We assess the efficacy of PamaCF in both clean and poisoning data contexts, focusing on the performance of recommender systems, as presented in Table 1. The denoise-based defense method, which does not directly defend against poisoning attacks but rather purifies noisy interactions, fails to improve recommendation performance in most cases. Detection-based methods, such as GraphRfi and LLM4Dec, exhibit some misclassifications of fake and genuine users, leading to a decline in recommendation performance.

In contrast, we observe a notable enhancement in recommendation quality when ACF methods (APR, SharpCF, and PamaCF) are utilized. This finding is consistent with results from previous studies [15, 20] and aligns with our prior theoretical analysis. Among the defense methods, PamaCF stands out, achieving the most significant improvements in recommendation performance compared to the backbone model and other baseline approaches. Specifically, PamaCF increases Recall@20 and NDCG@20 by 13.84% and 22.04% in average, respectively, compared to the backbone model.

**Robustness Against Poisoning Attacks**. We evaluate the capability of PamaCF in defending against poisoning attacks by examining the attack success ratio. Our experiments specifically target items with notably low popularity, as indicated by T-HR@50 and T-NDCG@50 scores of 0.0 when no attacks are present. Lower scores for T-HR@50 and T-NDCG@50 indicate stronger defense capabilities.

Table 2 presents the results, indicating that the purely denoise-based defense method is generally ineffective against most attacks and may even increase the attack's success ratio in some instances.

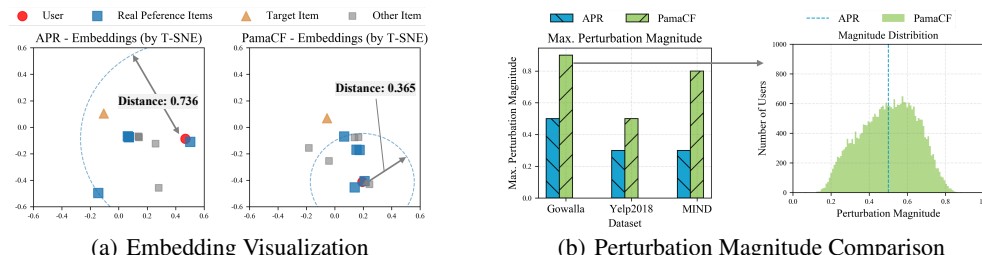

(a) Embedding Visualization        (b) Perturbation Magnitude Comparison

Figure 2: (a) PamaCF brings real preference items closer; (b) PamaCF achieves larger magnitudes.

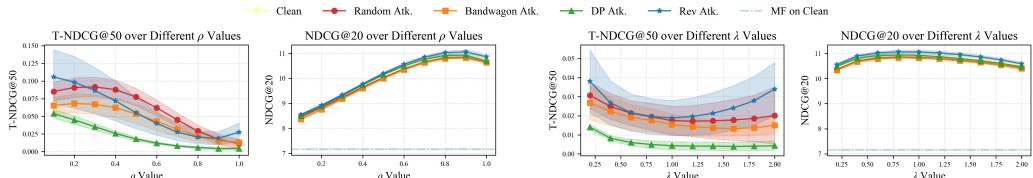

Figure 3: Left: Analysis of Hyper-Parameters $\rho$; Right: Analysis of Hyper-Parameters $\lambda$.

Detection-based methods, such as GraphRfi and LLM4Dec, show robust defense against attacks similar to their training data, i.e., random attacks. However, the effectiveness of GraphRfi declines against other attack types. In contrast, ACF methods demonstrate stable defense capabilities across various attacks. Specifically, PamaCF significantly reduces the success ratio of attacks, decreasing T-HR@50 and T-NDCG@50 by 49.92% and 43.73% in average, respectively, compared to the best baseline. These results highlight PamaCF's advanced defense capabilities against various attacks.

Additionally, PamaCF's defense effectiveness against attacks targeting popular items is further evaluated. The corresponding results for LightGCN [3] and NeuMF [36], along with the recommendation performance at top-10, are also presented. All supplementary results are in Appendix C.2.

## 5.3 Augmentation Analysis (RQ2)

In this section, we address **RQ2** by exploring why PamaCF can outperform traditional ACF (especially APR [15]) through embedding visualization and perturbation magnitude comparison.

**Embedding Visualization**. We randomly select a user and project the normalized embeddings of the user, real preference items, the target item given by attacks, and other items in the user's top-10 recommendation list into a two-dimensional space using T-SNE [38], as shown in Figure 2(a). We observe that PamaCF can bring real preference items closer, reducing the distance from the farthest real preference item from 0.736 to 0.365, while leading the target item farther away from all the real preference items. PamaCF's personalized perturbation magnitude lowers the ranking of both the target item and other items, thus improving robustness and performance.

**Perturbation Magnitude Comparison**. We compare the maximum perturbation magnitudes of APR and PamaCF, i.e., $\epsilon$ in Equation 1 for APR and $\rho$ in Equation 6 for PamaCF. Both $\epsilon$ and $\rho$ are selected through hyper-parameter tuning from $\{0.1, 0.2, \ldots, 1.0\}$. In the left part of Figure 2(b), we observe that PamaCF finds a higher perturbation magnitude. Additionally, the right portion of Figure 2(b) illustrates the distribution of personalized perturbation magnitudes across all users. These varying magnitudes for different users contribute to the improved effectiveness of PamaCF.

## 5.4 Hyper-Parameters Analysis (RQ3)

In this section, we answer **RQ3** by exploring the effects of the hyperparameters, magnitude $\rho$ and adversarial training weight $\lambda$, as defined in Equation 6. The results are illustrated in Figure 3.

**Analysis of Hyper-Parameters $\rho$**. With $\lambda$ fixed at 1.0, we vary $\rho$ from 0.1 to 1.0 in increments of 0.1. Our findings demonstrate a significant improvement in both robustness and performance as $\rho$ increases. Notably, even when $\rho$ exceeds 0.1, there is an enhancement in recommendation

performance compared to that of the backbone model, i.e., MF, with the range between 0.7 and 0.9 yielding the most significant enhancements.

**Analysis of Hyper-Parameters** $\lambda$. With $\rho$ set at 0.9, we adjust $\lambda$ from 0.2 to 2.0 in increments of 0.2. The analysis indicates that the defensive ability becomes stable once $\lambda$ surpasses 1.0 in most attacks. However, setting $\lambda$ too high gradually diminishes the recommendation performance of PamaCF. Despite this, the performance of PamaCF remains considerably improved compared to MF.

## 6 Conclusion

In this work, we theoretically analyze why Adversarial Collaborative Filtering (ACF) enhances both the performance and robustness of Collaborative Filtering (CF) systems against poisoning attacks. Additionally, by establishing bounds for reductions in recommendation error during ACF's optimization process, we discover that applying personalized perturbation magnitudes for users based on their embedding scales can significantly improve ACF's effectiveness. Leveraging these theoretical understandings, we introduce Personalized Magnitude Adversarial Collaborative Filtering (PamaCF). Comprehensive experiments confirm that PamaCF effectively defends against various attacks and significantly enhances the quality of recommendations.

**Limitations**. Our study identifies several limitations that require further investigation. Firstly, our theoretical analysis is based on certain assumptions, specifically with the Gaussian Recommender System. We intend to relax these assumptions in future work. Secondly, this study only examines adversarial training within CF recommendations. In future research, we plan to extend our analysis to include more recommendation scenarios, such as sequential recommendations.

**Broader Impacts**. Our work focuses on enhancing both the performance and robustness of recommender systems against poisoning attacks, thereby benefiting the overall development of recommender systems. We do not foresee any negative impacts resulting from our work.

## Acknowledgements

This work is funded by the National Key R&D Program of China (2022YFB3103700, 2022YFB3103701), the Strategic Priority Research Program of the Chinese Academy of Sciences under Grant No. XDB0680101, and the National Natural Science Foundation of China under Grant Nos. 62102402, 62272125, U21B2046. Huawei Shen is also supported by Beijing Academy of Artificial Intelligence (BAAI).

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

# A Related Work

## A.1 Collaborative Filtering

Collaborative Filtering (CF) has become a cornerstone of modern recommender systems, evidenced by its widespread application in various studies [3, 39, 40]. The fundamental premise of CF is that users with similar preferences are likely to exhibit similar behaviors, which can be leveraged to predict future recommendations [41]. A principal approach within CF is Matrix Factorization, which learns latent embeddings of users and items by decomposing the observed interaction matrix [2].

With the advent of deep learning, neural CF models have emerged, designed to capture more complex patterns in user preferences. For example, CDL [42] merges auxiliary item information through neural networks into CF, addressing challenges associated with data sparsity. Additionally, NCF [36] replaces the traditional dot product with a neural network, enhancing the modeling of user-item interactions. More recently, Graph Neural Networks have prompted the development of graph-based CF models, such as NGCF [37] and LightGCN [3], which have shown remarkable efficacy in recommender systems. However, despite these technological advances, susceptibility to poisoning attacks remains a significant challenge, compromising the robustness of these systems [8].

## A.2 Poisoning Attacks against Recommender Systems

Poisoning attacks within recommender systems involve injecting fake users into the training data to manipulate the exposure of certain items. Initial research predominantly focused on rule-based heuristic attacks, where profiles for these fake users were constructed using predetermined heuristic rules [30, 34, 43, 44]. For example, the Random Attack [30] generated fake users interacting with targeted items alongside a random selection of other items. In contrast, the Bandwagon Attack [34] generated fake user interactions to include targeted items and others selected for their high popularity.

As the technique of attacks has evolved, recent studies have shifted towards optimization-based methods for generating fake users [7, 6, 29, 45, 46, 47, 48, 49, 50, 51, 52]. For instance, the Rev Attack [7] formalizes the attack as a bi-level optimization problem, addressed using gradient-based techniques. Similarly, the DP Attack [6] specifically targets deep learning-based recommender systems.

## A.3 Robust Recommender Systems

Mainstream strategies for enhancing the robustness of CF systems against poisoning attacks broadly categorize into two main approaches [8]: (1) detecting and removing fake users [9, 10, 11, 12, 13, 14, 53], and (2) developing robust models via adversarial training, i.e., Adversarial Collaborative Filtering (ACF) [15, 16, 17, 18, 19, 20, 27].

Detection-based strategies focus either on pre-identifying and removing potential fake users from the dataset [9, 10, 11, 14] or on mitigating their influence during the training phase [12, 13]. These methods often rely on specific assumptions about the attacks [9, 12] or require supervised data regarding attacks [10, 11, 12, 13]. Among these, LoRec [13] utilizes large language models to enhance sequential recommendations, overcoming the limitations associated with specific knowledge in detection-based strategies. However, its scope is limited to sequential recommender systems and may not generalize well across different CF scenarios.

Conversely, ACF methodologies, particularly those aligned with the Adversarial Personalized Ranking (APR) framework [15], integrate adversarial perturbations at the parameter level (i.e., user and item embeddings) during the model training process [15, 17, 19, 20]. This approach follows a "min-max" optimization paradigm, designed to minimize the error in recommendations under parameter perturbations which aim to maximize the error [13]. Besides, numerous studies have demonstrated that ACF not only enhances the model's robustness but also improves its recommendation performance [15, 20, 27]. Nonetheless, despite its benefits in specific contexts through empirical validation, the intrinsic mechanisms of ACF's effectiveness and its universal applicability remain areas for further theoretical exploration.

---

**Algorithm 1** The Training Procedure of PamaCF-BPR

---
**Input:** Training set $\mathcal{D}$, uniform perturbation magnitude $\rho$, adversarial training weight $\lambda$, pre-training epochs $T_{\text{pre}}$, batch size $\mathbb{B}$
**Output:** Model parameters $\Theta = [\boldsymbol{U}, \boldsymbol{V}]$.
 1: Pre-train $\Theta = [\boldsymbol{U}, \boldsymbol{V}]$ for $T_{\text{pre}}$ epochs using Equation 8.
 2: **while** stopping criteria not met **do**
 3:     Draw batch of $\mathbb{B}$ pairs $(u, i, j)$ from $\mathcal{D}$.
 4:     **for** each $(u, i, j)$ in the batch **do**
 5:         Calculate $\Delta_u^{\text{PamaCF}}$, $\Delta_i^{\text{PamaCF}}$, and $\Delta_j^{\text{PamaCF}}$ using Equation 10.
 6:         Compute $\mathcal{L}_{\text{PamaCF}}((u, i, j)|\Theta)$ using Equation 9.
 7:     **end for**
 8:     Update $\Theta$ using the aggregated gradients from $\mathcal{L}_{\text{PamaCF}}(\Theta)$ in the batch.
 9: **end while**
10: **return** $\Theta = [\boldsymbol{U}, \boldsymbol{V}]$

---

# B    Methodology

For clarity, we present the PamaCF version of the widely used Bayesian Personalized Ranking (BPR) [31] loss function, which optimizes recommender models towards personalized ranking. Given the user set $\mathcal{U} = \{u\}$, the item set $\mathcal{V} = \{v\}$, and the training set $\mathcal{D} = \{(u, i, j) \mid u \in \mathcal{U} \wedge i \in \mathcal{V}_u \wedge j \in \mathcal{V} \setminus \mathcal{V}_u\}$, where $\mathcal{V}_u$ denotes the set of items with which user $u$ has interacted. The objective function (to be minimized) of BPR is formally given by:

$$\mathcal{L}_{\text{BPR}}(\Theta = [\boldsymbol{U}, \boldsymbol{V}]) = -\sum_{(u,i,j)\in\mathcal{D}} \ln \sigma \left( \langle \boldsymbol{U}_u, \boldsymbol{V}_i \rangle - \langle \boldsymbol{U}_u, \boldsymbol{V}_j \rangle \right), \tag{8}$$

where $\boldsymbol{U}$ and $\boldsymbol{V}$ represent the learned user and item embeddings, respectively.

The PamaCF version of the BPR loss function is defined as:

$$\mathcal{L}_{\text{PamaCF}}(\Theta) = \mathcal{L}_{\text{BPR}}(\Theta) + \lambda \mathcal{L}_{\text{BPR}}(\Theta + \Delta^{\text{PamaCF}}),$$
$$\text{where} \quad \Delta^{\text{PamaCF}} = \arg\max_{\Delta, \, \|\Delta_{u/i/j}\| \leq \rho \cdot c(\boldsymbol{u}, t), (u,i,j)\in\mathcal{D}} \mathcal{L}_{\text{BPR}}(\Theta + \Delta),$$

where $\lambda$ is the weight of adversarial training, $\rho$ represents the uniform perturbation magnitude for all users, and

$$c(\boldsymbol{u}, t) = \text{sig}\left( \frac{\|\boldsymbol{u}_{(t)}\| - \overline{\|\boldsymbol{u}_{(t)}\|}}{\|\boldsymbol{u}_{(t)}\|} \right).$$

The specific handling of a pair $(u, i, j) \in \mathcal{D}$ is expressed by:

$$\begin{aligned} \mathcal{L}_{\text{PamaCF}}((u, i, j)|\Theta) = &-\ln \sigma \left( \langle \boldsymbol{U}_u, \boldsymbol{V}_i \rangle - \langle \boldsymbol{U}_u, \boldsymbol{V}_j \rangle \right) \\ &-\lambda \ln \sigma \left( \langle \boldsymbol{U}_u + \Delta_u^{\text{PamaCF}}, \boldsymbol{V}_i + \Delta_i^{\text{PamaCF}} \rangle - \langle \boldsymbol{U}_u + \Delta_u^{\text{PamaCF}}, \boldsymbol{V}_j + \Delta_j^{\text{PamaCF}} \rangle \right), \end{aligned} \tag{9}$$

where

$$\begin{aligned} \Delta_u^{\text{PamaCF}} &= \rho c(\boldsymbol{u}, t) \frac{\Gamma_u}{\|\Gamma_u\|}, \quad \text{where} \quad \Gamma_u = \frac{\partial \mathcal{L}_{\text{BPR}}((u, i, j)|\Theta + \Delta^{\text{PamaCF}})}{\partial \Delta_u}, \\ \Delta_i^{\text{PamaCF}} &= \rho c(\boldsymbol{u}, t) \frac{\Gamma_i}{\|\Gamma_i\|}, \quad \text{where} \quad \Gamma_i = \frac{\partial \mathcal{L}_{\text{BPR}}((u, i, j)|\Theta + \Delta^{\text{PamaCF}})}{\partial \Delta_i}, \\ \Delta_j^{\text{PamaCF}} &= \rho c(\boldsymbol{u}, t) \frac{\Gamma_i}{\|\Gamma_j\|}, \quad \text{where} \quad \Gamma_j = \frac{\partial \mathcal{L}_{\text{BPR}}((u, i, j)|\Theta + \Delta^{\text{PamaCF}})}{\partial \Delta_j}. \end{aligned} \tag{10}$$

The procedure of training with PamaCF is illustrated in Algorithm 1.

# C    Experiments

## C.1    Supplements to Experimental Settings

**Datasets**. We employ three common benchmarks: the *Gowalla* check-in dataset [32], the *Yelp2018* business dataset, and the *MIND* news recommendation dataset [33]. The Gowalla and Yelp2018

Table 3: Dataset statistics

| DATASET | #Users | #Items | #Ratings | Avg.Inter. | Sparsity |
|---------|--------|--------|----------|-----------|----------|
| Gowalla | 29,858 | 40,981 | 1,027,370 | 34.4 | 99.92% |
| Yelp2018 | 31,668 | 38,048 | 1,561,406 | 49.3 | 99.88% |
| MIND | 141,920 | 36,214 | 20,693,122 | 145.8 | 99.60% |

datasets include interactions from all users. For the MIND dataset, we sample a subset of users following [13]. Following [3, 37], users and items with fewer than 10 interactions are excluded from our analysis. We allocate 80% of each user's historical interactions to the training set and the remainder for testing. Additionally, 10% of the interactions from the training set are randomly selected to form a validation set for hyperparameter tuning. Detailed statistics of the datasets are summarized in Table 3.

**Attack Methods**. We explore both heuristic (Random Attack [30], Bandwagon Attack [34]) and optimization-based (Rev Attack [7], DP Attack [6]) attack methods within a black-box context, where the attacker does not have access to the internal architecture or parameters of the target model.

- **Random Attack** (Heuristic Method) [30]: This method entails fake users including interactions with both the targeted items and a set of randomly chosen items.

- **Bandwagon Attack** (Heuristic Method) [34]: Fake users' interactions encompass the targeted items and those selected for their high popularity.

- **DP Attack** (Optimization-based Method) [6]: This approach is specifically designed to compromise deep learning-based recommender systems.

- **Rev Attack** (Optimization-based Method) [7]: The attack is conceptualized as a bi-level optimization problem, addressed through gradient-based methods.

**Defense Baselines**. We incorporate a variety of defense methods, including detection-based approaches (GraphRfi [12] and LLM4Dec [13]), adversarial collaborative filtering methods (APR [15] and SharpCF [20]), and a denoise-based strategy (StDenoise [35, 19]). In our study, we employ three common backbone recommendation models, MF [2], LightGCN [3], and NeurMF [36].

- **GraphRfi** [12]: Employs a combination of Graph Convolutional Networks and Neural Random Forests for identifying fake users.

- **LLM4Dec** [13]: Utilizes an LLM-based framework for fake users detection.

- **APR** [15]: Generates parameter perturbations and integrates these perturbations into training.

- **SharpCF** [20]: Adopts a sharpness-aware minimization approach to refine the adversarial training process proposed by APR.

- **StDenoise** [35, 19]: Applies a structural denoising technique that leverages the similarity between $U_u$ and $V_i$ for each $(u, i)$ pair, aiding in the removal of noise, as described in [35, 19].

Note that: With the need of item-side information, LLM4Dec is exclusively evaluated on the MIND dataset. Moreover, we observe that SharpCF, initially proposed for the MF model, exhibits unstable training performance when applied to the LightGCN model or the MIND dataset. Consequently, we present SharpCF results solely for the MF model on the Gowalla and Yelp2018 datasets.

**Implementation Details**. In our study, we employ three common backbone recommendation models, Matrix Factorization (MF) [2], LightGCN [3], and NeurMF [36]. To quantify the success ratio of attacks, we select $k = 50$ as the evaluation metric following [6, 7, 18], while for assessing recommendation performance, we utilize $k = 10, 20$ following [3, 37]. For each attack setting, we conduct experiments five times, taking the average value as the result and the standard deviation as the error bar. The configuration of both the defense methods and the recommendation models involves selecting a learning rate from $\{0.1, 0.01, \ldots, 1 \times 10^{-5}\}$, and a weight decay from $\{0, 0.1, \ldots, 1 \times 10^{-5}\}$. The implementation of GraphRfi follows its paper. For the detection-based methods, we employ the Random Attack to generate supervised attack data. The magnitude parameter of adversarial perturbations in both APR and PamaCF is determined from a range of $\{0.1, 0.2, \ldots, 1.0\}$. In terms of attack methods, we set the attack budget to $1\%$ and target five specific items. The hyperparameters align with those detailed in their original publications.

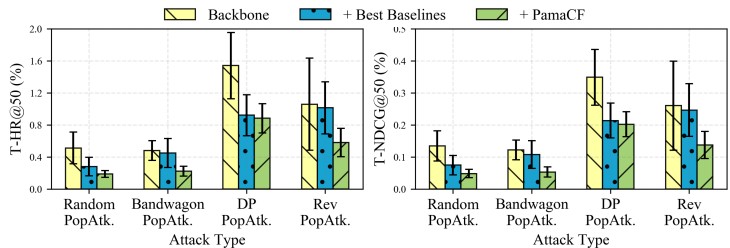

Figure 4: Robustness against popular items promotion.

Table 4: Robustness against target items promotion on Gowalla

| Dataset | Model | Random Attack(%) | | Bandwagon Attack(%) | | DP Attack(%) | | Rev Attack(%) | |
|---|---|---|---|---|---|---|---|---|---|
| | | T-HR@50[1] | T-NDCG@50 | T-HR@50 | T-NDCG@50 | T-HR@50 | T-NDCG@50 | T-HR@50 | T-NDCG@50 |
| Gowalla | **LightGCN** | $0.234 \pm 0.116$ | $0.056 \pm 0.031$ | $0.639 \pm 0.090$ | $0.153 \pm 0.024$ | $0.231 \pm 0.048$ | $0.048 \pm 0.010$ | $0.718 \pm 0.134$ | $0.149 \pm 0.026$ |
| | **+StDenoise** | $0.118 \pm 0.068$ | $0.029 \pm 0.019$ | $0.334 \pm 0.092$ | $\underline{0.079 \pm 0.020}$ | $0.585 \pm 0.092$ | $0.120 \pm 0.019$ | $1.304 \pm 0.184$ | $0.259 \pm 0.037$ |
| | **+GraphRfi** | $0.099 \pm 0.023$ | $0.023 \pm 0.006$ | $0.710 \pm 0.250$ | $0.161 \pm 0.052$ | $0.228 \pm 0.048$ | $0.046 \pm 0.010$ | $\underline{0.564 \pm 0.067}$ | $\underline{0.115 \pm 0.013}$ |
| | **+APR** | $\underline{0.089 \pm 0.053}$ | $\underline{0.021 \pm 0.015}$ | $\underline{0.332 \pm 0.050}$ | $0.079 \pm 0.012$ | $\underline{0.190 \pm 0.037}$ | $\underline{0.039 \pm 0.008}$ | $0.655 \pm 0.141$ | $0.132 \pm 0.027$ |
| | **+PamaCF** | $\mathbf{0.053 \pm 0.041}$ | $\mathbf{0.013 \pm 0.011}$ | $\mathbf{0.194 \pm 0.037}$ | $\mathbf{0.046 \pm 0.009}$ | $\mathbf{0.116 \pm 0.030}$ | $\mathbf{0.023 \pm 0.006}$ | $\mathbf{0.336 \pm 0.061}$ | $\mathbf{0.070 \pm 0.012}$ |
| | Gain[2] | +40.48% ↑ | +40.46% ↑ | +41.64% ↑ | +41.62% ↑ | +38.92% ↑ | +40.13% ↑ | +40.40% ↑ | +39.15% ↑ |
| | **NeurMF** | $0.404 \pm 0.196$ | $0.089 \pm 0.043$ | $0.887 \pm 0.260$ | $0.189 \pm 0.054$ | $0.047 \pm 0.017$ | $0.010 \pm 0.004$ | $0.210 \pm 0.077$ | $0.044 \pm 0.017$ |
| | **+StDenoise** | $0.468 \pm 0.296$ | $0.103 \pm 0.064$ | $0.898 \pm 0.356$ | $0.192 \pm 0.077$ | $0.060 \pm 0.024$ | $0.013 \pm 0.006$ | $0.194 \pm 0.044$ | $0.041 \pm 0.010$ |
| | **+GraphRfi** | $0.241 \pm 0.049$ | $0.052 \pm 0.010$ | $0.485 \pm 0.081$ | $\underline{0.103 \pm 0.017}$ | $0.041 \pm 0.013$ | $0.009 \pm 0.003$ | $0.248 \pm 0.061$ | $0.053 \pm 0.013$ |
| | **+APR** | $\underline{0.094 \pm 0.028}$ | $\underline{0.021 \pm 0.006}$ | $\underline{0.477 \pm 0.217}$ | $0.106 \pm 0.048$ | $\underline{0.046 \pm 0.022}$ | $\underline{0.010 \pm 0.005}$ | $0.426 \pm 0.064$ | $0.092 \pm 0.039$ |
| | **+PamaCF** | $\mathbf{0.074 \pm 0.022}$ | $\mathbf{0.017 \pm 0.006}$ | $\mathbf{0.168 \pm 0.096}$ | $\mathbf{0.038 \pm 0.021}$ | $\mathbf{0.032 \pm 0.021}$ | $\mathbf{0.007 \pm 0.005}$ | $\mathbf{0.186 \pm 0.032}$ | $\mathbf{0.041 \pm 0.011}$ |
| | Gain[1] | +20.98% ↑ | +17.65% ↑ | +64.81% ↑ | +62.57% ↑ | +22.99% ↑ | +19.87% ↑ | +3.99% ↑ | +0.11% ↑ |

[1] Target Item Hit Ratio (Equation 7); T-HR@50 and T-NDCG@50 of all target items on clean datasets are 0.000.
[2] The relative percentage increase of PamaCF's metrics to the best value of other baselines' metrics, i.e., $(\min(\text{T-HR}_{\text{Beslines}}) - \text{T-HR}_{\text{VAT}}) / \min(\text{T-HR}_{\text{Beslines}})$. Notably, only **three decimal places** are presented due to space limitations, though the actual ranking and calculations utilize the **full precision** of the data.
[3] The Rev attack method could not be executed on the dataset due to memory constraints, resulting in an out-of-memory error.

**Compute Resources**. The experiments are conducted using two primary GPU: the RTX 4090 with 24GB of VRAM and the A800 with 80GB of VRAM. For most baseline defense methods applied across all datasets, a single RTX 4090 GPU suffices, requiring several hours per experiment. However, the LLM4Dec method [13] demands an A800 GPU due to its higher resource requirements for processing Large Language Models. In terms of attack generation, heuristic poisoning attacks such as the Random Attack [30] and Bandwagon Attack [34] are generated within seconds and do not require specific GPU resources. Conversely, the time required to generate optimization-based poisoning attacks, such as the DP Attack [6] and Rev Attack [7], depends on the dataset. For the Gowalla and Yelp datasets, these attacks take hours to execute on an A800 GPU. The DP Attack on the MIND dataset extends to several days, also utilizing an A800 GPU. However, the Rev Attack cannot be completed on a single A800 GPU due to its even greater computational demands.

## C.2 Supplements to Performance Comparison

We assess PamaCF's defense capabilities against attacks targeting popular items on Gowalla. According to Figure 4, PamaCF exhibits strong defensibility, outperforming the best baseline even when attacks specifically promote popular items.

We also evaluate PamaCF's defense capabilities and recommendation performance when applied to LightGCN [3] and NeurMF [36], as shown in Table 4 and Table 5, which produces consistent results with MF [2].

Additionally, we report PamaCF's recommendation performance on MF for the Gowalla dataset, specifically for $k = 10$, using Recall@10 and NDCG@10. PamaCF demonstrates significantly greater improvement at $k = 10$, achieving a 29.59% increase in Recall and a 56.41% increase in NDCG, relative to the baseline model, as shown in Table 6.

Table 5: Recommendation Performance on Gowalla

| Model | Clean (%) | | Random Attack (%) | | Bandwagon Attack (%) | | DP Attack (%) | | Rev Attack (%) | |
|---|---|---|---|---|---|---|---|---|---|---|
| (Dataset) | Recall@20 | NDCG@20 | Recall@20 | NDCG@20 | Recall@20 | NDCG@20 | Recall@20 | NDCG@20 | Recall@20 | NDCG@20 |
| LightGCN | $12.54 \pm 0.03$ | $8.27 \pm 0.02$ | $12.46 \pm 0.03$ | $8.26 \pm 0.03$ | $12.50 \pm 0.05$ | $8.28 \pm 0.02$ | $12.83 \pm 0.02$ | $10.10 \pm 0.02$ | $12.99 \pm 0.04$ | $10.25 \pm 0.01$ |
| +StDenoise | $12.52 \pm 0.03$ | $9.92 \pm 0.02$ | $12.40 \pm 0.04$ | $9.81 \pm 0.04$ | $12.38 \pm 0.04$ | $9.77 \pm 0.03$ | $12.46 \pm 0.03$ | $9.84 \pm 0.02$ | $12.56 \pm 0.04$ | $9.93 \pm 0.02$ |
| +GraphRfi | $12.79 \pm 0.04$ | $10.04 \pm 0.00$ | $12.65 \pm 0.04$ | $9.91 \pm 0.02$ | $12.65 \pm 0.04$ | $9.91 \pm 0.02$ | $12.71 \pm 0.04$ | $9.95 \pm 0.01$ | $12.86 \pm 0.03$ | $10.08 \pm 0.02$ |
| +APR | $12.71 \pm 0.03$ | $9.50 \pm 0.03$ | $12.84 \pm 0.02$ | $9.82 \pm 0.03$ | $12.18 \pm 0.01$ | $9.31 \pm 0.03$ | $12.89 \pm 0.03$ | $9.87 \pm 0.03$ | $12.78 \pm 0.05$ | $9.53 \pm 0.03$ |
| +PamaCF | $\mathbf{13.18 \pm 0.02}$ | $\mathbf{10.28 \pm 0.02}$ | $\mathbf{13.02 \pm 0.03}$ | $\mathbf{10.15 \pm 0.02}$ | $\mathbf{13.00 \pm 0.02}$ | $\mathbf{10.12 \pm 0.02}$ | $\mathbf{13.09 \pm 0.02}$ | $\mathbf{10.20 \pm 0.02}$ | $\mathbf{13.24 \pm 0.04}$ | $\mathbf{10.34 \pm 0.02}$ |
| Gain | +3.09% ↑ | +2.38% ↑ | +1.45% ↑ | +2.42% ↑ | +2.78% ↑ | +2.10% ↑ | +1.57% ↑ | +0.99% ↑ | +1.93% ↑ | +0.81% ↑ |
| Gain w.r.t. MF | +5.15% ↑ | +24.23% ↑ | +4.56% ↑ | +22.91% ↑ | +4.06% ↑ | +22.29% ↑ | +1.99% ↑ | +0.99% ↑ | +1.93% ↑ | +0.81% ↑ |
| NeurMF | $9.93 \pm 0.28$ | $6.74 \pm 0.30$ | $9.65 \pm 0.16$ | $6.59 \pm 0.16$ | $9.76 \pm 0.19$ | $6.77 \pm 0.27$ | $9.68 \pm 0.57$ | $6.52 \pm 0.58$ | $9.58 \pm 0.31$ | $6.50 \pm 0.36$ |
| +StDenoise | $10.21 \pm 0.31$ | $6.92 \pm 0.33$ | $9.87 \pm 0.23$ | $6.71 \pm 0.24$ | $10.12 \pm 0.22$ | $6.98 \pm 0.21$ | $9.82 \pm 0.53$ | $6.53 \pm 0.55$ | $9.75 \pm 0.50$ | $6.56 \pm 0.56$ |
| +GraphRfi | $9.82 \pm 0.32$ | $6.68 \pm 0.41$ | $9.84 \pm 0.35$ | $6.78 \pm 0.44$ | $9.65 \pm 0.31$ | $6.50 \pm 0.36$ | $9.92 \pm 0.18$ | $6.78 \pm 0.22$ | $9.77 \pm 0.39$ | $6.63 \pm 0.48$ |
| +APR | $10.02 \pm 0.24$ | $6.92 \pm 0.24$ | $9.99 \pm 0.22$ | $6.90 \pm 0.23$ | $9.90 \pm 0.29$ | $6.91 \pm 0.35$ | $9.86 \pm 0.34$ | $6.74 \pm 0.42$ | $9.74 \pm 0.35$ | $6.67 \pm 0.41$ |
| +PamaCF | $\mathbf{10.26 \pm 0.17}$ | $\mathbf{7.06 \pm 0.18}$ | $\mathbf{10.27 \pm 0.21}$ | $\mathbf{7.13 \pm 0.27}$ | $\mathbf{10.28 \pm 0.27}$ | $\mathbf{7.23 \pm 0.35}$ | $\mathbf{10.14 \pm 0.40}$ | $\mathbf{6.87 \pm 0.44}$ | $\mathbf{10.02 \pm 0.36}$ | $\mathbf{6.85 \pm 0.38}$ |
| Gain | +0.53% ↑ | +1.97% ↑ | +2.85% ↑ | +3.32% ↑ | +1.56% ↑ | +3.55% ↑ | +2.18% ↑ | +1.24% ↑ | +2.54% ↑ | +2.67% ↑ |
| Gain w.r.t. MF | +3.41% ↑ | +4.74% ↑ | +6.46% ↑ | +8.26% ↑ | +5.32% ↑ | +6.68% ↑ | +4.71% ↑ | +5.32% ↑ | +4.65% ↑ | +5.33% ↑ |

[1] The relative percentage increase of PamaCF's metrics to the best value of other baselines' metrics. Notably, only **three decimal places** are presented due to space limitations, though the actual ranking and calculations utilize the **full precision** of the data.

Table 6: Recommendation Performance@10 on Gowalla

| Model | Clean (%) | | Random Attack (%) | | Bandwagon Attack (%) | | DP Attack (%) | | Rev Attack (%) | |
|---|---|---|---|---|---|---|---|---|---|---|
| (Dataset) | Recall@10 | NDCG@10 | Recall@10 | NDCG@10 | Recall@10 | NDCG@10 | Recall@10 | NDCG@10 | Recall@10 | NDCG@10 |
| MF | $7.49 \pm 0.08$ | $5.85 \pm 0.03$ | $7.47 \pm 0.03$ | $5.89 \pm 0.05$ | $7.41 \pm 0.06$ | $5.81 \pm 0.04$ | $7.24 \pm 0.11$ | $7.23 \pm 0.08$ | $7.24 \pm 0.04$ | $7.26 \pm 0.02$ |
| +StDenoise | $7.03 \pm 0.08$ | $7.17 \pm 0.10$ | $6.99 \pm 0.08$ | $7.16 \pm 0.06$ | $6.95 \pm 0.09$ | $7.13 \pm 0.03$ | $7.09 \pm 0.11$ | $7.22 \pm 0.09$ | $7.14 \pm 0.04$ | $7.29 \pm 0.03$ |
| +GraphRfi | $6.98 \pm 0.03$ | $7.03 \pm 0.06$ | $6.92 \pm 0.06$ | $6.96 \pm 0.06$ | $6.86 \pm 0.04$ | $6.91 \pm 0.06$ | $6.97 \pm 0.07$ | $7.02 \pm 0.09$ | $7.06 \pm 0.09$ | $7.08 \pm 0.06$ |
| +APR | $9.29 \pm 0.06$ | $9.69 \pm 0.06$ | $9.21 \pm 0.04$ | $9.58 \pm 0.01$ | $9.20 \pm 0.05$ | $9.56 \pm 0.03$ | $9.24 \pm 0.04$ | $9.64 \pm 0.05$ | $9.41 \pm 0.07$ | $9.78 \pm 0.09$ |
| +SharpCF | $8.81 \pm 0.09$ | $8.83 \pm 0.10$ | $8.80 \pm 0.09$ | $8.84 \pm 0.07$ | $8.71 \pm 0.08$ | $8.72 \pm 0.06$ | $8.93 \pm 0.11$ | $8.91 \pm 0.09$ | $8.92 \pm 0.05$ | $8.93 \pm 0.03$ |
| +PamaCF | $\mathbf{9.56 \pm 0.02}$ | $\mathbf{9.94 \pm 0.05}$ | $\mathbf{9.46 \pm 0.03}$ | $\mathbf{9.84 \pm 0.01}$ | $\mathbf{9.49 \pm 0.02}$ | $\mathbf{9.84 \pm 0.01}$ | $\mathbf{9.54 \pm 0.05}$ | $\mathbf{9.93 \pm 0.04}$ | $\mathbf{9.68 \pm 0.05}$ | $\mathbf{10.05 \pm 0.10}$ |
| Gain | +2.91% ↑ | +2.56% ↑ | +2.69% ↑ | +2.74% ↑ | +3.13% ↑ | +2.86% ↑ | +3.25% ↑ | +2.97% ↑ | +2.87% ↑ | +2.78% ↑ |
| Gain w.r.t. MF | +27.62% ↑ | +70.00% ↑ | +26.65% ↑ | +66.97% ↑ | +28.10% ↑ | +69.27% ↑ | +31.78% ↑ | +37.34% ↑ | +33.80% ↑ | +38.48% ↑ |

[1] The relative percentage increase of PamaCF's metrics to the best value of other baselines' metrics. Notably, only **three decimal places** are presented due to space limitations, though the actual ranking and calculations utilize the **full precision** of the data.

# D  Proofs

## D.1  Proofs for Section 3.1

### D.1.1  Proof of Theorem 1

To investigate the recommendation error of each user during the training period, we first analyze how the item embeddings change. We discover a transformation function that accurately measures the change in the item embedding from its initial state to its state after a certain number of training epochs. This insight is formally expressed in the following proposition.

**Proposition 1.** *Consider a Gaussian Recommender System $f_{(t)}$, undergoing training for $t$ epochs using the standard loss function specified in Equation 2. Given a learning rate $\eta$, there exists a function $M(t,\eta) : \mathbb{N}^+ \times \mathbb{R}^+ \to \mathbb{R}^+$ that quantify the transformation of the item embedding due to training. Specifically, we have:*

$$\boldsymbol{v}_{(t)} = M(t,\eta)\boldsymbol{v}_{(0)},$$

*where $\boldsymbol{v}_{(0)}$ denotes the initial item embedding and $\boldsymbol{v}_{(t)}$ represents the item embedding after $t$ epochs of training.*

***Proof of Proposition 1.*** Consider the training process of the Gaussian Recommender System $f_{(t)}$ over $t$ epochs, with each update through the standard loss outlined in Equation 2. The update mechanism for user and item embeddings at the $t^{\text{th}}$ epoch can be described as follows:

$$\begin{aligned}
\boldsymbol{u}_{(t)} &= \boldsymbol{u}_{(t-1)} + \eta \cdot r\boldsymbol{v}_{(t-1)}, \\
\boldsymbol{v}_{(t)} &= \boldsymbol{v}_{(t-1)} + \eta \cdot \sum_{(\boldsymbol{u},r)\in\mathcal{I}} r\boldsymbol{u}_{(t-1)}.
\end{aligned} \tag{11}$$

Considering the sum $\sum_{(\boldsymbol{u},r)} r\boldsymbol{u}_{(t-1)}$, we have:

$$\sum_{(\boldsymbol{u},r)} r\boldsymbol{u}_{(t-1)} = \sum_{(\boldsymbol{u},r)} \left(r\boldsymbol{u}_{(t-2)} + \eta r^2 \boldsymbol{v}_{(t-2)}\right)$$

$$= \sum_{(\boldsymbol{u},r)} \left(r\boldsymbol{u}_{(t-3)} + \eta\left(\boldsymbol{v}_{(t-2)} + \boldsymbol{v}_{(t-3)}\right)\right)$$

$$\cdots$$

$$= \sum_{(\boldsymbol{u},r)} \left(r\boldsymbol{u}_{(0)} + \eta \sum_{j=0}^{t-2} \boldsymbol{v}_{(j)}\right)$$

$$= n\boldsymbol{v}_{(0)} + n\eta \sum_{j=0}^{t-2} \boldsymbol{v}_{(j)},$$

where $n$ is the number of users. This leads to the recursive update for $\boldsymbol{v}_{(t)}$:

$$\boldsymbol{v}_{(t)} = \boldsymbol{v}_{(t-1)} + n\eta\boldsymbol{v}_{(0)} + n\eta^2 \sum_{j=0}^{t-2} \boldsymbol{v}_{(j)}$$

$$= (1 + n\eta^2)\boldsymbol{v}_{(t-2)} + 2n\eta\boldsymbol{v}_{(0)} + 2n\eta^2 \sum_{j=0}^{t-3} \boldsymbol{v}_{(j)}$$

$$= (1 + n\eta^2 + 2n\eta^2)\boldsymbol{v}_{(t-3)} + \left(n\eta \cdot (1 + n\eta^2) + 2n\eta\right)\boldsymbol{v}_{(0)} + \left(n\eta^2 \cdot (1 + n\eta^2) + 2n\eta^2\right) \sum_{j=0}^{t-4} \boldsymbol{v}_{(j)}.$$

To simplify, we introduce $a(k), b(k)$ and $c(k)$ to represent the cumulative scaling factors as:

$$a(k) = \begin{cases} 1 & k = 1 \\ a(k-1) + c(k-1) & k \geq 2 \end{cases},$$

$$b(k) = \begin{cases} n\eta & k = 1 \\ n\eta \cdot a(k-1) + b(k-1) & k \geq 2 \end{cases}, \tag{12}$$

$$c(k) = \begin{cases} n\eta^2 & k = 1 \\ n\eta^2 \cdot a(k-1) + c(k-1) & k \geq 2 \end{cases},$$

yielding a general form for $\boldsymbol{v}_{(t)}$:

$$\boldsymbol{v}_{(t)} = a(k)\boldsymbol{v}_{(t-k)} + b(k)\boldsymbol{v}_{(0)} + c(k) \sum_{j=0}^{t-k-1} \boldsymbol{v}_{(j)}$$

$$= a(t-1)\boldsymbol{v}_{(1)} + b(t-1)\boldsymbol{v}_{(0)} + c(t-1)\boldsymbol{v}_{(0)}.$$

Based on

$$\boldsymbol{v}_{(1)} = \boldsymbol{v}_{(0)} + \eta \sum_{(\boldsymbol{u},r)\in\mathcal{I}} r\boldsymbol{u}_{(0)} = (1 + n\eta)\boldsymbol{v}_{(0)},$$

let $M(t,\eta) : \mathbb{N}^+ \times \mathbb{R}^+ \to \mathbb{R}^+$ be the transformation function related to training epochs $t$ and learning rate $\eta$, which is defined as:

$$M(t,\eta) = \begin{cases} 1 + n\eta & t = 1 \\ (1 + n\eta)a(t-1) + b(t-1) + c(t-1) & t > 1 \end{cases}.$$

Then we can get:

$$\boldsymbol{v}_{(t)} = M(t,\eta)\boldsymbol{v}_{(0)}.$$

This proves that the item embedding $\boldsymbol{v}_{(t)}$ after $t$ epochs of training is a scaled version of the initial embedding $\boldsymbol{v}_{(0)}$, with the scaling factor $M(t,\eta)$ being a function of the number of epochs $t$ and the learning rate $\eta$. $\qquad\square$

**Fact 1.** *Let $\varepsilon \in \mathbb{R}^d$ be drawn from $\mathcal{N}(\mathbf{0}, \sigma^2 I)$, with $\sigma > 0$. Let $\mathbf{w} \in \mathbb{R}^d$ represent any unit vector. Then, $\langle \varepsilon, \mathbf{w} \rangle$ follows a normal distribution $\mathcal{N}(0, \sigma^2)$.*

**Proof of Fact 1.** Consider $\varepsilon = [\varepsilon_1, \ldots, \varepsilon_d]$, with each $\varepsilon_i$ following $\mathcal{N}(0, \sigma^2)$. Let $\mathbf{w} = [w_1, \ldots, w_d]$, satisfying $w_1^2 + \cdots + w_d^2 = 1$. Then $\langle \varepsilon, \mathbf{w} \rangle$ is distributed as $\mathcal{N}(\mathbb{E}[\langle \varepsilon, \mathbf{w} \rangle], \mathbb{D}[\langle \varepsilon, \mathbf{w} \rangle])$, where:

$$\mathbb{E}[\langle \varepsilon, \mathbf{w} \rangle] = \mathbb{E}[\varepsilon_1 w_1 + \cdots + \varepsilon_d w_d] = 0,$$
$$\mathbb{D}[\langle \varepsilon, \mathbf{w} \rangle] = \mathbb{D}[\varepsilon_1 w_1 + \cdots + \varepsilon_d w_d]$$
$$= \mathbb{D}[\varepsilon_1 w_1] + \cdots + \mathbb{D}[\varepsilon_d w_d]$$
$$= (w_1^2 + \cdots + w_d^2)\sigma^2$$
$$= \sigma^2.$$

Hence, Fact 1 is proved. $\square$

**Proof of Theorem 1.** By invoking Proposition 1, we establish the basis for evaluating the impact of training on the recommendation error at the $(t + 1)^{\text{th}}$ epoch. Proposition 1 specifies the scaling relationship between the initial and trained item embeddings, leading to the following update expressions for user and item embeddings. For the standard loss though Equation 2, we have:

$$\mathbf{u}_{(t+1)} = \mathbf{u}_{(t)} + \eta r M(t, \eta) \mathbf{v}_{(0)},$$
$$\mathbf{v}_{(t+1)} = M(t + 1, \eta) \mathbf{v}_{(0)}.$$

Considering the above update rules, the recommendation error at the $(t + 1)^{\text{th}}$ epoch is given by:

$$\mathbb{P}_{\mathbf{v}_{(0)} \sim \mathcal{N}(\bar{\mathbf{u}}, \frac{\sigma^2}{n-1} I)} \left[ f_{(t+1)}(\mathbf{u}, \mathbf{v}) \neq r \mid (\mathbf{u}, r) \right]$$

$$= \mathbb{P}_{\mathbf{v}_{(0)} \sim \mathcal{N}(\bar{\mathbf{u}}, \frac{\sigma^2}{n-1} I)} \left[ r \cdot \langle \mathbf{u}_{(t+1)}, \mathbf{v}_{(t+1)} \rangle \leq 0 \right]$$

$$= \mathbb{P}_{\mathbf{v}_{(0)} \sim \mathcal{N}(\bar{\mathbf{u}}, \frac{\sigma^2}{n-1} I)} \left[ r \cdot \langle \left( \mathbf{u}_{(t)} + \eta r M(t, \eta) \mathbf{v}_{(0)} \right), M(t + 1, \eta) \mathbf{v}_{(0)} \rangle \leq 0 \right] \quad (13)$$

$$= \mathbb{P}_{\mathbf{v}_{(0)} \sim \mathcal{N}(\bar{\mathbf{u}}, \frac{\sigma^2}{n-1} I)} \left[ r \cdot \langle \left( \mathbf{u}_{(t)} + \eta r M(t, \eta) \mathbf{v}_{(0)} \right), \mathbf{v}_{(0)} \rangle \leq 0 \right]$$

$$= \mathbb{P}_{\mathbf{v}_{(0)} \sim \mathcal{N}(\bar{\mathbf{u}}, \frac{\sigma^2}{n-1} I)} \left[ r \cdot \langle \mathbf{u}_{(t)}, \mathbf{v}_{(0)} \rangle \leq -\eta M(t, \eta) \|\mathbf{v}_{(0)}\|^2 \right].$$

For adversarial loss as detailed by Equation 1, we have:

$$\Delta_{\text{adv}} = \arg \max_{\Delta, \|\Delta\| \leq \epsilon} \mathcal{L}(\Theta + \Delta).$$

According to the first-order Taylor expansion, we have:

$$\Delta_{\text{adv}} \approx \arg \max_{\Delta, \|\Delta\| \leq \epsilon} \mathcal{L}(\Theta) + \langle \Delta, \nabla_\Theta \mathcal{L}(\Theta) \rangle$$
$$= \arg \max_{\Delta, \|\Delta\| \leq \epsilon} \langle \Delta, \nabla_\Theta \mathcal{L}(\Theta) \rangle$$
$$= \epsilon \frac{\nabla_\Theta \mathcal{L}(\Theta)}{\|\nabla_\Theta \mathcal{L}(\Theta)\|},$$

leading to specific perturbations $\Delta_{\mathbf{u}}$ and $\Delta_{\mathbf{v}}$ in Equation 3:

$$\Delta_{\mathbf{u}} = \epsilon \frac{\Gamma_{\mathbf{v}}}{\|\Gamma_{\mathbf{u}}\|}, \quad \text{where} \quad \Gamma_{\mathbf{u}} = \frac{\partial \mathcal{L}(\mathbf{u}, \mathbf{v} | \Theta)}{\partial \mathbf{u}} = -r\mathbf{v},$$
$$\Delta_{\mathbf{v}} = \epsilon \frac{\Gamma_{\mathbf{v}}}{\|\Gamma_{\mathbf{v}}\|}, \quad \text{where} \quad \Gamma_{\mathbf{v}} = \frac{\partial \mathcal{L}(\mathbf{v}, \mathbf{u} | \Theta)}{\partial \mathbf{v}} = -r\mathbf{u}.$$

$$(14)$$

Subsequently, the updated embeddings through adversarial loss are expressed as:

$$
\begin{aligned}
\boldsymbol{u}_{(t+1)}^{\mathrm{adv}} &= \boldsymbol{u}_{(t)} + \eta \cdot \left( r\boldsymbol{v}_{(t)} + \lambda r \left( \boldsymbol{v}_{(t)} - \epsilon \frac{r\boldsymbol{u}_{(t)}}{\|\boldsymbol{u}_{(t)}\|} \right) \right) \\
&= \boldsymbol{u}_{(t)} + \eta(1+\lambda)r\boldsymbol{v}_{(t)} - \eta\lambda\epsilon \frac{\boldsymbol{u}_{(t)}}{\|\boldsymbol{u}_{(t)}\|} \\
&= \left( 1 - \frac{\eta\lambda\epsilon}{\|\boldsymbol{u}_{(t)}\|} \right) \boldsymbol{u}_{(t)} + \eta(1+\lambda)r \cdot M(t,\eta)\boldsymbol{v}_{(0)} \\
\boldsymbol{v}_{(t+1)}^{\mathrm{adv}} &= \boldsymbol{v}_{(t)} + \eta \cdot \left( \sum r\boldsymbol{u}_{(t)} + \lambda \sum r \left( \boldsymbol{u}_{(t)} - \epsilon \frac{r\boldsymbol{v}_{(t)}}{\|\boldsymbol{v}_{(t)}\|} \right) \right) \\
&= \boldsymbol{v}_{(t)} + \eta(1+\lambda) \sum r\boldsymbol{u}_{(t)} - \frac{n\eta\lambda\epsilon}{\|\boldsymbol{v}_{(t)}\|}\boldsymbol{v}_{(t)} \\
&= \left( M(t,\eta) + n\eta(1+\lambda) \left( 1 + \eta + \eta \sum_{j=1}^{t-1} M(j,\eta) \right) - \frac{n\eta\lambda\epsilon}{\|\boldsymbol{v}_{(0)}\|} \right) \boldsymbol{v}_{(0)}.
\end{aligned}
\tag{15}
$$

Given $\|\bar{\boldsymbol{u}}\| \gg \sigma$, we can approximate $\|\boldsymbol{v}_{(0)}\|$ with $\mathbb{E}[\|\boldsymbol{v}_{(0)}\|]$. Given $\epsilon < \frac{\min(\|\boldsymbol{u}_{(t)}\|,\|\bar{\boldsymbol{u}}\|)}{\eta\lambda}$, according to the expansion of $M(t,\eta)$ in Proposition 1, it follows that $\left( M(t,\eta) + n\eta(1+\lambda) \left( 1 + \eta + \eta \sum_{j=1}^{t-1} M(j,\eta) \right) - \frac{n\eta\lambda\epsilon}{\|\boldsymbol{v}_{(0)}\|} \right) > 0$. Considering these update rules, the recommendation error at the $(t+1)^{\mathrm{th}}$ epoch is determined by:

$$
\begin{aligned}
\mathbb{P}_{\boldsymbol{v}_{(0)}\sim\mathcal{N}(\bar{\boldsymbol{u}},\frac{\sigma^2}{n-1}I)}^{\mathrm{adv}} & \left[ f_{(t+1)}(\boldsymbol{u},\boldsymbol{v}) \neq r \mid (\boldsymbol{u},r) \right] \\
&= \mathbb{P}_{\boldsymbol{v}_{(0)}\sim\mathcal{N}(\bar{\boldsymbol{u}},\frac{\sigma^2}{n-1}I)} \left[ r \cdot \langle \boldsymbol{u}_{(t+1)}^{\mathrm{adv}}, \boldsymbol{v}_{(t+1)}^{\mathrm{adv}} \rangle \leq 0 \right] \\
&= \mathbb{P}_{\boldsymbol{v}_{(0)}\sim\mathcal{N}(\bar{\boldsymbol{u}},\frac{\sigma^2}{n-1}I)} \left[ r \cdot \left\langle \left( \left( 1 - \frac{\eta\lambda\epsilon}{\|\boldsymbol{u}_{(t)}\|} \right) \boldsymbol{u}_{(t)} + \eta(1+\lambda)r \cdot M(t,\eta)\boldsymbol{v}_{(0)} \right), \boldsymbol{v}_{(0)} \right\rangle \leq 0 \right] \\
&= \mathbb{P}_{\boldsymbol{v}_{(0)}\sim\mathcal{N}(\bar{\boldsymbol{u}},\frac{\sigma^2}{n-1}I)} \left[ r \cdot \left\langle \left( 1 - \frac{\eta\lambda\epsilon}{\|\boldsymbol{u}_{(t)}\|} \right) \boldsymbol{u}_{(t)}, \boldsymbol{v}_{(0)} \right\rangle + \eta(1+\lambda)M(t,\eta)\langle\boldsymbol{v}_{(0)},\boldsymbol{v}_{(0)}\rangle \leq 0 \right] \\
&= \mathbb{P}_{\boldsymbol{v}_{(0)}\sim\mathcal{N}(\bar{\boldsymbol{u}},\frac{\sigma^2}{n-1}I)} \left[ r \cdot \left\langle \left( 1 - \frac{\eta\lambda\epsilon}{\|\boldsymbol{u}_{(t)}\|} \right) \boldsymbol{u}_{(t)}, \boldsymbol{v}_{(0)} \right\rangle \leq -\eta(1+\lambda)M(t,\eta)\|\boldsymbol{v}_{(0)}\|^2 \right].
\end{aligned}
\tag{16}
$$

Let $\gamma_{(t)}^{(\boldsymbol{u})} = \left( 1 - \frac{\eta\lambda\epsilon}{\|\boldsymbol{u}_{(t)}\|} \right)^{-1}$. Given the condition $\epsilon < \frac{\min(\|\boldsymbol{u}_{(t)}\|,\|\bar{\boldsymbol{u}}\|)}{\eta\lambda}$, it follows $\gamma_{(t)}^{(\boldsymbol{u})} > 1$. The final form of the recommendation error at the $(t+1)^{\mathrm{th}}$ epoch under adversarial training is:

$$
\begin{aligned}
\mathbb{P}_{\boldsymbol{v}_{(0)}\sim\mathcal{N}(\bar{\boldsymbol{u}},\frac{\sigma^2}{n-1}I)}^{\mathrm{adv}} & \left[ f_{(t+1)}(\boldsymbol{u},\boldsymbol{v}) \neq r \mid (\boldsymbol{u},r) \right] \\
&= \mathbb{P}_{\boldsymbol{v}_{(0)}\sim\mathcal{N}(\bar{\boldsymbol{u}},\frac{\sigma^2}{n-1}I)} \left[ r \cdot \langle \boldsymbol{u}_{(t)}, \boldsymbol{v}_{(0)} \rangle \leq -\eta(1+\lambda)\gamma_{(t)}^{(\boldsymbol{u})} M(t,\eta)\|\boldsymbol{v}_{(0)}\|^2 \right]
\end{aligned}
$$

This leads us to:

$$
\begin{aligned}
& \mathbb{P}_{\boldsymbol{v}_{(0)}\sim\mathcal{N}(\bar{\boldsymbol{u}},\frac{\sigma^2}{n-1}I)} \left[ f_{(t+1)}(\boldsymbol{u},\boldsymbol{v}) \neq r \mid (\boldsymbol{u},r) \right] - \mathbb{P}_{\boldsymbol{v}_{(0)}\sim\mathcal{N}(\bar{\boldsymbol{u}},\frac{\sigma^2}{n-1}I)}^{\mathrm{adv}} \left[ f_{(t+1)}(\boldsymbol{u},\boldsymbol{v}) \neq r \mid (\boldsymbol{u},r) \right] \\
&= \mathbb{P}_{\boldsymbol{v}_{(0)}\sim\mathcal{N}(\bar{\boldsymbol{u}},\frac{\sigma^2}{n-1}I)} \left[ -\eta(1+\lambda)\gamma_{(t)}^{\boldsymbol{u}} M(t,\eta)\|\boldsymbol{v}_{(0)}\|^2 < r\langle \boldsymbol{u}_{(t)}, \boldsymbol{v}_{(0)} \rangle \leq -\eta M(t,\eta)\|\boldsymbol{v}_{(0)}\|^2 \right] \\
&= \mathbb{P}_{\boldsymbol{v}_{(0)}\sim\mathcal{N}(\bar{\boldsymbol{u}},\frac{\sigma^2}{n-1}I)} \left[ -\eta(1+\lambda)\gamma_{(t)}^{\boldsymbol{u}} M(t,\eta)\frac{\|\boldsymbol{v}_{(0)}\|^2}{\|\boldsymbol{u}_{(t)}\|} < \langle \frac{r\boldsymbol{u}_{(t)}}{\|\boldsymbol{u}_{(t)}\|}, \boldsymbol{v}_{(0)} \rangle \leq -\eta M(t,\eta)\frac{\|\boldsymbol{v}_{(0)}\|^2}{\|\boldsymbol{u}_{(t)}\|} \right].
\end{aligned}
$$

Given that $\|\bar{\boldsymbol{u}}\| \gg \sigma$, we can approximate the $\|\boldsymbol{v}_{(0)}\|^2$ by using $\mathbb{E}_{\boldsymbol{v}_{(0)} \sim \mathcal{N}(\bar{\boldsymbol{u}}, \frac{\sigma^2}{n-1} I)} \left[\|\boldsymbol{v}_{(0)}\|^2\right]$ as an estimate. Therefore, we have:

$$\mathbb{P}_{\boldsymbol{v}_{(0)} \sim \mathcal{N}(\bar{\boldsymbol{u}}, \frac{\sigma^2}{n-1} I)} \left[f_{(t+1)}(\boldsymbol{u}, \boldsymbol{v}) \neq r \mid (\boldsymbol{u}, r)\right] - \mathbb{P}^{\mathrm{adv}}_{\boldsymbol{v}_{(0)} \sim \mathcal{N}(\bar{\boldsymbol{u}}, \frac{\sigma^2}{n-1} I)} \left[f_{(t+1)}(\boldsymbol{u}, \boldsymbol{v}) \neq r \mid (\boldsymbol{u}, r)\right]$$

$$\approx \mathbb{P}_{\boldsymbol{v}_{(0)} \sim \mathcal{N}(\bar{\boldsymbol{u}}, \frac{\sigma^2}{n-1} I)} \left[-\eta(1+\lambda)\gamma^{\boldsymbol{u}}_{(t)} M(t, \eta) \frac{\|\bar{\boldsymbol{u}}\|^2 + \frac{d\sigma^2}{n-1}}{\|\boldsymbol{u}_{(t)}\|} < \langle \frac{r\boldsymbol{u}_{(t)}}{\|\boldsymbol{u}_{(t)}\|}, \boldsymbol{v}_{(0)} \rangle \leq -\eta M(t, \eta) \frac{\|\bar{\boldsymbol{u}}\|^2 + \frac{d\sigma^2}{n-1}}{\|\boldsymbol{u}_{(t)}\|}\right],$$

where $d$ is the dimension of $\boldsymbol{v}_{(0)}$.

Let $\boldsymbol{\varepsilon} \sim \mathcal{N}\left(\boldsymbol{0}, \frac{\sigma^2}{n-1} \boldsymbol{I}\right)$, we have $\boldsymbol{v}_{(0)} = \bar{\boldsymbol{u}} + \boldsymbol{\varepsilon}$. Thus:

$$\mathbb{P}_{\boldsymbol{v}_{(0)} \sim \mathcal{N}(\bar{\boldsymbol{u}}, \frac{\sigma^2}{n-1} I)} \left[f_{(t+1)}(\boldsymbol{u}, \boldsymbol{v}) \neq r \mid (\boldsymbol{u}, r)\right] - \mathbb{P}^{\mathrm{adv}}_{\boldsymbol{v}_{(0)} \sim \mathcal{N}(\bar{\boldsymbol{u}}, \frac{\sigma^2}{n-1} I)} \left[f_{(t+1)}(\boldsymbol{u}, \boldsymbol{v}) \neq r \mid (\boldsymbol{u}, r)\right]$$

$$= \mathbb{P}_{\boldsymbol{\varepsilon} \sim \mathcal{N}\left(\boldsymbol{0}, \frac{\sigma^2}{n-1} \boldsymbol{I}\right)} \left[-\eta(1+\lambda)\gamma^{\boldsymbol{u}}_{(t)} M(t, \eta) \frac{\|\bar{\boldsymbol{u}}\|^2 + \frac{d\sigma^2}{n-1}}{\|\boldsymbol{u}_{(t)}\|} < \langle \frac{r\boldsymbol{u}_{(t)}}{\|\boldsymbol{u}_{(t)}\|}, \bar{\boldsymbol{u}} + \boldsymbol{\varepsilon} \rangle \leq -\eta M(t, \eta) \frac{\|\bar{\boldsymbol{u}}\|^2 + \frac{d\sigma^2}{n-1}}{\|\boldsymbol{u}_{(t)}\|}\right]$$

$$= \mathbb{P}_{\boldsymbol{\varepsilon} \sim \mathcal{N}\left(\boldsymbol{0}, \frac{\sigma^2}{n-1} \boldsymbol{I}\right)} \left[-\eta(1+\lambda)\gamma^{\boldsymbol{u}}_{(t)} M(t, \eta) \frac{\|\bar{\boldsymbol{u}}\|^2 + \frac{d\sigma^2}{n-1}}{\|\boldsymbol{u}_{(t)}\|} - \langle \frac{r\boldsymbol{u}_{(t)}}{\|\boldsymbol{u}_{(t)}\|}, \bar{\boldsymbol{u}} \rangle \right.$$

$$\left. < \langle \frac{r\boldsymbol{u}_{(t)}}{\|\boldsymbol{u}_{(t)}\|}, \boldsymbol{\varepsilon} \rangle \leq -\eta M(t, \eta) \frac{\|\bar{\boldsymbol{u}}\|^2 + \frac{d\sigma^2}{n-1}}{\|\boldsymbol{u}_{(t)}\|} - \langle \frac{r\boldsymbol{u}_{(t)}}{\|\boldsymbol{u}_{(t)}\|}, \bar{\boldsymbol{u}} \rangle\right].$$

$$(17)$$

According to Fact 1, $\langle \frac{r\boldsymbol{u}_{(t)}}{\|\boldsymbol{u}_{(t)}\|}, \boldsymbol{\varepsilon} \rangle \sim \mathcal{N}\left(0, \frac{\sigma^2}{n-1}\right)$, then:

$$\mathbb{P}_{\boldsymbol{v}_{(0)} \sim \mathcal{N}(\bar{\boldsymbol{u}}, \frac{\sigma^2}{n-1} I)} \left[f_{(t+1)}(\boldsymbol{u}, \boldsymbol{v}) \neq r \mid (\boldsymbol{u}, r)\right] - \mathbb{P}^{\mathrm{adv}}_{\boldsymbol{v}_{(0)} \sim \mathcal{N}(\bar{\boldsymbol{u}}, \frac{\sigma^2}{n-1} I)} \left[f_{(t+1)}(\boldsymbol{u}, \boldsymbol{v}) \neq r \mid (\boldsymbol{u}, r)\right]$$

$$= \Phi\left(\frac{\sqrt{n-1}}{\sigma} \left(\eta(1+\lambda)\gamma^{\boldsymbol{u}}_{(t)} M(t, \eta) \frac{\|\bar{\boldsymbol{u}}\|^2 + \frac{d\sigma^2}{n-1}}{\|\boldsymbol{u}_{(t)}\|} + \langle \frac{r\boldsymbol{u}_{(t)}}{\|\boldsymbol{u}_{(t)}\|}, \bar{\boldsymbol{u}} \rangle\right)\right)$$

$$- \Phi\left(\frac{\sqrt{n-1}}{\sigma} \left(\eta M(t, \eta) \frac{\|\bar{\boldsymbol{u}}\|^2 + \frac{d\sigma^2}{n-1}}{\|\boldsymbol{u}_{(t)}\|} + \langle \frac{r\boldsymbol{u}_{(t)}}{\|\boldsymbol{u}_{(t)}\|}, \bar{\boldsymbol{u}} \rangle\right)\right),$$

where $\Phi()$ is the CDF of standard Gaussian distribution. Given $\epsilon < \frac{\min(\|\boldsymbol{u}_{(t)}\|, \|\bar{\boldsymbol{u}}\|)}{\eta\lambda}$, it follows that $(1+\lambda)\gamma^{(\boldsymbol{u})}_{(t)} > 1 + \lambda > 1$. This condition implies a lower recommendation error at the $(t+1)^{\mathrm{th}}$ epoch under adversarial training compared to standard training:

$$\mathbb{P}_{\boldsymbol{v}_{(0)} \sim \mathcal{N}(\bar{\boldsymbol{u}}, \frac{\sigma^2}{n-1} I)} \left[f_{(t+1)}(\boldsymbol{u}, \boldsymbol{v}) \neq r \mid (\boldsymbol{u}, r)\right] - \mathbb{P}^{\mathrm{adv}}_{\boldsymbol{v}_{(0)} \sim \mathcal{N}(\bar{\boldsymbol{u}}, \frac{\sigma^2}{n-1} I)} \left[f_{(t+1)}(\boldsymbol{u}, \boldsymbol{v}) \neq r \mid (\boldsymbol{u}, r)\right] > 0.$$

Therefore, Theorem 1 is proved. $\qquad\square$

### D.1.2 Proof of Theorem 2

***Proof of Theorem 2***. In light of Theorem 1 and the impact of poisoning attacks, our objective is to measure the alteration in the recommendation error within a poisoning attack, i.e., $\alpha$-poisoned recommendation error.

A *poisoning attack* on Gaussian Recommender System injects a *poisoning user set* $\mathcal{I}' = \{(\boldsymbol{u}'_1, r'_1), (\boldsymbol{u}'_2, r'_2), \ldots, (\boldsymbol{u}'_{n'}, r'_{n'})\}$, with each tuple $(\boldsymbol{u}', r') \in \mathbb{R}^d \times \{\pm 1\}$ representing data maliciously crafted by attackers. Considering the initialized poisoned item embedding $\boldsymbol{v}'$:

$$\boldsymbol{v}'_{(0)} = \frac{1}{n+n'} \left(\sum_{(\boldsymbol{u}, r) \in \mathcal{I}} r\boldsymbol{u}_{(0)} + \sum_{(\boldsymbol{u}', r') \in \mathcal{I}'} r'\boldsymbol{u}'_{(0)}\right), \qquad (18)$$

by employing Theorem 1 as a basis (similar to Equations 13 and 16), we derive:

$$\mathbb{P}_{\boldsymbol{v}_{(0)}\sim\mathcal{N}(\bar{\boldsymbol{u}},\frac{\sigma^2}{n-1}I)}\left[f_{(t+1),\alpha}(\boldsymbol{u},\boldsymbol{v}')\neq r\mid(\boldsymbol{u},r)\right]-\mathbb{P}^{\mathrm{adv}}_{\boldsymbol{v}_{(0)}\sim\mathcal{N}(\bar{\boldsymbol{u}},\frac{\sigma^2}{n-1}I)}\left[f_{(t+1),\alpha}(\boldsymbol{u},\boldsymbol{v}')\neq r\mid(\boldsymbol{u},r)\right]$$

$$=\mathbb{P}_{\boldsymbol{v}_{(0)}\sim\mathcal{N}(\bar{\boldsymbol{u}},\frac{\sigma^2}{n-1}I)}\left[-\eta(1+\lambda)\gamma^{\boldsymbol{u}}_{(t)}M(t,\eta)\frac{\|\boldsymbol{v}'_{(0)}\|^2}{\|\boldsymbol{u}_{(t)}\|}<\langle\frac{r\boldsymbol{u}_{(t)}}{\|\boldsymbol{u}_{(t)}\|},\boldsymbol{v}'_{(0)}\rangle\leq-\eta M(t,\eta)\frac{\|\boldsymbol{v}'_{(0)}\|^2}{\|\boldsymbol{u}_{(t)}\|}\right]$$

$$=\mathbb{P}_{\boldsymbol{v}_{(0)}\sim\mathcal{N}(\bar{\boldsymbol{u}},\frac{\sigma^2}{n-1}I)}\left[-\eta(1+\lambda)\gamma^{\boldsymbol{u}}_{(t)}M(t,\eta)\frac{\|\boldsymbol{v}'_{(0)}\|^2}{\|\boldsymbol{u}_{(t)}\|}\right.$$

$$\left.<\langle\frac{r\boldsymbol{u}_{(t)}}{\|\boldsymbol{u}_{(t)}\|},\frac{1}{n+n'}\left(n\boldsymbol{v}_{(0)}+\sum_{(\boldsymbol{u}',r')\in\mathcal{I}'}r'\boldsymbol{u}'_{(0)}\right)\rangle\leq-\eta M(t,\eta)\frac{\|\boldsymbol{v}'_{(0)}\|^2}{\|\boldsymbol{u}_{(t)}\|}\right]$$

$$=\mathbb{P}_{\boldsymbol{v}_{(0)}\sim\mathcal{N}(\bar{\boldsymbol{u}},\frac{\sigma^2}{n-1}I)}\left[-\eta\frac{n+n'}{n}(1+\lambda)\gamma^{\boldsymbol{u}}_{(t)}M(t,\eta)\frac{\|\boldsymbol{v}'_{(0)}\|^2}{\|\boldsymbol{u}_{(t)}\|}\right.$$

$$\left.<\langle\frac{r\boldsymbol{u}_{(t)}}{\|\boldsymbol{u}_{(t)}\|},\boldsymbol{v}_{(0)}\rangle+\frac{1}{n}\sum_{(\boldsymbol{u}',r')\in\mathcal{I}'}rr'\langle\frac{\boldsymbol{u}_{(t)}}{\|\boldsymbol{u}_{(t)}\|},\boldsymbol{u}'_{(0)}\rangle\leq-\eta\frac{n+n'}{n}M(t,\eta)\frac{\|\boldsymbol{v}'_{(0)}\|^2}{\|\boldsymbol{u}_{(t)}\|}\right]$$

$$=\mathbb{P}_{\boldsymbol{v}_{(0)}\sim\mathcal{N}(\bar{\boldsymbol{u}},\frac{\sigma^2}{n-1}I)}\left[-\eta(1+\lambda)\frac{n+n'}{n}\gamma^{\boldsymbol{u}}_{(t)}M(t,\eta)\frac{\|\boldsymbol{v}'_{(0)}\|^2}{\|\boldsymbol{u}_{(t)}\|}-\frac{1}{n}\sum_{(\boldsymbol{u}',r')\in\mathcal{I}'}rr'\langle\frac{\boldsymbol{u}_{(t)}}{\|\boldsymbol{u}_{(t)}\|},\boldsymbol{u}'_{(0)}\rangle\right.$$

$$\left.<\langle\frac{r\boldsymbol{u}_{(t)}}{\|\boldsymbol{u}_{(t)}\|},\boldsymbol{v}_{(0)}\rangle\leq-\eta\frac{n+n'}{n}M(t,\eta)\frac{\|\boldsymbol{v}'_{(0)}\|^2}{\|\boldsymbol{u}_{(t)}\|}-\frac{1}{n}\sum_{(\boldsymbol{u}',r')\in\mathcal{I}'}rr'\langle\frac{\boldsymbol{u}_{(t)}}{\|\boldsymbol{u}_{(t)}\|},\boldsymbol{u}'_{(0)}\rangle\right],$$

where $\gamma^{(\boldsymbol{u})}_{(t)}=\left(1-\frac{\eta\lambda\epsilon}{\|\boldsymbol{u}_{(t)}\|}\right)^{-1}$.

Given $\|\bar{\boldsymbol{u}}\|\gg\sigma$, we can use $\mathbb{E}_{\boldsymbol{v}_{(0)}\sim\mathcal{N}(\bar{\boldsymbol{u}},\frac{\sigma^2}{n-1}I)}\left[\|\boldsymbol{v}'_{(0)}\|^2\right]$ to approximate the $\|\boldsymbol{v}'_{(0)}\|^2$ in the above. Similar to the proof of Theorem 1, specifically Equation 17, and under the precondition $\epsilon<\frac{\min(\|\boldsymbol{u}_{(t)}\|,\|\bar{\boldsymbol{u}}\|)}{\eta\lambda}$, we have:

$$\mathbb{P}_{\boldsymbol{v}_{(0)}\sim\mathcal{N}(\bar{\boldsymbol{u}},\frac{\sigma^2}{n-1}I)}\left[f_{(t+1),\alpha}(\boldsymbol{u},\boldsymbol{v}')\neq r\mid(\boldsymbol{u},r)\right]-\mathbb{P}^{\mathrm{adv}}_{\boldsymbol{v}_{(0)}\sim\mathcal{N}(\bar{\boldsymbol{u}},\frac{\sigma^2}{n-1}I)}\left[f_{(t+1),\alpha}(\boldsymbol{u},\boldsymbol{v}')\neq r\mid(\boldsymbol{u},r)\right]>0.$$

Hence, Theorem 2 is proved. $\qquad\square$

### D.2 Proofs for Section 3.2

#### D.2.1 Proof of Theorem 3

Extending Proposition 1 to ACF yields the following proposition, which captures the transformation of item embedding due to adversarial loss as defined in Equation 3.

**Proposition 2.** *Consider a Gaussian Recommender System $f_{(t)}$, per-trained on standard loss over $t$ epochs, then trained by the adversarial loss specified in Equation 3 over $k$ epochs. Given learning rate $\eta$, adversarial training weight $\lambda$, and perturbation magnitude $\epsilon$, when $\epsilon<\frac{\|\bar{\boldsymbol{u}}\|}{\eta\lambda}$, and $\|\bar{\boldsymbol{u}}\|\gg\sigma$, there exists a transformation function $M_{\mathrm{adv}}(t,\eta,\lambda,\epsilon):\mathbb{N}^+\times\mathbb{R}^+\times\mathbb{R}^+\times\mathbb{R}^+\to\mathbb{R}^+$, such that the item embedding at $(t+k)^{\mathrm{th}}$ epoch, $\boldsymbol{v}_{(t+k)}$, is related to the initial embedding $\boldsymbol{v}_{(0)}$ by:*

$$\boldsymbol{v}_{(t+k)}=\frac{M_{\mathrm{adv}}(t+k,\eta,\lambda,\epsilon)}{M_{\mathrm{adv}}(t,\eta,\lambda,\epsilon)}M(t,\eta)\boldsymbol{v}_{(0)},$$

*where $M(t,\eta)$ is the transformation function given by standard loss in Proposition 1.*

The proof of Proposition 2 follows a reasoning analogous to that of Proposition 1. Due to the similarity, the detailed proof is omitted for brevity.

**Proof of Theorem 3.** Given $\epsilon < \frac{\|\bar{u}\|}{\eta\lambda}$, drawing from Proposition 2 and Theorem 1 (specifically Equation 15), the update rules for user and item embeddings in the ACF at the $(t + k + 1)^{\text{th}}$ epoch are presented as:

$$
\begin{aligned}
\boldsymbol{u}_{(t+k+1)} &= \left(1 - \frac{\eta\lambda\epsilon}{\|\boldsymbol{u}_{(t+k)}\|}\right) \boldsymbol{u}_{(t+k)} + \eta(1+\lambda)r \cdot \frac{M_{\text{adv}}(t+k, \eta, \lambda, \epsilon)}{M_{\text{adv}}(t, \eta, \lambda, \epsilon)} M(t, \eta)\boldsymbol{v}_{(0)}, \\
\boldsymbol{v}_{(t+k+1)} &= \frac{M_{\text{adv}}(t+k+1, \eta, \lambda, \epsilon)}{M_{\text{adv}}(t, \eta, \lambda, \epsilon)} M(t, \eta)\boldsymbol{v}_{(0)}.
\end{aligned}
\tag{19}
$$

Considering the above update rules, the recommendation error at the $(t + k + 1)^{\text{th}}$ epoch is given by:

$$
\begin{aligned}
&\mathbb{P}^{\text{adv}}_{\boldsymbol{v}_{(0)} \sim \mathcal{N}(\bar{\boldsymbol{u}}, \frac{\sigma^2}{n-1}I)} \left[f_{(t+k+1)}(\boldsymbol{u}, \boldsymbol{v}) \neq r \mid (\boldsymbol{u}, r)\right] \\
&= \mathbb{P}_{\boldsymbol{v}_{(0)} \sim \mathcal{N}(\bar{\boldsymbol{u}}, \frac{\sigma^2}{n-1}I)} \left[r \cdot \langle \boldsymbol{u}_{(t+k+1)}, \boldsymbol{v}_{(t+k+1)} \rangle \leq 0\right] \\
&= \mathbb{P}_{\boldsymbol{v}_{(0)} \sim \mathcal{N}(\bar{\boldsymbol{u}}, \frac{\sigma^2}{n-1}I)} \left[r \cdot \langle \left(1 - \frac{\eta\lambda\epsilon}{\|\boldsymbol{u}_{(t+k)}\|}\right) \boldsymbol{u}_{(t+k)}, \boldsymbol{v}_{(0)} \rangle \leq -\eta(1+\lambda)C_{t+k}\|\boldsymbol{v}_{(0)}\|^2\right],
\end{aligned}
\tag{20}
$$

where $C_{t+k} = \frac{M_{\text{adv}}(t+k, \eta, \lambda, \epsilon)}{M_{\text{adv}}(t, \eta, \lambda, \epsilon)} M(t, \eta)$. Let $\gamma^{(\boldsymbol{u})}_{(t+k)} = \left(1 - \frac{\eta\lambda\epsilon}{\|\boldsymbol{u}_{(t+k)}\|}\right)^{-1}$. Given the condition $\epsilon < \frac{\min(\|\boldsymbol{u}_{(t+k)}\|, \|\bar{\boldsymbol{u}}\|)}{\eta\lambda}$, it follows $\gamma^{(\boldsymbol{u})}_{(t+k)} > 1$. The recommendation error at $(t+1)^{\text{th}}$ epoch through adversarial loss can be expressed as:

$$
\begin{aligned}
&\mathbb{P}^{\text{adv}}_{\boldsymbol{v}_{(0)} \sim \mathcal{N}(\bar{\boldsymbol{u}}, \frac{\sigma^2}{n-1}I)} \left[f_{(t+k+1)}(\boldsymbol{u}, \boldsymbol{v}) \neq r \mid (\boldsymbol{u}, r)\right] \\
&\qquad\qquad = \mathbb{P}_{\boldsymbol{v}_{(0)} \sim \mathcal{N}(\bar{\boldsymbol{u}}, \frac{\sigma^2}{n-1}I)} \left[r \cdot \langle \boldsymbol{u}_{(t+k)}, \boldsymbol{v}_{(0)} \rangle \leq -\eta(1+\lambda)\gamma^{(\boldsymbol{u})}_{(t+k)}C_{t+k}\|\boldsymbol{v}_{(0)}\|^2\right].
\end{aligned}
$$

With

$$
\mathbb{P}^{\text{adv}}_{\boldsymbol{v}_{(0)} \sim \mathcal{N}(\bar{\boldsymbol{u}}, \frac{\sigma^2}{n-1}I)} \left[f_{(t+k)}(\boldsymbol{u}, \boldsymbol{v}) \neq r \mid (\boldsymbol{u}, r)\right] = \mathbb{P}_{\boldsymbol{v}_{(0)} \sim \mathcal{N}(\bar{\boldsymbol{u}}, \frac{\sigma^2}{n-1}I)} \left[r \cdot \langle \boldsymbol{u}_{(t+k)}, \boldsymbol{v}_{(0)} \rangle \leq 0\right],
$$

the change in recommendation error can be written as:

$$
\begin{aligned}
&\Delta^{\text{adv}}_{(t+k+1)} \mathbb{P}^{\text{adv}}_{\boldsymbol{v}_{(0)} \sim \mathcal{N}(\bar{\boldsymbol{u}}, \frac{\sigma^2}{n-1}I)} [f(\boldsymbol{u}, \boldsymbol{v}) \neq r \mid (\boldsymbol{u}, r)] \\
&= \mathbb{P}_{\boldsymbol{v}_{(0)} \sim \mathcal{N}(\bar{\boldsymbol{u}}, \frac{\sigma^2}{n-1}I)} \left[-\eta(1+\lambda)\gamma^{(\boldsymbol{u})}_{(t+k)}C_{t+k}\|\boldsymbol{v}_{(0)}\|^2 < r \cdot \langle \boldsymbol{u}_{(t+k)}, \boldsymbol{v}_{(0)} \rangle \leq 0\right] \\
&= \mathbb{P}_{\boldsymbol{v}_{(0)} \sim \mathcal{N}(\bar{\boldsymbol{u}}, \frac{\sigma^2}{n-1}I)} \left[-\eta(1+\lambda)\gamma^{(\boldsymbol{u})}_{(t+k)}\frac{C_{t+k}}{\|\boldsymbol{u}_{(t+k)}\|}\|\boldsymbol{v}_{(0)}\|^2 < r \cdot \langle \frac{\boldsymbol{u}_{(t+k)}}{\|\boldsymbol{u}_{(t+k)}\|}, \boldsymbol{v}_{(0)} \rangle \leq 0\right]
\end{aligned}
$$

Considering $\boldsymbol{v}_{(0)} \sim \mathcal{N}\left(\bar{\boldsymbol{u}}, \frac{\sigma^2}{n-1}I\right)$, let $\boldsymbol{\varepsilon} \sim \mathcal{N}\left(\boldsymbol{0}, \frac{\sigma^2}{n-1}\boldsymbol{I}\right)$, we have $\boldsymbol{v}_{(0)} = \bar{\boldsymbol{u}} + \boldsymbol{\varepsilon}$. Then we obtain:

$$
\begin{aligned}
&\Delta^{\text{adv}}_{(t+k+1)} \mathbb{P}^{\text{adv}}_{\boldsymbol{v}_{(0)} \sim \mathcal{N}(\bar{\boldsymbol{u}}, \frac{\sigma^2}{n-1}I)} [f(\boldsymbol{u}, \boldsymbol{v}) \neq r \mid (\boldsymbol{u}, r)] \\
&= \mathbb{P}_{\boldsymbol{\varepsilon} \sim \mathcal{N}\left(\boldsymbol{0}, \frac{\sigma^2}{n-1}\boldsymbol{I}\right)} \left[-\eta(1+\lambda)\gamma^{(\boldsymbol{u})}_{(t+k)}\frac{C_{t+k}}{\|\boldsymbol{u}_{(t+k)}\|}\|\boldsymbol{v}_{(0)}\|^2 < \langle \frac{r\boldsymbol{u}_{(t+k)}}{\|\boldsymbol{u}_{(t+k)}\|}, \bar{\boldsymbol{u}} + \boldsymbol{\varepsilon} \rangle \leq 0\right] \\
&= \mathbb{P}_{\boldsymbol{\varepsilon} \sim \mathcal{N}\left(\boldsymbol{0}, \frac{\sigma^2}{n-1}\boldsymbol{I}\right)} \left[-\eta(1+\lambda)\gamma^{(\boldsymbol{u})}_{(t+k)}\frac{C_{t+k}}{\|\boldsymbol{u}_{(t+k)}\|}\|\boldsymbol{v}_{(0)}\|^2 - \langle \frac{r\boldsymbol{u}_{(t+k)}}{\|\boldsymbol{u}_{(t+k)}\|}, \bar{\boldsymbol{u}} \rangle < \langle \frac{r\boldsymbol{u}_{(t+k)}}{\|\boldsymbol{u}_{(t+k)}\|}, \boldsymbol{\varepsilon} \rangle \leq -\langle \frac{r\boldsymbol{u}_{(t+k)}}{\|\boldsymbol{u}_{(t+k)}\|}, \bar{\boldsymbol{u}} \rangle\right].
\end{aligned}
$$

Given that $\|\bar{\boldsymbol{u}}\| \gg \sigma$, we can approximate the $\|\boldsymbol{v}_{(0)}\|^2$ by using $\mathbb{E}_{\boldsymbol{v}_{(0)} \sim \mathcal{N}(\bar{\boldsymbol{u}}, \frac{\sigma^2}{n-1}I)} \left[\|\boldsymbol{v}_{(0)}\|^2\right]$ as an estimate. Thus, we have:

$$
\begin{aligned}
&\Delta^{\text{adv}}_{(t+k+1)} \mathbb{P}^{\text{adv}}_{\boldsymbol{v}_{(0)} \sim \mathcal{N}(\bar{\boldsymbol{u}}, \frac{\sigma^2}{n-1}I)} [f(\boldsymbol{u}, \boldsymbol{v}) \neq r \mid (\boldsymbol{u}, r)] \\
&\approx \mathbb{P}_{\boldsymbol{\varepsilon} \sim \mathcal{N}\left(\boldsymbol{0}, \frac{\sigma^2}{n-1}\boldsymbol{I}\right)} \left[-\eta(1+\lambda)\gamma^{(\boldsymbol{u})}_{(t+k)}C_{t+k}\frac{\|\bar{\boldsymbol{u}}\|^2 + \frac{d\sigma^2}{n-1}}{\|\boldsymbol{u}_{(t+k)}\|} - \langle \frac{r\boldsymbol{u}_{(t+k)}}{\|\boldsymbol{u}_{(t+k)}\|}, \bar{\boldsymbol{u}} \rangle < \langle \frac{r\boldsymbol{u}_{(t+k)}}{\|\boldsymbol{u}_{(t+k)}\|}, \boldsymbol{\varepsilon} \rangle \leq -\langle \frac{r\boldsymbol{u}_{(t+k)}}{\|\boldsymbol{u}_{(t+k)}\|}, \bar{\boldsymbol{u}} \rangle\right].
\end{aligned}
$$

By Fact 1, we have $\langle \frac{r\boldsymbol{u}_{(t)}}{\|\boldsymbol{u}_{(t)}\|}, \boldsymbol{\varepsilon}\rangle \sim \mathcal{N}\left(0, \frac{\sigma^2}{n-1}\right)$, then:

$$\Delta^{\text{adv}}_{(t+k+1)}\mathbb{P}^{\text{adv}}_{\boldsymbol{v}_{(0)}\sim\mathcal{N}(\bar{\boldsymbol{u}}, \frac{\sigma^2}{n-1}I)}[f(\boldsymbol{u},\boldsymbol{v})\neq r \mid (\boldsymbol{u},r)]$$

$$=\Phi\left(\frac{\sqrt{n-1}}{\sigma}\left(-\langle\frac{r\boldsymbol{u}_{(t+k)}}{\|\boldsymbol{u}_{(t+k)}\|},\bar{\boldsymbol{u}}\rangle\right)\right)-\Phi\left(\frac{\sqrt{n-1}}{\sigma}\left(-\eta(1+\lambda)\gamma^{(\boldsymbol{u})}_{(t+k)}C_{t+k}\frac{\|\bar{\boldsymbol{u}}\|^2+\frac{d\sigma^2}{n-1}}{\|\boldsymbol{u}_{(t+k)}\|}-\langle\frac{r\boldsymbol{u}_{(t+k)}}{\|\boldsymbol{u}_{(t+k)}\|},\bar{\boldsymbol{u}}\rangle\right)\right),$$

where $\Phi()$ is the CDF of standard Gaussian distribution.

Obviously,

$$\langle\frac{r\boldsymbol{u}_{(t+k)}}{\|\boldsymbol{u}_{(t+k)}\|},\bar{\boldsymbol{u}}\rangle\in[-\|\bar{\boldsymbol{u}}\|,\|\bar{\boldsymbol{u}}\|].$$

Let

$$\Psi(\boldsymbol{u},t+k)=(1+\lambda)\gamma^{\boldsymbol{u}}_{(t+k)}\frac{C_{t+k}}{\|\boldsymbol{u}_{(t+k)}\|}.$$

Using the CDF properties of the standard normal distribution, we can conclude:

$$\Delta^{\text{adv}}_{(t+k+1)}\mathbb{P}^{\text{adv}}_{\boldsymbol{v}_{(0)}\sim\mathcal{N}(\bar{\boldsymbol{u}}, \frac{\sigma^2}{n-1}I)}[f(\boldsymbol{u},\boldsymbol{v})\neq r \mid (\boldsymbol{u},r)]\quad\geq$$

$$\Phi\left(\frac{\sqrt{n-1}}{\sigma}\left(\|\bar{\boldsymbol{u}}\|+\eta(\|\bar{\boldsymbol{u}}\|^2+\frac{d\sigma^2}{n-1})\Psi(\boldsymbol{u},t+k)\right)\right)-\Phi\left(\frac{\sqrt{n-1}}{\sigma}(\|\bar{\boldsymbol{u}}\|)\right),$$

equality holds when

$$\langle\frac{r\boldsymbol{u}_{(t+k)}}{\|\boldsymbol{u}_{(t+k)}\|},\bar{\boldsymbol{u}}\rangle=\|\bar{\boldsymbol{u}}\|.$$

Furthermore,

$$\Delta^{\text{adv}}_{(t+k+1)}\mathbb{P}^{\text{adv}}_{\boldsymbol{v}_{(0)}\sim\mathcal{N}(\bar{\boldsymbol{u}}, \frac{\sigma^2}{n-1}I)}[f(\boldsymbol{u},\boldsymbol{v})\neq r \mid (\boldsymbol{u},r)]\quad\leq\quad 2\Phi\left(\frac{\sqrt{n-1}\eta}{2\sigma}(\|\bar{\boldsymbol{u}}\|^2+\frac{d\sigma^2}{n-1})\Psi(\boldsymbol{u},t+k)\right)-1.$$

Equality is achieved when

$$\langle\frac{r\boldsymbol{u}_{(t+k)}}{\|\boldsymbol{u}_{(t+k)}\|},\bar{\boldsymbol{u}}\rangle=-\frac{1}{2}\eta(\|\bar{\boldsymbol{u}}\|^2+\frac{d\sigma^2}{n-1})\Psi(\boldsymbol{u},t+k).$$

Therefore, Theorem 3 is proved. $\qquad\qquad\square$

### D.2.2 Proof of Theorem 4

***Proof of Theorem 4.*** Given $\epsilon<\frac{\|\bar{\boldsymbol{u}}\|}{\eta\lambda}$, according to Proposition 2 and Theorem 1 (specifically Equation 15), we have:

$$\begin{aligned}
\boldsymbol{u}_{(t+k+1)}&=\left(1-\frac{\eta\lambda\epsilon}{\|\boldsymbol{u}_{(t+k)}\|}\right)\boldsymbol{u}_{(t+k)}+\eta(1+\lambda)r\cdot\frac{M_{\text{adv}}(t+k,\eta,\lambda,\epsilon)}{M_{\text{adv}}(t,\eta,\lambda,\epsilon)}M(t,\eta)\boldsymbol{v}'_{(0)},\\
\boldsymbol{v}'_{(t+k+1)}&=\frac{M_{\text{adv}}(t+k+1,\eta,\lambda,\epsilon)}{M_{\text{adv}}(t,\eta,\lambda,\epsilon)}M(t,\eta)\boldsymbol{v}'_{(0)},
\end{aligned}\tag{21}$$

where $\boldsymbol{v}'_{(0)}$ is the poisoned item embedding as given by Equation 18.

Considering the above update rules, the $\alpha$-poisoned recommendation error at the $(t+k+1)^{\text{th}}$ epoch is given by:

$$\begin{aligned}
&\mathbb{P}^{\text{adv}}_{\boldsymbol{v}_{(0)}\sim\mathcal{N}(\bar{\boldsymbol{u}}, \frac{\sigma^2}{n-1}I)}\left[f_{(t+k+1),\alpha}(\boldsymbol{u},\boldsymbol{v}')\neq r \mid (\boldsymbol{u},r)\right]\\
&=\mathbb{P}_{\boldsymbol{v}_{(0)}\sim\mathcal{N}(\bar{\boldsymbol{u}}, \frac{\sigma^2}{n-1}I)}\left[r\cdot\langle\boldsymbol{u}_{(t+k+1)},\boldsymbol{v}'_{(t+k+1)}\rangle\leq 0\right]\\
&=\mathbb{P}_{\boldsymbol{v}_{(0)}\sim\mathcal{N}(\bar{\boldsymbol{u}}, \frac{\sigma^2}{n-1}I)}\left[r\cdot\langle\left(1-\frac{\eta\lambda\epsilon}{\|\boldsymbol{u}_{(t+k)}\|}\right)\boldsymbol{u}_{(t+k)},\boldsymbol{v}'_{(0)}\rangle\leq-\eta(1+\lambda)C_{t+k}\|\boldsymbol{v}'_{(0)}\|^2\right],
\end{aligned}$$

where $C_{t+k} = \frac{M_{\mathrm{adv}}(t+k+1,\eta,\lambda,\epsilon)}{M_{\mathrm{adv}}(t,\eta,\lambda,\epsilon)} M(t,\eta)$. Let $\gamma_{(t+k)}^{(\boldsymbol{u})} = \left(1 - \frac{\eta\lambda\epsilon}{\|\boldsymbol{u}_{(t+k)}\|}\right)^{-1}$. Given the condition $\epsilon < \frac{\min(\|\boldsymbol{u}_{(t+k)}\|, \|\bar{\boldsymbol{u}}\|)}{\eta\lambda}$, it follows $\gamma_{(t+k)}^{(\boldsymbol{u})} > 1$. The recommendation error at the $(t+1)^{\mathrm{th}}$ epoch under adversarial loss can be expressed as:

$$\mathbb{P}^{\mathrm{adv}}_{\boldsymbol{v}_{(0)}\sim\mathcal{N}(\bar{\boldsymbol{u}},\frac{\sigma^2}{n-1}I)} \left[f_{(t+k+1),\alpha}(\boldsymbol{u},\boldsymbol{v}') \neq r \mid (\boldsymbol{u},r)\right]$$
$$= \mathbb{P}_{\boldsymbol{v}_{(0)}\sim\mathcal{N}(\bar{\boldsymbol{u}},\frac{\sigma^2}{n-1}I)} \left[r \cdot \langle \boldsymbol{u}_{(t+k)}, \boldsymbol{v}'_{(0)}\rangle \leq -\eta(1+\lambda)\gamma_{(t+k)}^{(\boldsymbol{u})} C_{t+k}\|\boldsymbol{v}'_{(0)}\|^2\right].$$

With

$$\mathbb{P}^{\mathrm{adv}}_{\boldsymbol{v}_{(0)}\sim\mathcal{N}(\bar{\boldsymbol{u}},\frac{\sigma^2}{n-1}I)} \left[f_{(t+k),\alpha}(\boldsymbol{u},\boldsymbol{v}') \neq r \mid (\boldsymbol{u},r)\right] = \mathbb{P}_{\boldsymbol{v}_{(0)}\sim\mathcal{N}(\bar{\boldsymbol{u}},\frac{\sigma^2}{n-1}I)} \left[r \cdot \langle \boldsymbol{u}_{(t+k)}, \boldsymbol{v}'_{(0)}\rangle \leq 0\right],$$

the change in $\alpha$-poisoned recommendation error can be written as:

$$\Delta^{\mathrm{adv}}_{(t+k+1)}\mathbb{P}^{\mathrm{adv}}_{\boldsymbol{v}_{(0)}\sim\mathcal{N}(\bar{\boldsymbol{u}},\frac{\sigma^2}{n-1}I)}[f_\alpha(\boldsymbol{u},\boldsymbol{v}') \neq r \mid (\boldsymbol{u},r)]$$
$$=\mathbb{P}_{\boldsymbol{v}_{(0)}\sim\mathcal{N}(\bar{\boldsymbol{u}},\frac{\sigma^2}{n-1}I)} \left[-\eta(1+\lambda)\gamma_{(t+k)}^{(\boldsymbol{u})} C_{t+k}\|\boldsymbol{v}'_{(0)}\|^2 < r \cdot \langle \boldsymbol{u}_{(t+k)}, \boldsymbol{v}'_{(0)}\rangle \leq 0\right]$$
$$=\mathbb{P}_{\boldsymbol{v}_{(0)}\sim\mathcal{N}(\bar{\boldsymbol{u}},\frac{\sigma^2}{n-1}I)} \left[-\eta(1+\lambda)\gamma_{(t+k)}^{(\boldsymbol{u})} \frac{C_{t+k}}{\|\boldsymbol{u}_{(t+k)}\|}\|\boldsymbol{v}'_{(0)}\|^2 < r \cdot \langle \frac{\boldsymbol{u}_{(t+k)}}{\|\boldsymbol{u}_{(t+k)}\|}, \boldsymbol{v}'_{(0)}\rangle \leq 0\right]$$
$$=\mathbb{P}_{\boldsymbol{v}_{(0)}\sim\mathcal{N}(\bar{\boldsymbol{u}},\frac{\sigma^2}{n-1}I)} \left[-\eta(1+\lambda)\gamma_{(t+k)}^{(\boldsymbol{u})} \frac{C_{t+k}}{\|\boldsymbol{u}_{(t+k)}\|}\|\boldsymbol{v}'_{(0)}\|^2 < \langle \frac{r\boldsymbol{u}_{(t+k)}}{\|\boldsymbol{u}_{(t+k)}\|}, \frac{1}{n+n'}\left(n\boldsymbol{v}_{(0)} + \sum_{(\boldsymbol{u}',r')\in\mathcal{I}'}(r'\boldsymbol{u}'_{(0)})\right)\rangle \leq 0\right]$$
$$=\mathbb{P}_{\boldsymbol{v}_{(0)}\sim\mathcal{N}(\bar{\boldsymbol{u}},\frac{\sigma^2}{n-1}I)} \Bigg[-\eta\frac{n+n'}{n}(1+\lambda)\gamma_{(t+k)}^{(\boldsymbol{u})} \frac{C_{t+k}}{\|\boldsymbol{u}_{(t+k)}\|}\|\boldsymbol{v}'_{(0)}\|^2 - \frac{1}{n}\sum_{(\boldsymbol{u}',r')\in\mathcal{I}'}rr'\langle \frac{\boldsymbol{u}_{(t+k)}}{\|\boldsymbol{u}_{(t+k)}\|}, \boldsymbol{u}'_{(0)}\rangle$$
$$< \langle \frac{r\boldsymbol{u}_{(t+k)}}{\|\boldsymbol{u}_{(t+k)}\|}, \boldsymbol{v}_{(0)}\rangle \leq -\frac{1}{n}\sum_{(\boldsymbol{u}',r')\in\mathcal{I}'}rr'\langle \frac{\boldsymbol{u}_{(t+k)}}{\|\boldsymbol{u}_{(t+k)}\|}, \boldsymbol{u}'_{(0)}\rangle\Bigg]$$

Considering $\boldsymbol{v}_{(0)} \sim \mathcal{N}\left(\bar{\boldsymbol{u}}, \frac{\sigma^2}{n-1}I\right)$, let $\varepsilon \sim \mathcal{N}\left(\boldsymbol{0}, \frac{\sigma^2}{n-1}\boldsymbol{I}\right)$, we have $\boldsymbol{v}_{(0)} = \bar{\boldsymbol{u}} + \varepsilon$. Then we obtain:

$$\Delta^{\mathrm{adv}}_{(t+k+1)}\mathbb{P}^{\mathrm{adv}}_{\boldsymbol{v}_{(0)}\sim\mathcal{N}(\bar{\boldsymbol{u}},\frac{\sigma^2}{n-1}I)}[f_\alpha(\boldsymbol{u},\boldsymbol{v}') \neq r \mid (\boldsymbol{u},r)]$$
$$=\mathbb{P}_{\varepsilon\sim\mathcal{N}(\boldsymbol{0},\frac{\sigma^2}{n-1}\boldsymbol{I})} \Bigg[-\eta\frac{n+n'}{n}(1+\lambda)\gamma_{(t+k)}^{(\boldsymbol{u})} \frac{C_{t+k}}{\|\boldsymbol{u}_{(t+k)}\|}\|\boldsymbol{v}'_{(0)}\|^2 - \frac{1}{n}\sum_{(\boldsymbol{u}',r')\in\mathcal{I}'}rr'\langle \frac{\boldsymbol{u}_{(t+k)}}{\|\boldsymbol{u}_{(t+k)}\|}, \boldsymbol{u}'_{(0)}\rangle$$
$$< \langle \frac{r\boldsymbol{u}_{(t+k)}}{\|\boldsymbol{u}_{(t+k)}\|}, \bar{\boldsymbol{u}} + \varepsilon\rangle \leq -\frac{1}{n}\sum_{(\boldsymbol{u}',r')\in\mathcal{I}'}rr'\langle \frac{\boldsymbol{u}_{(t+k)}}{\|\boldsymbol{u}_{(t+k)}\|}, \boldsymbol{u}'_{(0)}\rangle\Bigg]$$
$$=\mathbb{P}_{\varepsilon\sim\mathcal{N}(\boldsymbol{0},\frac{\sigma^2}{n-1}\boldsymbol{I})} \Bigg[-\eta\frac{n+n'}{n}(1+\lambda)\gamma_{(t+k)}^{(\boldsymbol{u})} \frac{C_{t+k}}{\|\boldsymbol{u}_{(t+k)}\|}\|\boldsymbol{v}'_{(0)}\|^2 - \frac{1}{n}\sum_{(\boldsymbol{u}',r')\in\mathcal{I}'}rr'\langle \frac{\boldsymbol{u}_{(t+k)}}{\|\boldsymbol{u}_{(t+k)}\|}, \boldsymbol{u}'_{(0)}\rangle - \langle \frac{r\boldsymbol{u}_{(t+k)}}{\|\boldsymbol{u}_{(t+k)}\|}, \bar{\boldsymbol{u}}\rangle$$
$$< \langle \frac{r\boldsymbol{u}_{(t+k)}}{\|\boldsymbol{u}_{(t+k)}\|}, \varepsilon\rangle \leq -\frac{1}{n}\sum_{(\boldsymbol{u}',r')\in\mathcal{I}'}rr'\langle \frac{\boldsymbol{u}_{(t+k)}}{\|\boldsymbol{u}_{(t+k)}\|}, \boldsymbol{u}'_{(0)}\rangle - \langle \frac{r\boldsymbol{u}_{(t+k)}}{\|\boldsymbol{u}_{(t+k)}\|}, \bar{\boldsymbol{u}}\rangle\Bigg].$$

Given that $\|\bar{\boldsymbol{u}}\| \gg \sigma$, we can approximate $\|\boldsymbol{v}'_{(0)}\|^2$ by using:

$$\mathbb{E}_{\boldsymbol{v}_{(0)}\sim\mathcal{N}(\bar{\boldsymbol{u}},\frac{\sigma^2}{n-1}I)} \left[\|\boldsymbol{v}'_{(0)}\|^2\right] = \mathbb{E}_{\boldsymbol{v}_{(0)}\sim\mathcal{N}(\bar{\boldsymbol{u}},\frac{\sigma^2}{n-1}I)} \left[\|\frac{n}{n+n'}\boldsymbol{v}_{(0)} + \frac{1}{n+n'}\sum_{(\boldsymbol{u}',r)\in\mathcal{I}}r'\boldsymbol{u}'\|^2\right],$$

as an estimate. For simplicity, we use $\mathbb{E}\left[\|\boldsymbol{v}'_{(0)}\|^2\right]$ to represent $\mathbb{E}_{\boldsymbol{v}_{(0)}\sim\mathcal{N}(\bar{\boldsymbol{u}},\frac{\sigma^2}{n-1}I)}\left[\|\boldsymbol{v}'_{(0)}\|^2\right]$. Thus, we have:

$$\Delta^{\text{adv}}_{(t+k+1)} \mathbb{P}^{\text{adv}}_{\boldsymbol{v}_{(0)} \sim \mathcal{N}(\bar{\boldsymbol{u}}, \frac{\sigma^2}{n-1} I)} [f_\alpha(\boldsymbol{u}, \boldsymbol{v}') \neq r \mid (\boldsymbol{u}, r)]$$

$$\approx \mathbb{P}_{\boldsymbol{\varepsilon} \sim \mathcal{N}\left(\boldsymbol{0}, \frac{\sigma^2}{n-1} \boldsymbol{I}\right)} \left[ -\eta \frac{n+n'}{n} (1+\lambda) \gamma^{(\boldsymbol{u})}_{(t+k)} \frac{C_{t+k}}{\|\boldsymbol{u}_{(t+k)}\|} \mathbb{E}\left[\|\boldsymbol{v}'_{(0)}\|^2\right] - \frac{1}{n} \sum_{(\boldsymbol{u}', r') \in \mathcal{I}'} rr' \langle \frac{\boldsymbol{u}_{(t+k)}}{\|\boldsymbol{u}_{(t+k)}\|}, \boldsymbol{u}'_{(0)} \rangle - \langle \frac{r\boldsymbol{u}_{(t+k)}}{\|\boldsymbol{u}_{(t+k)}\|}, \bar{\boldsymbol{u}} \rangle \right.$$

$$\left. < \langle \frac{r\boldsymbol{u}_{(t+k)}}{\|\boldsymbol{u}_{(t+k)}\|}, \boldsymbol{\varepsilon} \rangle \leq -\frac{1}{n} \sum_{(\boldsymbol{u}', r') \in \mathcal{I}'} rr' \langle \frac{\boldsymbol{u}_{(t+k)}}{\|\boldsymbol{u}_{(t+k)}\|}, \boldsymbol{u}'_{(0)} \rangle - \langle \frac{r\boldsymbol{u}_{(t+k)}}{\|\boldsymbol{u}_{(t+k)}\|}, \bar{\boldsymbol{u}} \rangle \right].$$

By Fact 1, we have $\langle \frac{r\boldsymbol{u}_{(t+k)}}{\|\boldsymbol{u}_{(t+k)}\|}, \boldsymbol{\varepsilon} \rangle \sim \mathcal{N}\left(0, \frac{\sigma^2}{n-1}\right)$, then:

$$\Delta^{\text{adv}}_{(t+k+1)} \mathbb{P}^{\text{adv}}_{\boldsymbol{v}_{(0)} \sim \mathcal{N}(\bar{\boldsymbol{u}}, \frac{\sigma^2}{n-1} I)} [f_\alpha(\boldsymbol{u}, \boldsymbol{v}') \neq r \mid (\boldsymbol{u}, r)]$$

$$= \Phi\left( \frac{\sqrt{n-1}}{\sigma} \left( -\frac{1}{n} \sum_{(\boldsymbol{u}', r') \in \mathcal{I}'} rr' \langle \frac{\boldsymbol{u}_{(t+k)}}{\|\boldsymbol{u}_{(t+k)}\|}, \boldsymbol{u}'_{(0)} \rangle - \langle \frac{r\boldsymbol{u}_{(t+k)}}{\|\boldsymbol{u}_{(t+k)}\|}, \bar{\boldsymbol{u}} \rangle \right) \right)$$

$$- \Phi\left( \frac{\sqrt{n-1}}{\sigma} \left( -\eta \frac{n+n'}{n} (1+\lambda) \gamma^{(\boldsymbol{u})}_{(t+k)} \frac{C_{t+k}}{\|\boldsymbol{u}_{(t+k)}\|} \mathbb{E}\left[\|\boldsymbol{v}'_{(0)}\|^2\right] - \frac{1}{n} \sum_{(\boldsymbol{u}', r') \in \mathcal{I}'} rr' \langle \frac{\boldsymbol{u}_{(t+k)}}{\|\boldsymbol{u}_{(t+k)}\|}, \boldsymbol{u}'_{(0)} \rangle - \langle \frac{r\boldsymbol{u}_{(t+k)}}{\|\boldsymbol{u}_{(t+k)}\|}, \bar{\boldsymbol{u}} \rangle \right) \right),$$

where $\Phi()$ is the CDF of standard Gaussian distribution.

Obviously,

$$\sum_{(\boldsymbol{u}', r') \in \mathcal{I}'} rr' \langle \frac{\boldsymbol{u}_{(t+k)}}{\|\boldsymbol{u}_{(t+k)}\|}, \boldsymbol{u}'_{(0)} \rangle \in \left[ -n'\sqrt{d}\alpha, n'\sqrt{d}\alpha \right],$$

$$\langle \frac{r\boldsymbol{u}_{(t+k)}}{\|\boldsymbol{u}_{(t+k)}\|}, \bar{\boldsymbol{u}} \rangle \in \left[ -\|\bar{\boldsymbol{u}}\|, \|\bar{\boldsymbol{u}}\| \right],$$

$$\mathbb{E}\left[\|\boldsymbol{v}'_{(0)}\|^2\right] \in \left( \frac{n^2\|\bar{\boldsymbol{u}}\|^2 - 2nn'\alpha\|\bar{\boldsymbol{u}}\|_0}{(n+n')^2} + \frac{n^2 d\sigma^2}{(n-1)(n+n')^2}, \frac{n^2\|\bar{\boldsymbol{u}}\|^2 + (n')^2 d\alpha^2 + 2nn'\alpha\|\bar{\boldsymbol{u}}\|_0}{(n+n')^2} + \frac{n^2 d\sigma^2}{(n-1)(n+n')^2} \right]$$

where $n'$ is the number of fake users, $d$ is the dimension of $\boldsymbol{u}'$, and $\alpha = \max_{(\boldsymbol{u}', r') \in \mathcal{I}'} \|\boldsymbol{u}'\|_\infty$.

Let

$$\Psi(\boldsymbol{u}, t+k) = (1+\lambda) \gamma^{\boldsymbol{u}}_{(t+k)} \frac{C_{t+k}}{\|\boldsymbol{u}_{(t+k)}\|},$$

$$\beta = \frac{n'}{n} \sqrt{d}\alpha + \|\bar{\boldsymbol{u}}\|.$$

According to the CDF properties of the standard normal distribution, we can conclude that:

$$\Delta^{\text{adv}}_{(t+k+1)} \mathbb{P}^{\text{adv}}_{\boldsymbol{v}_{(0)} \sim \mathcal{N}(\bar{\boldsymbol{u}}, \frac{\sigma^2}{n-1} I)} [f_\alpha(\boldsymbol{u}, \boldsymbol{v}') \neq r \mid (\boldsymbol{u}, r)] \quad >$$

$$\Phi\left( \frac{\sqrt{n-1}}{\sigma} \left( \beta + \eta \left( \frac{n^2\|\bar{\boldsymbol{u}}\|^2 - 2nn'\alpha\|\bar{\boldsymbol{u}}\|_0}{n(n+n')} + \frac{nd\sigma^2}{(n-1)(n+n')} \right) \Psi(\boldsymbol{u}, t+k) \right) \right) - \Phi\left( \frac{\sqrt{n-1}}{\sigma} (\beta) \right),$$

reaches the minimum value when

$$\sum_{(\boldsymbol{u}', r') \in \mathcal{I}'} rr' \langle \frac{\boldsymbol{u}_{(t+k)}}{\|\boldsymbol{u}_{(t+k)}\|}, \boldsymbol{u}'_{(0)} \rangle = n'\sqrt{d}\alpha,$$

$$\langle \frac{r\boldsymbol{u}_{(t+k)}}{\|\boldsymbol{u}_{(t+k)}\|}, \bar{\boldsymbol{u}} \rangle = \|\bar{\boldsymbol{u}}\|,$$

and $\mathbb{E}\left[\|\boldsymbol{v}'_{(0)}\|^2\right]$ reaches the minimum value $\frac{n^2\|\bar{\boldsymbol{u}}\|^2 - 2nn'\alpha\|\bar{\boldsymbol{u}}\|_0}{(n+n')^2} + \frac{n^2 d\sigma^2}{(n-1)(n+n')^2}$.

Moreover,

$$\Delta^{\text{adv}}_{(t+k+1)} \mathbb{P}^{\text{adv}}_{\boldsymbol{v}_{(0)} \sim \mathcal{N}(\bar{\boldsymbol{u}}, \frac{\sigma^2}{n-1} I)} [f_\alpha(\boldsymbol{u}, \boldsymbol{v}') \neq r \mid (\boldsymbol{u}, r)] \quad \leq$$

$$2\Phi\left( \frac{\sqrt{n-1}\eta}{2\sigma} \left( \frac{n^2\|\bar{\boldsymbol{u}}\|^2 + (n')^2\alpha + 2nn'\alpha\|\bar{\boldsymbol{u}}\|_0}{n(n+n')} + \frac{nd\sigma^2}{(n-1)(n+n')} \right) \Psi(\boldsymbol{u}, t+k) \right) - 1,$$

equality holds when

$$\mathbb{E}\left[\|\bm{v}'_{(0)}\|^2\right] = \frac{n^2\|\bar{\bm{u}}\|^2 + (n')^2\alpha + 2nn'\alpha\|\bar{\bm{u}}\|_0}{(n+n')^2} + \frac{n^2 d\sigma^2}{(n-1)(n+n')^2},$$

$$\frac{1}{n}\sum_{(\bm{u}',r')\in\mathcal{I}'} rr'\langle \frac{\bm{u}_{(t+k)}}{\|\bm{u}_{(t+k)}\|}, \bm{u}'_{(0)}\rangle + \langle \frac{r\bm{u}_{(t+k)}}{\|\bm{u}_{(t+k)}\|}, \bar{\bm{u}}\rangle = -\frac{1}{2}\eta\frac{n+n'}{n}\mathbb{E}\left[\|\bm{v}'_{(0)}\|^2\right]\Psi(\bm{u},t+k).$$

Hence, Theorem 4 is proved. $\qquad\square$

### D.3 Proofs for Section 4

Given any dot-product-based loss function $\mathcal{L}(\Theta)$, characterized by its dependency on the product of user and item embeddings, the gradients of user and item embeddings at the $t^{\text{th}}$ epoch can be expressed as follows:

$$\begin{aligned}
\nabla_{\bm{u}_{(t)}}\mathcal{L}(\bm{u},\bm{v}|\Theta_{(t)}) &= \phi(r,\bm{u}_{(t)},\bm{v}_{(t)})\bm{v}_{(t)}, \\
\nabla_{\bm{v}_{(t)}}\mathcal{L}(\bm{u},\bm{v}|\Theta_{(t)}) &= \psi(r,\bm{u}_{(t)},\bm{v}_{(t)})\bm{u}_{(t)},
\end{aligned} \tag{22}$$

where $\phi(\cdot)$ and $\psi(\cdot)$ denote coefficient functions derived from $\mathcal{L}(\Theta)$, mapping from the embeddings' space to the scalar values.

Considering the proofs of Theorem 3 and Theorem 4, there is a coefficient $\gamma_{(t)}^{(\bm{u})}$ for the user $\bm{u}$. When $\gamma_{(t)}^{(\bm{u})} > 1$ is satisfied, the effectiveness of ACF can be guaranteed. Here, we derive the coefficient $\gamma_{(t)}^{(\bm{u})}$ in multi-item recommendation scenarios with dot-product-based loss through the following corollary.

**Corollary 2.** *Assuming the incorporation of adversarial training as defined in Equation 1 with dot-product-based loss function $\mathcal{L}(\Theta)$, and given the learning rate $\eta$, the adversarial training weight $\lambda$, and the perturbation scale $\epsilon$, the $\gamma_{(t)}^{(\bm{u})}$ for user $\bm{u}$ is given by:*

$$\gamma_{(t)}^{(\bm{u})} = \left(1 - \frac{\eta\lambda\epsilon}{\|\bm{u}_t\|}\sum_{\bm{v}\in\mathcal{N}_{\bm{u}}}|\psi(r,\bm{u}_{(t)},\bm{v}_{(t)})|\right)^{-1}, \tag{23}$$

*where $\mathcal{N}_{\bm{u}}$ is the item set that user $\bm{u}$ interacts with.*

***Proof of Corollary 2.*** Recall Equation 1. Given a dot-product-based loss function $\mathcal{L}(\Theta)$ within the framework of adversarial training:

$$\begin{aligned}
\mathcal{L}_{\text{ACF}}(\Theta) &= \mathcal{L}(\Theta) + \lambda\mathcal{L}(\Theta + \Delta^{\text{adv}}), \\
\text{where}\quad \Delta^{\text{adv}} &= \arg\max_{\Delta,\|\Delta\|\leq\epsilon}\mathcal{L}(\Theta + \Delta),
\end{aligned}$$

where $\epsilon > 0$ defines the magnitude of perturbation, and $\lambda$ is the adversarial training weight. Considering any pair $(\bm{u},\bm{v})$, the perturbations can be computed as (similar to Equation 14):

$$\begin{aligned}
\Delta_{\bm{u}_{(t)}} &= \epsilon\frac{\nabla_{\bm{u}_{(t)}}\mathcal{L}(\bm{u},\bm{v}|\Theta_{(t)})}{\|\nabla_{\bm{u}_{(t)}}\mathcal{L}(\bm{u},\bm{v}|\Theta_{(t)})\|}, \\
\Delta_{\bm{v}_{(t)}} &= \epsilon\frac{\nabla_{\bm{v}_{(t)}}\mathcal{L}(\bm{u},\bm{v}|\Theta_{(t)})}{\|\nabla_{\bm{v}_{(t)}}\mathcal{L}(\bm{u},\bm{v}|\Theta_{(t)})\|}.
\end{aligned}$$

The update equations for the embedding of user $\bm{u}$ at the $t^{\text{th}}$ epoch under adversarial perturbations can be expressed as follows:

$$\bm{u}_{(t+1)} = \bm{u}_{(t)} - \sum_{\bm{v}\in\mathcal{N}_{\bm{u}}}\left(\eta\cdot\nabla_{\bm{u}_{(t)}}\mathcal{L}(\bm{u},\bm{v}\mid\Theta_{(t)}) + \eta\lambda\nabla_{\bm{u}_{(t)}}\mathcal{L}(\bm{u},\bm{v}\mid\Theta_{(t)}+\Delta_{\text{adv}})\right),$$

where $\mathcal{N}_{\bm{u}}$ is the set of items that user $\bm{u}$ interacts with.

By employing the first-order Taylor expansion on $(\boldsymbol{u}_{(t)} + \Delta_{\boldsymbol{u}_{(t)}})$ and $(\boldsymbol{v}_{(t)} + \Delta_{\boldsymbol{v}_{(t)}})$ , we have:

$$
\begin{aligned}
\nabla_{\boldsymbol{u}_{(t)}} &\mathcal{L}(\boldsymbol{u}, \boldsymbol{v} | \Theta_{(t)} + \Delta_{\mathrm{adv}}) \\
&\approx \nabla_{\boldsymbol{u}_{(t)}} \left[ \mathcal{L}(\boldsymbol{u}, \boldsymbol{v} | \Theta_{(t)}) + \langle \Delta_{\boldsymbol{v}_{(t)}}, \nabla_{\boldsymbol{v}_{(t)}} \mathcal{L}(\boldsymbol{u}, \boldsymbol{v} | \Theta_{(t)}) \rangle + \langle \Delta_{\boldsymbol{u}_{(t)}}, \nabla_{\boldsymbol{u}_{(t)}} \mathcal{L}(\boldsymbol{u}, \boldsymbol{v} | \Theta_{(t)}) \rangle \right. \\
&\qquad\qquad \left. + \langle \Delta_{\boldsymbol{v}_{(t)}}, \left( \nabla_{\boldsymbol{u}_{(t)}} \nabla_{\boldsymbol{v}_{(t)}} \mathcal{L}(\boldsymbol{u}, \boldsymbol{v} | \Theta_{(t)}) \right)^{\top} \Delta_{\boldsymbol{u}_{(t)}} \rangle \right] \\
&\approx \nabla_{\boldsymbol{u}_{(t)}} \mathcal{L}(\boldsymbol{u}, \boldsymbol{v} | \Theta_{(t)}) + \left( \nabla_{\boldsymbol{u}_{(t)}} \nabla_{\boldsymbol{v}_{(t)}} \mathcal{L}(\boldsymbol{u}, \boldsymbol{v} | \Theta_{(t)}) \right)^{\top} \Delta_{\boldsymbol{v}_{(t)}} + \left( \nabla^2_{\boldsymbol{u}_{(t)}} \mathcal{L}(\boldsymbol{u}, \boldsymbol{v} | \Theta_{(t)}) \right)^{\top} \Delta_{\boldsymbol{u}_{(t)}}.
\end{aligned}
$$

Subsequently, the update mechanism for the user embedding, incorporating both direct and adversarial gradients, is computed as:

$$
\begin{aligned}
\boldsymbol{u}_{(t+1)} =& \boldsymbol{u}_{(t)} - \sum_{\boldsymbol{v} \in \mathcal{N}_{\boldsymbol{u}}} \left( \eta \cdot (1 + \lambda) \nabla_{\boldsymbol{u}_{(t)}} \mathcal{L}(\boldsymbol{u}, \boldsymbol{v} | \Theta_{(t)}) + \eta \lambda \epsilon \left( \nabla_{\boldsymbol{u}_{(t)}} \nabla_{\boldsymbol{v}_{(t)}} \mathcal{L}(\boldsymbol{u}, \boldsymbol{v} | \Theta_{(t)}) \right)^{\top} \left( \frac{\nabla_{\boldsymbol{v}_{(t)}} \mathcal{L}(\boldsymbol{u}, \boldsymbol{v} | \Theta_{(t)})}{\|\nabla_{\boldsymbol{v}_{(t)}} \mathcal{L}(\boldsymbol{u}, \boldsymbol{v} | \Theta_{(t)})\|} \right) \right. \\
&\qquad\qquad\qquad \left. + \eta \lambda \epsilon \left( \nabla^2_{\boldsymbol{u}_{(t)}} \mathcal{L}(\boldsymbol{u}, \boldsymbol{v} | \Theta_{(t)}) \right)^{\top} \left( \frac{\nabla_{\boldsymbol{u}_{(t)}} \mathcal{L}(\boldsymbol{u}, \boldsymbol{v} | \Theta_{(t)})}{\|\nabla_{\boldsymbol{u}_{(t)}} \mathcal{L}(\boldsymbol{u}, \boldsymbol{v} | \Theta_{(t)})\|} \right) \right) \\
=& \boldsymbol{u}_{(t)} - \sum_{\boldsymbol{v} \in \mathcal{N}_{\boldsymbol{u}}} \eta(1 + \lambda) \phi(r, \boldsymbol{u}_{(t)}, \boldsymbol{v}_{(t)}) \boldsymbol{v}_{(t)} \\
&- \sum_{\boldsymbol{v} \in \mathcal{N}_{\boldsymbol{u}}} \eta \lambda \epsilon \frac{\psi(r, \boldsymbol{u}_{(t)}, \boldsymbol{v}_{(t)}) \cdot \left( \boldsymbol{u}_{(t)} \left( \nabla_{\boldsymbol{u}_{(t)}} \psi(r, \boldsymbol{u}_{(t)}, \boldsymbol{v}_{(t)}) \right)^{\top} + \psi(r, \boldsymbol{u}_{(t)}, \boldsymbol{v}_{(t)}) \cdot \boldsymbol{I} \right)^{\top} \boldsymbol{u}_{(t)}}{\|\nabla_{\boldsymbol{v}_{(t)}} \mathcal{L}(\boldsymbol{u}, \boldsymbol{v} | \Theta_{(t)})\|} \\
&- \sum_{\boldsymbol{v} \in \mathcal{N}_{\boldsymbol{u}}} \eta \lambda \epsilon \frac{\phi(r, \boldsymbol{u}_{(t)}, \boldsymbol{v}_{(t)}) \cdot \left( \boldsymbol{v}_{(t)} \left( \nabla_{\boldsymbol{u}_{(t)}} \phi(r, \boldsymbol{u}_{(t)}, \boldsymbol{v}_{(t)}) \right)^{\top} \right)^{\top} \boldsymbol{v}_{(t)}}{\|\nabla_{\boldsymbol{u}_{(t)}} \mathcal{L}(\boldsymbol{u}, \boldsymbol{v} | \Theta_{(t)})\|} \\
=& \boldsymbol{u}_{(t)} - \sum_{\boldsymbol{v} \in \mathcal{N}_{\boldsymbol{u}}} \eta(1 + \lambda) \phi(r, \boldsymbol{u}_{(t)}, \boldsymbol{v}_{(t)}) \boldsymbol{v}_{(t)} \\
&- \sum_{\boldsymbol{v} \in \mathcal{N}_{\boldsymbol{u}}} \eta \lambda \epsilon \frac{\psi^2(r, \boldsymbol{u}_{(t)}, \boldsymbol{v}_{(t)})}{\|\nabla_{\boldsymbol{v}_{(t)}} \mathcal{L}(\boldsymbol{u}, \boldsymbol{v} | \Theta_{(t)})\|} \boldsymbol{u}_{(t)} \\
&- \sum_{\boldsymbol{v} \in \mathcal{N}_{\boldsymbol{u}}} \eta \lambda \epsilon \frac{\psi(r, \boldsymbol{u}_{(t)}, \boldsymbol{v}_{(t)}) \|\boldsymbol{u}_{(t)}\|^2}{\|\nabla_{\boldsymbol{v}_{(t)}} \mathcal{L}(\boldsymbol{u}, \boldsymbol{v} | \Theta_{(t)})\|} \nabla_{\boldsymbol{u}_{(t)}} \psi(r, \boldsymbol{u}_{(t)}, \boldsymbol{v}_{(t)}) \\
&- \sum_{\boldsymbol{v} \in \mathcal{N}_{\boldsymbol{u}}} \eta \lambda \epsilon \frac{\phi(r, \boldsymbol{u}_{(t)}, \boldsymbol{v}_{(t)}) \|\boldsymbol{v}_{(t)}\|^2}{\|\nabla_{\boldsymbol{u}_{(t)}} \mathcal{L}(\boldsymbol{u}, \boldsymbol{v} | \Theta_{(t)})\|} \nabla_{\boldsymbol{u}_{(t)}} \phi(r, \boldsymbol{u}_{(t)}, \boldsymbol{v}_{(t)}).
\end{aligned}
$$

Considering Equation 22, for a loss function $\mathcal{L}(\Theta)$ based on dot-products, the coefficients of gradients for user embedding $\boldsymbol{u}$ and item embedding $\boldsymbol{v}$, denoted as $\psi(\cdot)$ and $\phi(\cdot)$ respectively, are still functions based on the dot-product of user embedding $\boldsymbol{u}$ and item embedding $\boldsymbol{v}$. Consequently, the gradients of $\psi(\cdot)$ and $\phi(\cdot)$ with respect to user embedding $\boldsymbol{u}$ depend on item embedding $\boldsymbol{v}$. Specifically, $\nabla_{\boldsymbol{u}_{(t)}} \psi(r, \boldsymbol{u}_{(t)}, \boldsymbol{v}_{(t)}) = \xi(r, \boldsymbol{u}_{(t)}, \boldsymbol{v}_{(t)}) \boldsymbol{v}_{(t)}$ and $\nabla_{\boldsymbol{u}_{(t)}} \phi(r, \boldsymbol{u}_{(t)}, \boldsymbol{v}_{(t)}) = \xi^{'}(r, \boldsymbol{u}_{(t)}, \boldsymbol{v}_{(t)}) \boldsymbol{v}_{(t)}$. Thus, the updated expression for the user embedding $\boldsymbol{u}_{(t+1)}$ under adversarial training conditions is delineated as follows:

$$
\begin{aligned}
\boldsymbol{u}_{(t+1)} = & \left( 1 - \sum_{\boldsymbol{v} \in \mathcal{N}_{\boldsymbol{u}}} \eta \lambda \epsilon \frac{\psi^2(r, \boldsymbol{u}_{(t)}, \boldsymbol{v}_{(t)})}{\|\nabla_{\boldsymbol{v}_{(t)}} \mathcal{L}(\boldsymbol{u}, \boldsymbol{v} | \Theta_{(t)})\|} \right) \boldsymbol{u}_{(t)} \\
& - \eta \sum_{\boldsymbol{v} \in \mathcal{N}_{\boldsymbol{u}}} \left( (1 + \lambda) \phi(r, \boldsymbol{u}_{(t)}, \boldsymbol{v}_{(t)}) + \lambda \epsilon \frac{\psi(r, \boldsymbol{u}_{(t)}, \boldsymbol{v}_{(t)}) \xi(r, \boldsymbol{u}_{(t)}, \boldsymbol{v}_{(t)}) \|\boldsymbol{u}_{(t)}\|^2}{\|\nabla_{\boldsymbol{v}_{(t)}} \mathcal{L}(\boldsymbol{u}, \boldsymbol{v} | \Theta_{(t)})\|} \right. \\
& \qquad\qquad \left. + \lambda \epsilon \frac{\phi(r, \boldsymbol{u}_{(t)}, \boldsymbol{v}_{(t)}) \xi^{'}(r, \boldsymbol{u}_{(t)}, \boldsymbol{v}_{(t)}) \|\boldsymbol{v}_{(t)}\|^2}{\|\nabla_{\boldsymbol{u}_{(t)}} \mathcal{L}(\boldsymbol{u}, \boldsymbol{v} | \Theta_{(t)})\|} \right) \boldsymbol{v}_{(t)}
\end{aligned}
$$

Following the aforementioned Equation 16 and Equation 20, the $\gamma^{(\boldsymbol{u})}$ for the user $\boldsymbol{u}$ in the context of multi-item ACF with a dot-product loss function is given by:

$$
\begin{aligned}
\gamma_{(t)}^{(\boldsymbol{u})} &= \left(1 - \sum_{\boldsymbol{v} \in \mathcal{N}_{\boldsymbol{u}}} \eta\lambda\epsilon \frac{\psi^2(r, \boldsymbol{u}_{(t)}, \boldsymbol{v}_{(t)})}{\|\nabla_{\boldsymbol{v}_{(t)}}\mathcal{L}(\boldsymbol{u}, \boldsymbol{v}|\Theta_{(t)})\|}\right)^{-1} \\
&= \left(1 - \sum_{\boldsymbol{v} \in \mathcal{N}_{\boldsymbol{u}}} \eta\lambda\epsilon \frac{|\psi(r, \boldsymbol{u}_{(t)}, \boldsymbol{v}_{(t)})||\psi(r, \boldsymbol{u}_{(t)}, \boldsymbol{v}_{(t)})|}{\|\nabla_{\boldsymbol{v}_{(t)}}\mathcal{L}(\boldsymbol{u}, \boldsymbol{v}|\Theta_{(t)})\|}\right)^{-1} \quad (24) \\
&= \left(1 - \frac{\eta\lambda\epsilon}{\|\boldsymbol{u}_t\|} \sum_{\boldsymbol{v} \in \mathcal{N}_{\boldsymbol{u}}} |\psi(r, \boldsymbol{u}_{(t)}, \boldsymbol{v}_{(t)})|\right)^{-1}.
\end{aligned}
$$

Therefore, Corollary 2 is proved. $\qquad\square$

***Proof of Corollary 1.*** For any dot-product-based loss function $\mathcal{L}(\boldsymbol{u}, \boldsymbol{v}|\Theta)$, the coefficient functions in Equation 22 can be given by:

$$
\begin{aligned}
\nabla_{\boldsymbol{u}_{(t)}}\mathcal{L}(\boldsymbol{u}_{(t)}, \boldsymbol{v}_{(t)}|\Theta) &= \phi(r, \boldsymbol{u}_{(t)}, \boldsymbol{v}_{(t)})\boldsymbol{v}_{(t)}, \\
\nabla_{\boldsymbol{v}_{(t)}}\mathcal{L}(\boldsymbol{u}_{(t)}, \boldsymbol{v}_{(t)}|\Theta) &= \psi(r, \boldsymbol{u}_{(t)}, \boldsymbol{v}_{(t)})\boldsymbol{u}_{(t)}.
\end{aligned}
$$

Building upon Corollary 2, we can express $\gamma_{(t)}^{(\boldsymbol{u})}$ as:

$$
\gamma_{(t)}^{(\boldsymbol{u})} = \left(1 - \frac{\eta\lambda\epsilon}{\|\boldsymbol{u}_t\|} \sum_{\boldsymbol{v} \in \mathcal{N}_{\boldsymbol{u}}} |\psi(r, \boldsymbol{u}_{(t)}, \boldsymbol{v}_{(t)})|\right)^{-1}.
$$

Considering the proofs of Theorem 3 and Theorem 4, under $0 < (\gamma_{(t)}^{(\boldsymbol{u})})^{-1} < 1$, we can guarantee the effectiveness of ACF. Therefore, it implies:

$$
0 < \epsilon_{(t)}^{(\boldsymbol{u})} < \|\boldsymbol{u}_{(t)}\| \cdot \frac{1}{\sum_{\boldsymbol{v} \in \mathcal{N}_{\boldsymbol{u}}} \eta\lambda|\psi(r, \boldsymbol{u}_{(t)}, \boldsymbol{v}_{(t)})|}.
$$

In actual training, the maximum perturbation magnitudes will also be affected by other factors. From the perspective of Corollary 2, we can only conclude that the maximum perturbation magnitude $\epsilon_{(t),\max}^{(\boldsymbol{u})}$ for user $\boldsymbol{u}$ at epoch $t$ is positively related to $\|\boldsymbol{u}_{(t)}\|$.

Therefore, Corollary 1 is proved. $\qquad\square$

