# OpenReview forum: "Understanding and Improving Adversarial Collaborative Filtering for Robust Recommendation"
_NeurIPS.cc/2024/Conference — NeurIPS 2024 poster_

### Official Review · Reviewer_MDNE · 2024-07-06

**Soundness:** 3
**Presentation:** 4
**Contribution:** 3
**Rating:** 7
**Confidence:** 3

**Summary:**

This work investigates the efficacy and robustness of Adversarial Collaborative Filtering (ACF) in recommender systems. The authors present theoretical analyses to demonstrate how ACF enhances traditional collaborative filtering (CF) by mitigating the negative impact of data poisoning attacks. They extend these theoretical insights to further improve existing ACF methods by dynamically and personally assigning perturbation magnitudes based on users’ embedding scales. Experiments based on representative CF backbones and various attack methods strengthen the validity of this work.

**Strengths:**

S1. The work addresses a critical issue in recommender systems—robustness against attacks. The paper introduces novel theoretical insights into the benefits of ACF over traditional CF, particularly in mitigating data poisoning attacks. The combination of theoretical and empirical analyses enhances the originality of the work.

S2. The authors provide rigorous mathematical proofs and extensive experimental evaluations to support their claims. The inclusion of multiple datasets and various attack methods strengthens the validity of their findings.

S3. The paper is well-structured, with clear explanations of theoretical concepts and experimental methodologies. The use of figures and tables helps in comprehensively presenting the results.

**Weaknesses:**

W1. To extend theoretical understandings from the simple single-item CF scenario to practical multi-item scenarios,  Corollary 1 is restricted to the dot-product-based loss function.

W2. The experiment only involves two CF backbones, namely MF and LightGCN. The results will be more convincing by adding more CF methods.

**Questions:**

Please refer to the Weaknesses section.

**Limitations:**

The authors have discussed the limitations of their work.

---

> ### Author Rebuttal · Authors · 2024-08-06
>
> Thanks for your positive feedback. We deeply value your recognition of our work's contributions, particularly its importance, clear explanations, novel theoretical insights, rigorous mathematical proofs, extensive experimental evaluations, and the effective combination of theoretical and empirical analyses. Below, we provide detailed responses to each of your questions, offering clarification and additional support.
>
> ---
>
> **Q1: To extend theoretical understandings from the simple single-item CF scenario to practical multi-item scenarios, Corollary 1 is restricted to the dot-product-based loss function.**
>
> **A1**: Thank you for your question. The core of our recommendation scenario centers on the dot-product interaction between users and items. While Corollary 1 focuses on this foundational setup, exploring general formulations rather than specific modeling approaches can enhance theoretical applicability.
> - To assess the robustness of Corollary 1 beyond dot-product scenarios, we will include experimental results using NeurMF with non-dot-product-based loss functions.
> - We have found that PamaCF **achieves state-of-the-art performance, even in non-dot-product-based loss functions**.
>
> - **Recommendation Performance**: The table can be seen in the pdf (Table 3) in the global response.
> - **Robustness against target items promotion**: The table can be seen in the pdf (Table 4) in the global response.
>
> ---
>
> **Q2: The experiment only involves two CF backbones, namely MF and LightGCN. The results will be more convincing by adding more CF methods.**
>
> **A2**: Thank you for your suggestion. We appreciate your feedback, and in response, we have expanded our experimental evaluation to include results from NeurMF, as shown in the tables referenced in Q1. We have found that **Our findings demonstrate that PamaCF achieves state-of-the-art performance within the NeurMF**. This addition broadens the scope of collaborative filtering methods examined in our study.
>
> ---
>
> We sincerely appreciate your time, effort, and valuable suggestions during the review process. We trust that these clarifications adequately address your concerns.

---

> > ### Comment · Reviewer_MDNE · 2024-08-13
> >
> > Thanks for the authors' explanations. I have no further questions.

---

> > > ### Author Response · Authors · 2024-08-13
> > >
> > > We are grateful that our response successfully resolved your concerns. Thank you for your feedback and for positively evaluating our work.
> > >
> > > Sincerely,
> > > The Authors

---

> ### Author Response · Authors · 2024-08-12
> **Follow up**
>
> Thank you for taking the time and effort to review our paper. We have carefully considered your comments and responded to each point with detailed clarifications and supporting evidence. As the deadline for the discussion period between authors and reviewers is approaching, we kindly ask whether our responses have sufficiently addressed your concerns. Please let us know if there are any further questions or issues that you would like us to address. We look forward to hearing from you soon.
>
> Thank you once again for your time and effort.
>
> Sincerely,
> The Authors

---

### Official Review · Reviewer_6kCb · 2024-07-10

**Soundness:** 2
**Presentation:** 3
**Contribution:** 2
**Rating:** 5
**Confidence:** 4

**Summary:**

This work investigates adversarial collaborative filtering, providing theoretical evidence for the effectiveness of such methods and proposing a novel method based on the personalized magnitude of perturbation.

Overall, this work studies on an interesting problem and offer some theoretical insights. However, this work also has some limitations particularly on the impractical assumptions. As such, I give the score of 5.

**Strengths:**

1.	This work studies on an interesting and important problem.

2.	This work provides a theoretical understanding of the effectiveness of the adversarial collaborative filtering, and introduces a new method based on this understanding.

3.  Extensive experiments are conducted to validate the effectiveness of the proposal.

**Weaknesses:**

1.	My primary concern lies with the assumptions made in the theoretical analysis. There is a noticeable gap between the theoretical analysis and practical applications in several aspects: a) the strategy for initializing embeddings (presented in Definition 1) is seldom adopted in existing RS; b) the optimization in practical RS typically uses BCE, BPR, or softmax loss, rather than the loss function described in equation (2); c) preference prediction usually does not employ the sign function; d) recommendation is often framed as a ranking problem with corresponding metrics or losses, rather than being defined as in ‘Definition 2’ for analyzing recommendation error.

I understand these impractical assumptions could facilitate theoretical derivation, but they clearly become a limitation of this work.

2.	My other concern is with the theoretical analysis itself.

2.1	 It is counter-intuitive that the theoretical bound does not depend on the item embeddings, given the symmetric roles of user and item embeddings in recommendation systems. Could you please provide some explanations? Or the relations $v=1/n \sum_i r_i u_i$ are always held?

2.2	It is interesting to discover that CF has a unique advantage such that the adversarial mechanism works even for clean data, unlike in basic classification tasks. Why is collaborative filtering so special? The problem formulation for collaborative filtering appears similar to basic classification tasks. Could the distinction be due to the inner product? More discussion on which characteristics of collaborative filtering lead to this advantageous property would be highly interesting.

Some minor ones:

3.	The theoretical part is not easy to follow.  This may be due to my unfamiliarity with theories on adversarial learning. Nevertheless, I recommend that the authors provide intuitive explanations for the theorems and a brief outline of the proof procedure.

4.	The paper should include more experimental details. It is unclear whether the experimental setup (e.g., initialization) follows the problem definition 1 or aligns with previous works like [2][3].

**Questions:**

Please refer to weaknesses.

**Limitations:**

the authors have discussed the limitations.

---

> ### Author Rebuttal · Authors · 2024-08-06
>
> We sincerely appreciate your positive comments and recognition of the significance of our paper, including its theoretical contributions, novel methodology, and comprehensive experiments. In response to your feedback, we address your concerns and clarify some misunderstandings as follows:
>
> ---
>
> **Q1: The strategy for initializing embeddings is seldom adopted in existing RS**.
>
> **A1**: Thanks for your feedback. The Gaussian-based initialization method is chosen to derive upper and lower bounds and **does not alter our core findings**: (1) ACF outperforms traditional CF; (2) User-adaptive magnitude further enhances the effectiveness of ACF.
> - For instance, the conclusion in Theorem 1 is already demonstrated in Appendix D1.1 at line 667, but incorporating the properties of Gaussian distribution allows for a more accurate formulation.
> - In Appendix D2.1 at line 706, the expression already indicates the need for different magnitudes; introducing Gaussian distribution properties results in more accurate and readable upper and lower bounds.
>
> ---
>
>
> **Q2: The optimization in practical RS typically uses BCE, BPR, or softmax loss, rather than the loss function described in Eq. 2.**
>
> **A2**: Section 3's simplified pointwise loss ensures derivation clarity and simplicity, as in [1].
> - To broaden the applicability of our findings, Corollary 1 in Section 4 **extends this to complex losses** (Appendix D.3).
> - Moreover, in the implementation of PamaCF (Eq. 6 in the main text), we adopt the extended and widely used pairwise loss, i.e., PamaCF-BPR loss, detailed in Appendix B.
>
> ---
>
> **Q3: Preference prediction usually does not employ the sign function.**
>
> **A3**: The sign function aids recommendation error definition **without affecting our derivations**. In Appendices D.1, D.2, and D.3, we demonstrate that $\mathbb{P}[\mathrm{sgn}(\langle u, v \rangle) \neq r] = \mathbb{P}[r \cdot \langle u, v \rangle < 0]$, effectively addressing any potential impact of the sign function on our results.
>
> ---
>
> **Q4: Recommendation is often framed as a ranking problem, rather than being defined as in 'Definition 2' for analyzing recommendation error.**
>
> **A4**: Thanks for your question. Our approach in Section 3 simplifies analysis by focusing on item impact via a pointwise framework. This approach provides theoretical clarity by avoiding the complexities introduced by ranking tasks. Section 4's Corollary 1 **extends these insights beyond the constraints of Definitions 1-3**, as detailed in Appendix D.3.
>
> ---
>
> **Q5: It is counter-intuitive that the theoretical bound does not depend on the item embeddings.**
>
> **A5**: **Theoretical bounds indeed depends on the item embeddings**.
> - When fixing a particular user, the item embeddings $v_{(0)}$ is equivalent to sampling from $\mathcal{N}(\bar{u}, \frac{\sigma^2}{n-1}I)$ (as mentioned in lines 120-123).
> - When deriving upper and lower bounds, we map $v_{(t)}$ to $v_{(0)}$ and then take expectations, which is why $v_{(t)}$ does not explicitly appear in the expressions. For details, refer to lines 710 or 733 in Appendix D.2.
>
> ---
>
> **Q6: Are the relations $v = \frac{1}{n} \sum_i r_i u_i$ always held?**
>
> **A6**: No, this relation holds at epoch = 0. However, with a given learning rate $\eta$, our derivation shows that $v_{(t)} = M(t, \eta) \frac{1}{n} \sum r u_{(0)}$, where $M(t, \eta)$ is a mapping function detailed in Proposition 1 (lines 625-630).
>
> ---
>
> **Q7: It is interesting to discover that CF has a unique advantage such that the adversarial mechanism works even for clean data, unlike in basic classification tasks. Why is collaborative filtering so special?**
>
> **A7**: Thanks for acknowledging our work. The key distinction may lie in the nature of the tasks, particularly in the parameter search space. Consider the following simplification (one-layer classifier and MF):
>
> - Traditional classification tasks are typically formulated as follows:
>   - Given a set of samples $(x_i, y_i)_{i=0}^n$, where $x \in \mathbb{R}^{d_1}$ and $y \in \mathbb{R}^{d_2}$, the goal is to train a classifier $f(x_i) = w^Tx_i + b$.
>   - The trainable component is the classifier $w \in \mathbb{R}^{d_1 \times d_2}$, and adversarial perturbations compel $w$ to locate a decision boundary within $\mathbb{R}^{d_1 \times d_2}$ that satisfies adversarial losses.
>
> - For recommender systems, where $f(u_i, v_i) = u_i^T v_i$, involving $M$ users and $N$ items:
>   - The application of adversarial perturbations allows for a broader parameter search space, specifically $MN \times \mathbb{R}^d \times \mathbb{R}^d$.
>
> Traditional classification tasks **require $w$ to satisfy all instances and their adversarial counterparts**, whereas in recommendation tasks, **adjustments to user and item representations can satisfy each other's adversarial counterparts**.
>
> ---
>
> **Q8: I recommend that the authors provide intuitive explanations for the theorems and a brief outline of the proof procedure.**
>
> **A8**: Thank you for your suggestion. We will enhance the clarity by providing intuitive explanations and outlining the proofs. Due to space constraints on the responses, we will further supplement these in subsequent versions.
>
> ---
>
> **Q9: The paper should include more experimental details.**
>
> **A9**: Thanks for your suggestion. To address the need for additional experimental details, we have included them in Appendix C.1:
> - Detailed discussions on attack and defense setups are provided from lines 589 to 600.
> - As you mentioned, our backbone model follows LightGCN. We will emphasize this aspect more prominently in future revisions.
>
>
> ---
>
> We appreciate your thorough feedback and recognition of the significance of our research. We believe these clarifications adequately address your concerns and look forward to your reconsideration and reevaluation of our work.
>
>
> [1] Fight Fire with Fire: Towards Robust Recommender Systems via Adversarial Poisoning Training. SIGIR'21

---

> > ### Comment · Reviewer_6kCb · 2024-08-12
> >
> > Thank you for the detailed response. My concerns have not been well addressed, particular in the gap between the theoretical analyses and practical applications. The initializing, the prediction functions and the model architectures may not be closely alignment with the pratical. Nevertheless, I acknowledge the theoretical contribution of this work and would like keep the positive score of this work.

---

> > > ### Author Response · Authors · 2024-08-12
> > > **Further Clarification on Initialization, Prediction Functions, and Model Architectures**
> > >
> > > Thank you for your comments and for acknowledging the theoretical contributions of our work.
> > >
> > > We would like to clarify that the initialization (Q1) and prediction functions (Q3) are employed solely to simplify the formulas and aid in defining certain concepts. **These choices do not impact the theoretical results presented in our study**. Furthermore, we use the **widely adopted matrix factorization (MF) model for theoretical derivation** [1][2], which is consistent with existing mainstream architecture.
> > >
> > > ---
> > >
> > > ## Gaussian Initialization (Q1,A1)
> > >
> > > Regarding Gaussian initialization (Q1), the Gaussian initialization method is chosen to derive more precise and clearer bounds. **Even without introducing Gaussian initialization, we still achieve the following results:**
> > >
> > > - ACF outperforms traditional CF.
> > > - User-adaptive magnitude further enhances the effectiveness of ACF.
> > >
> > > For instance, as detailed in Appendix D1.1, line 667, we have shown:
> > >
> > > $\mathbb{P}_{{v}_0 \sim \mathcal{N}(\bar{{u}}, \frac{\sigma^2}{n-1}I)}  \left[f({u}, {v}) \neq r \mid ({u}, r) \right]$ -
> > >
> > > $~~~~~~\mathbb{P}_{{v}_0 \sim \mathcal{N}(\bar{{u}}, \frac{\sigma^2}{n-1}I)}^{~\mathrm{adv}}  \left[f({u}, {v}) \neq r \mid ({u}, r) \right]$
> > >
> > > $~~~~= \mathbb{P}_{{v}_0 \sim \mathcal{N}(\bar{{u}}, \frac{\sigma^2}{n-1}I)}\left[-\eta (1+\lambda) \gamma_t^{{u}} M(t, \eta) \frac{\Vert {v}_0\Vert^2}{\Vert {u}_t\Vert} < \langle \frac{r{u}_t}{\Vert {u}_t \Vert}, {v}_0 \rangle \leq -\eta M(t, \eta) \frac{\Vert {v}_0\Vert^2}{\Vert {u}_t\Vert} \right] \ge 0,$
> > >
> > > This result already supports Theorem 1. By leveraging the properties of the Gaussian distribution, we are able to derive a more precise and clearer formula:
> > >
> > > $\mathbb{P}_{{v}_0 \sim \mathcal{N}(\bar{{u}}, \frac{\sigma^2}{n-1}I)}\left[-\eta (1+\lambda) \gamma_t^{{u}} M(t, \eta) \frac{\Vert {v}_0\Vert^2}{\Vert {u}_t\Vert} < \langle \frac{r{u}_t}{\Vert {u}_t \Vert}, {v}_0 \rangle \leq -\eta M(t, \eta) \frac{\Vert {v}_0\Vert^2}{\Vert {u}_t\Vert} \right]$
> > >
> > > $~~~~ = \Phi\left( \frac{\sqrt{n-1}}{\sigma}\left( \eta (1+\lambda) \gamma_t^{{u}} M(t, \eta) \frac{\Vert \bar{{u}} \Vert^2 + \frac{d \sigma^2}{n-1}}{\Vert {u}_t\Vert} + \langle \frac{r{u}_t}{\Vert {u}_t \Vert}, \bar{{u}} \rangle \right)\right)$
> > >
> > > $~~~~~~ - \Phi \left( \frac{\sqrt{n-1}}{\sigma} \left(\eta M(t, \eta) \frac{\Vert \bar{{u}} \Vert^2 + \frac{d \sigma^2}{n-1}}{\Vert {u}_t\Vert} + \langle \frac{r{u}_t}{\Vert {u}_t \Vert}, \bar{{u}} \rangle\right) \right) > 0.$
> > >
> > > This result also applies to Theorems 2-4, as discussed in A1 of our rebuttal.
> > >
> > > ---
> > >
> > > ## Prediction Functions (Q3,A3)
> > >
> > > Regarding the prediction functions (Q3), instead of the sign function, we could have used another prediction function. **Even without the sign function, we can still derive the subsequent results**:
> > >
> > > - ACF outperforms traditional CF.
> > > - User-adaptive magnitude further enhances the effectiveness of ACF.
> > >
> > > We only utilized the sign function specifically to define the recommendation error, which aligns with existing publications [3][4]. This choice allows us to express the recommendation error as:
> > >
> > > $\mathbb{P}_{{v}_0 \sim \mathcal{N}(\bar{{u}}, \frac{\sigma^2}{n-1}I)}\left[\mathrm{sgn}(\langle{u}_t, {v}_t)\rangle \neq r \mid ({u}, r)\right].$
> > >
> > > $~~~~= \mathbb{P}_{{v}_0 \sim \mathcal{N}(\bar{{u}}, \frac{\sigma^2}{n-1}I)} \left[\langle{u}_t, {v}_t\rangle \cdot r < 0 \mid ({u}, r) \right],$
> > >
> > > as we elaborated in A3 of our rebuttal.
> > >
> > > ---
> > >
> > > ## Model Architectures
> > >
> > > In both Theorems in Section 3 and the Corollary in Section 4, we adopt MF as the backbone model, which is consistent with existing mainstream architecture [1][2].
> > >
> > > ---
> > >
> > > It seems there may be some misunderstanding that has led to the perception of a gap between our theoretical analyses and practical applications. We have provided further explanations on the three points you mentioned, and we hope you will reconsider the contributions of our work. Thank you very much!
> > >
> > >
> > > [1] Adversarial Personalized Ranking for Recommendation. SIGIR'18
> > > [2] Adversarial Collaborative Filtering for Free. RecSys'23
> > > [3] Adversarially robust generalization requires more data. NeurIPS'21
> > > [4] Fight Fire with Fire: Towards Robust Recommender Systems via Adversarial Poisoning Training. SIGIR'21

---

> ### Author Response · Authors · 2024-08-12
> **Follow up**
>
> Thank you for taking the time and effort to review our paper. We have carefully considered your comments and responded to each point with detailed clarifications and supporting evidence. As the deadline for the discussion period between authors and reviewers is approaching, we kindly ask whether our responses have sufficiently addressed your concerns. Please let us know if there are any further questions or issues that you would like us to address. We look forward to hearing from you soon.
>
> Thank you once again for your time and effort.
>
> Sincerely,
> The Authors

---

### Official Review · Reviewer_svg8 · 2024-07-17

**Soundness:** 2
**Presentation:** 4
**Contribution:** 3
**Rating:** 5
**Confidence:** 4

**Summary:**

This paper targets adversarial collaborative filtering and provides both deeper understanding and improvement. The paper provides a theoretical explanation for the ACF's improvement upon robustness and effectiveness. PamaCF, which is presented in Appendix, can more robustly performs CF, is evaluated on three recommendation datasets.

**Strengths:**

1. The paper is generally well-written and easy to follow.
2. The paper provides a theoretical explanation for the ACF's improvement in robustness and effectiveness.

**Weaknesses:**

1. Part of the experiments are missing and incomplete.
2. The loss terms in Section 3 do not fully align with the cited papers.

**Questions:**

1. What's the performance of various ACF methods on LightGCN? It would be more comprehensive to report another Table similar to Table 1 about LightGCN.
2. Why does the paper present the recommendation performance at HR@50 and NDCG@50 in Table 1 and present the performance against attacks in Table 2 with T-HR@50 and T-NDCG@50?
3. Is target item promotions the only attack PamaCF can prevent?
4. The loss of Gaussian Recsys in Eq 2 does not fully align with [18]. So is the same regarding Eq 3 and [15]. Why does the author make such changes? The loss term is the foundation of all derivity. All previous works are based upon pairwise loss while the paper is based on pointwise loss.

**Limitations:**

Although the paper presents an important theoretical and experimental analysis of ACF, the reviewer feels a strong sense of misalignment between Section 3 and 4, where the theoretical results in Section 3 do not fully motivate PamaCF in Section 4. The result of Section 3 seems to be motivated by pointwise loss, while PamaCF is built upon pairwise loss. Also, part of the experimental setup is not convincing enough. However, the reviewer might be persuaded with further evidence.

---

> ### Author Rebuttal · Authors · 2024-08-06
>
> Thanks for your detailed feedback and for acknowledging the readability and theoretical clarity of our paper. Below, we address each of your questions to clarify misunderstandings and address concerns. We hope this clarifies the contributions of our work and look forward your reconsideration.
>
> ---
>
> **W1: Part of the experiments are missing and incomplete.**
>
> **W-A1**: Due to space constraints of main text, we focused on presenting results that strongly support our claims. Other reviewers also praised the thoroughness and persuasiveness of our experiments. To address your concerns comprehensively, detailed explanations and additional support will be provided in A1-A3, corresponding to Q1-Q3.
>
> ---
>
> **Q1: What is the performance of different ACF methods on LightGCN?**
>
> **A1**: Thanks for your suggestion. In response, experimental results for LightGCN will be included. Our findings show improved recommendation performance for PamaCF integrated with LightGCN. **Refer to Table 1 in the PDF in global response for details.**
>
>
>
> ---
>
> **Q2: Why does the paper use HR@50 and NDCG@50 in Table 1 to present recommendation performance, while Table 2 uses T-HR@50 and T-NDCG@50 to evaluate performance against attacks?**
>
> **A2:** Thanks for your question. In Table 1, we **actually used HR@20 and NDCG@20**, not HR@50 and NDCG@50. Your query addresses two points: (1) the choice of metrics and (2) the difference in k values. Here are our responses:
>
> 1. HR@k and NDCG@k evaluate recommendation performance, while T-HR@k and T-NDCG@k assess system robustness against attacks. We provide detailed explanations in the **Evaluation Metrics section** (lines 256-269).
> 2. Our selection of k values follows standard practices in the field, detailed in the **Implementation Details section** (lines 589-592). This approach ensures consistency with research that distinguishes between metrics for recommendation and defense strategies [1], underscoring the reliability of our defense methods.
>    - HR@20 and NDCG@20 focus on the top 20 recommendations, typical in collaborative filtering assessments [2].
>    - T-HR@50 and T-NDCG@50 evaluate defense effectiveness against attacks, commonly using a top 50 ranking [3][4].
>
>
> ---
>
>
> **Q3: Is preventing target item promotions the only attack PamaCF can mitigate?**
>
> **A3**: **PamaCF is capable of defending against various poisoning attacks**. These attacks typically aim to promote specific items or degrade system performance. Item promotion attacks are well-documented [3][4], while performance degradation attacks involve harmful item embeddings mainly in federated recommender systems [5].
>
> - Theoretical analysis (Section 3) of ACF considers the impact of poisoning attacks on recommendation errors, without restricting attack objectives. **Theoretically, PamaCF can mitigate poisoning attacks aimed at different objectives.**
> - Empirical validation shows PamaCF not only counters item promotion attacks but also enhances overall recommendation performance.
>   - To further validate PamaCF's robustness, we simulate performance degradation attacks with random user behaviors, confirming its effectiveness.
>   - PamaCF demonstrates strong defense capabilities even against performance degradation attacks.
>   - **Refer to Table 2 in the PDF in global response for details.**
>
>
> ---
>
>
> **Q4 (W2): The loss function of Gaussian Recsys in Eq. 2 does not fully correspond to [6]. So is the same regarding Eq. 3 and [7]**
>
> **A4**: **The loss function used in Gaussian Recsys does correspond to [6]**. [6] covers various loss functions, addressing any confusion about the specific one referenced. We refer to the loss function detailed in [6]'s "DETAILED PROOFS" section. The mention of "introducing the adversarial loss [7]" (line 112) in our paper refers to adopting the adversarial training paradigm (as described in Eq. 1 in the Preliminary section), where **adversarial perturbations are added to the original loss (Eq. 2), rather than using the APR loss function from [7]**.
>
> - For instance, the adversarial component discussed in [6] involves $\Delta_{v} = \arg\min_{\Delta, \Vert \Delta \Vert \le \epsilon} \langle ru, v + \Delta_{v} \rangle$, where the original loss is $\mathcal{L}(\Theta+\Delta) = \langle r(u+ \Delta_{u}), v + \Delta_{v} \rangle$. This aligns precisely with our employed loss function in Section 3.
>
>
> ---
>
> **Q5: All previous works are based on pairwise loss, whereas the paper is based on pointwise loss.**
>
> **A5**: It's worth noting that many methods also adopt pointwise loss [8][9]. Besides, in Section 3, our theoretical results leverage pointwise loss for clarity and simplicity.  Section 4's Corollary 1 **extends the discussed pointwise loss to encompass more complex functions**, detailed in Appendix D.3.
>
> ---
>
> Thanks sincerely for your thorough review of our paper. We have addressed each of your concerns and clarified misunderstandings. Based on these clarifications, we kindly request your reconsideration of the overall assessment of this paper. We firmly believe that our work contributes valuable insights and advances the state-of-the-art in robust recommender systems, making a significant contribution to the research community.
>
>
> [1] LoRec: Combating Poisons with Large Language Model for Robust Sequential Recommendation. SIGIR'24
> [2] LightGCN - Simplifying and Powering Graph Convolution Network for Recommendation. SIGIR'20
> [3] Revisiting Injective Attacks on Recommender Systems. NeurIPS'22
> [4] Revisiting Adversarially Learned Injection Attacks Against Recommender Systems. RecSys'20
> [5] UA-FedRec: untargeted attack on federated news recommendation. SIGKDD'23
> [6] Fight Fire with Fire: Towards Robust Recommender Systems via Adversarial Poisoning Training. SIGIR'21
> [7] Adversarial Personalized Ranking for Recommendation. SIGIR'18
> [8] Denoising Implicit Feedback for Recommendation. WSDM'21
> [9] BSL: Understanding and improving softmax loss for recommendation. ICDE'24

---

> ### Author Response · Authors · 2024-08-12
> **Follow up**
>
> Thank you for taking the time and effort to review our paper. We have carefully considered your comments and responded to each point with detailed clarifications and supporting evidence. As the deadline for the discussion period between authors and reviewers is approaching, we kindly ask whether our responses have sufficiently addressed your concerns. Please let us know if there are any further questions or issues that you would like us to address. We look forward to hearing from you soon.
>
> Thank you once again for your time and effort.
>
> Sincerely,
> The Authors

---

> ### Author Response · Authors · 2024-08-13
> **Kindly Reminder**
>
> Thank you once again for dedicating your time and effort to reviewing our paper. We are following up on our message sent yesterday regarding the discussion period between authors and reviewers, which concludes on August 13.
>
> We understand that you may have a busy schedule and truly appreciate the time and effort you have already invested in reviewing our work. We have carefully considered your comments and provided detailed clarifications and support for each concern in our rebuttal.
>
> Specifically:
>
> - **For W1:** Other reviewers (gsZx, 6kCb, MDNE) also praised the thoroughness and persuasiveness of our experiments. We understand your concerns and have provided detailed explanations and additional support in A1-A3, corresponding to Q1-Q3.
> - **For W2:** We clarified that the loss terms in Section 3 indeed align with the cited papers, as discussed in A4 corresponding to Q4.
>
> We are eager to learn whether our response has sufficiently addressed your concerns and if there are any further questions or suggestions. Please accept our apologies for reaching out again so soon; however, we are committed to meeting the deadline and ensuring the timely progression of our paper through the discussion process. If you require any additional information or clarification, please do not hesitate to reach out.
>
> Once again, we express our gratitude for your feedback and your continued assistance in this process.
>
> Sincerely,
> The Authors

---

> > ### Comment · Reviewer_svg8 · 2024-08-13
> >
> > Thanks to the author for the rebuttal. However, the reviewer is still not convinced this paper is meaningful from the perspective of collaborative filtering. The evaluation is not solid based on the recommendation review perspective. At least two k should be adopted for HR@k and NDCG@k to validate that this method is not biased towards specific metrics. Also, adopting different losses between baseline and proposed methods is not convincing enough to demonstrate PamaCF's effectiveness for recommendation tasks. Besides, some of the responses are still not clear to the reviewer.
> >
> > Hence, the reviewer will keep the score. If other reviewers and AC strongly think this paper is acceptable, the reviewer respect their decision.

---

> > > ### Author Response · Authors · 2024-08-13
> > > **Further Clarification**
> > >
> > > Thank you for your comments. We appreciate your feedback. However, there are still some misunderstandings in your replies.
> > >
> > > ---
> > >
> > > ### Regarding the Loss Function
> > >
> > > Your understanding of our loss function in the experiments is incorrect. **We indeed use the same base loss function (BPR loss) across all the baselines.** Please refer to our code in the Supplementary Material (line 223 in `./utls/trainer.py`), where all the methods share a unified loss function.
> > >
> > > ---
> > >
> > > ### Regarding the Selection of $k$ in Our Evaluation
> > >
> > > Our experimental setup involves:
> > >
> > > - Two types of backbone models
> > > - Four types of attacks
> > > - Five defense baselines
> > > - Four evaluation metrics
> > > - Three datasets
> > >
> > > For simplicity, clarity of experimental results, and due to space limitations, we only presented results with the most common top $k = 20$ choices, as seen in existing collaborative filtering works [1-5]. We also evaluated different $k$ values. Below are partial results for $k = 10$. You can also see from the code in the Supplementary Material (line 41 in `./meta_config.py`) that we set up multiple $k$ values.
> > >
> > > |   Bandwagon Atk. Gowalla   | HR@10                    | NDCG@10                    |
> > > |------------------------|----------------------------|----------------------------|
> > > | **MF** (Clean)                 | 7.494 ± 0.080              | 5.846 ± 0.035              |
> > > | **MF**                   | 7.410 ± 0.055              | 5.812 ± 0.044              |
> > > | +**StDenoise**         | 6.954 ± 0.086              | 7.132 ± 0.029              |
> > > | +**GraphRfi**          | 6.860 ± 0.042              | 6.910 ± 0.060              |
> > > | +**APR**               | _9.204 ± 0.046_            | _9.564 ± 0.027_            |
> > > | +**SharpCF**           | 8.705 ± 0.076              | 8.723 ± 0.058              |
> > > | +**PamaCF**            | **9.492 ± 0.021**          | **9.838 ± 0.013**          |
> > > | **Gain**               | +3.13%                     | +2.86%                     |
> > > | **Gain w.r.t. MF**     | +28.10%                    | +69.27%                    |
> > >
> > > **We find that our method achieves consistent optimality even with different $k$.**
> > >
> > > We believe this selection is reasonable. We are willing to provide results for multiple $k$ values in the final version. We believe that the results under this different $k$, as well as the experiments in multiple scenarios, are sufficient to demonstrate the superiority of the methodology.
> > >
> > > [1] Neural graph collaborative filtering. SIGIR'19
> > > [2] LightGCN - Simplifying and Powering Graph Convolution Network for Recommendation. SIGIR'20
> > > [3] Empowering collaborative filtering with principled adversarial contrastive loss. NeurIPS'23
> > > [4] Invariant collaborative filtering to popularity distribution shift. WebConf'23
> > > [5] Distributionally Robust Graph-based Recommendation System. WebConf'24
> > >
> > > ---
> > >
> > > ### Contribution of Our Work on Collaborative Filtering
> > >
> > > Adversarial training has been empirically demonstrated to improve both model robustness and recommendation performance. In our work, we have **explained its effectiveness in different scenarios from a theoretical point of view** and **provided ways to further enhance the effectiveness of adversarial training.** We believe that our work is valuable and meaningful for the collaborative filtering field, as also noted by the other three reviewers.
> > >
> > > ---
> > >
> > > For any of the previous replies, if there is anything else that you feel is unclear to you, please let us know. We would be happy to discuss further.
> > >
> > > ---
> > >
> > > Based on these clarifications, we kindly request you to reconsider the overall assessment of our paper. We firmly believe that our work contributes valuable insights and advances the state-of-the-art in robust recommender systems, making a significant contribution to the research community.
> > >
> > > Sincerely,
> > > The Authors

---

> > > > ### Comment · Reviewer_svg8 · 2024-08-13
> > > >
> > > > Thanks to the author for consistently providing so many new information. The reviewer still thinks the paper should be improved upon evaluation in the very initial version. Frankly speaking, the reviewer has not seen any experiment added during the rebuttal period does not align with the previous conclusion or the reviewer's expectation. So from a personal perspectives, they does not help me any bit when reevaluting the paper.
> > > >
> > > > However, given most of my misunderaranding has been addressed and the authors actively report different @k metrics, the reviewer will raise the score. But the reviewer strongly encourage the author to redo the experiments by reporting different @k metrics, add the reported tables and make the paper consistent between theory and experiment.

---

> > > > > ### Author Response · Authors · 2024-08-14
> > > > >
> > > > > Thank you very much for taking the time to review our response and for your willingness to increase the score of our paper. We are delighted to hear that our response has addressed your previous misunderstandings.
> > > > >
> > > > > We would like to clarify that the tables reported in our paper consistently align theory with experiment, a view that has also been recognized by the other three reviewers. We understand your concerns. We have indeed conducted experiments with different @k metrics and have saved the corresponding results. For simplicity, clarity, and due to space limitations, we only presented results with the most common choice of top $k = 20$. We will include the additional experimental results provided during the rebuttal period in the Appendix of subsequent versions.
> > > > >
> > > > > Thank you once again for your time and effort in reviewing our paper.
> > > > >
> > > > > Sincerely,
> > > > > The Authors

---

### Official Review · Reviewer_gsZx · 2024-07-17

**Soundness:** 4
**Presentation:** 3
**Contribution:** 3
**Rating:** 6
**Confidence:** 3

**Summary:**

Adversarial training has been observed to degrade model performance on clean samples in the CV domain, however, ACF in recommender systems can not only enhance the robustness against poisoning attacks but also improve recommendation performance. This paper provides a comprehensive theoretical understanding of this phenomenon. Specifically, this paper shows the performance with adversarial training is better than without adversarial training in both clean and $\alpha$-poisoned scenarios. In addition, this paper provides the lower and upper bounds for the recommendation error. This paper also proposes a learning approach that can be used in more practice scenarios. Experiments on three datasets demonstrate the effectiveness of the proposed method in terms of robustness and recommendation performance.

**Strengths:**

$\bullet~$ The problem studied is important and relevant.

 $\bullet~$ The idea is interesting and novel.

$\bullet~$ The motivation is reasonable.

 $\bullet~$ The theoretical results are sound and are verified in Figure 1.

 $\bullet~$ The evaluations are solid and convincing.

**Weaknesses:**

$\bullet~$ Can authors discuss the feasibility of relaxing the assumption in theorems such as the Gaussian recommendation system? In addition, is the the extension to multiple items natural?

$\bullet~$ It will be better if the magnitude of user latent vectors is provided in Figure 1. Thus we can align the empirical findings and theoretical findings more straightforwardly. Maybe we can observe that the magnitude of the latent vector of user 3 is larger than that of user 2.

 $\bullet~$ Is there a more reasonable way to design the form of $c(\mathbf{u}, t)$ and equation 5? In addition, how can we choose $\rho$ in practice? Is $\rho$ learnable or it is only a hyper-parameter?

 $\bullet~$ Some presentation issues. For example, the $\alpha$ in line 184 should be bolded, and it should be $\Phi(\cdot)$ in line 176. In addition, the results in Table 1 should retain 2 digits instead of 3 digits to enhance the readability. The text in Figures 2 and 3 (such as x-axis and y-axis) should be larger.

**Questions:**

See the weaknesses part for the questions.

**Limitations:**

Yes, this author adequately discusses the limitation and broader impact.

---

> ### Author Rebuttal · Authors · 2024-08-06
>
> Thanks for your positive feedback. We sincerely appreciate your recognition of the importance, novelty, reasonable motivation, sound theoretical results, and the solid evaluations presented in our work. Below, we address each of your questions to resolve your concerns:
>
> ---
>
> **Q1: Can authors discuss the feasibility of relaxing the assumption in theorems such as the Gaussian recommender system? In addition, is the extension to multiple items natural?**
>
> **A1**: We appreciate your question. The assumption in the Gaussian recommender system serves primarily for clarity and simplicity of theoretical results, as also adapted in [1]. To relax this assumption, one approach involves considering more complex loss functions or scenarios involving interactions with multiple items.
> - **We elaborate on these scenarios in Corollary 1**, with detailed derivations available in Appendix D.3.
> - Specifically, for any dot-product-based loss function involving multiple items, we ascertain the range of perturbation magnitude for each user.
> - While the formulations in Appendix D.3 are complex, they are crucial for identifying the relationship between perturbation magnitudes and user embeddings.
>
> ---
>
> **Q2: It will be better if the magnitude of user latent vectors is provided in Figure 1. Thus we can align the empirical findings and theoretical findings more straightforwardly.**
>
> **A2**: Thank you for this excellent suggestion. We will incorporate the magnitude of user latent vectors into Figure 1. This addition will enhance the alignment between empirical findings and theoretical insights. The table below presents $\Vert u \Vert^2$ for the five users in Figure 1, which will be included in the revised paper. **The revised image can be seen in the pdf (Figure 1) in the global response**.
>
> |UserID|user 1|user 2|user 3|user 4|user 5|
> |--|------|------|------|------|------|
> |$\Vert u \Vert^2$|3.7333|2.0926|11.895|6.7923|6.8239|
>
> ---
>
> **Q3: How can we choose $\rho$ in practice? Is $\rho$ learnable or is it only a hyper-parameter?**
>
> **A3**: $\rho$ is considered a hyper-parameter used to adjust the overall magnitude in practical applications. In Experiment 5.4, we extensively studied the impact of $\rho$ on method performance, revealing that:
> - Even with a small $\rho$, noticeable improvements are observed.
> - Optimal values for $\rho$ typically fall within the range of 0.1 to 1.0 across diverse datasets. In practice, we typically search for $\rho$ within this range at intervals of 0.1.
>
> ---
>
> **Q4: Is there a more reasonable way to design the form of $c(u,t)$ and equation 5?**
>
> **A4**: Our paper introduces a specific design pattern for $c(u,t)$ that aligns with the criteria outlined in Corollary 1. The term $\rho \cdot c(u,t)$ allows for flexible adjustment of overall magnitude. Moving forward, we aim to explore more intricate design patterns aimed at better leveraging the properties identified in Corollary 1. For instance, we will investigate alternative mapping functions to replace the sigmoid function, which may better capture the relationships outlined in Corollary 1. These efforts are aimed at enhancing the effectiveness and adaptability of our approach in practical applications.
>
> ---
>
> **Q5: Some presentation issues.**
>
> **A5**: Thank you for your valuable suggestions aimed at enhancing the presentation of our paper. We will take the following actions to address these concerns:
> - Ensure accurate representation by verifying the correct usage of bold symbols.
> - Improve readability by reducing the number of digits in the tables.
> - Enhance visibility by increasing the text size in Figures 2 and 3.
>
> ---
>
> We appreciate your detailed comments and constructive feedback, as well as your recognition of the importance of our work. We believe these clarifications will effectively address your concerns. We eagerly await your reconsideration and reevaluation of our work.
>
>
> [1] Fight Fire with Fire: Towards Robust Recommender Systems via Adversarial Poisoning Training. SIGIR'21

---

> ### Author Response · Authors · 2024-08-12
> **Follow up**
>
> Thank you for taking the time and effort to review our paper. We have carefully considered your comments and responded to each point with detailed clarifications and supporting evidence. As the deadline for the discussion period between authors and reviewers is approaching, we kindly ask whether our responses have sufficiently addressed your concerns. Please let us know if there are any further questions or issues that you would like us to address. We look forward to hearing from you soon.
>
> Thank you once again for your time and effort.
>
> Sincerely,
> The Authors

---

> > ### Comment · Reviewer_gsZx · 2024-08-13
> >
> > Thanks for the detailed rebuttal, which addresses my concerns. I will keep my rating unchanged.

---

> > > ### Author Response · Authors · 2024-08-13
> > >
> > > We are delighted to hear that our response effectively addressed your concerns. We sincerely appreciate your support and the positive evaluation of our work.
> > >
> > > Sincerely,
> > > The Authors

---

### Author Rebuttal · Authors · 2024-08-06

We would like to express our sincere gratitude to all reviewers for your valuable feedback and for taking the time to evaluate our work. We are very encouraged that our main contributions have been acknowledged by all reviewers. We have taken great care to address any concerns or misunderstandings raised by the reviewers by providing detailed clarifications and additional support. We believe that our efforts have been worthwhile and are looking forward to your continued attention to our paper.

To facilitate the assessment of our contributions, let us summarize the strengths recognized by reviewers:

- Important research problem (Reviewers gsZx, 6kCb, MDNE)
- Interesting and novel idea (Reviewers gsZx, MDNE)
- Reasonable motivation (Reviewers gsZx, MDNE)
- Sound theoretical results (Reviewers gsZx, svg8, 6kCb, MDNE)
- Rigorous mathematical proofs (Reviewer MDNE)
- Solid and convincing experiments (Reviewers gsZx, 6kCb, MDNE)
- Well-structured and well-written paper (Reviewers svg8, MDNE)

In our responses, we have diligently analyzed concerns and provided comprehensive evidence and justifications. Where there were misunderstandings, we offered detailed clarifications with references to specific sections of the paper. Additional figures and tables can be found in the attached PDF.

In conclusion, we believe that our paper makes meaningful contributions to the field of robust recommender system and will benefit the research community. We would like to thank all the reviewers for your valuable and constructive comments for improving our paper. We kindly request that the reviewers reevaluate our work based on our responses. If there are any further concerns that would prevent the reviewer from increasing the score, please let us know, and we would be happy to address these concerns during the discussion phase. We look forward to hearing from you.

Best regards,
Authors of paper 5740

---

### Author Response · Authors · 2024-08-14
**Summary**

We would like to express our sincere gratitude to all the reviewers for their valuable feedback and for dedicating time to evaluate our work. We are pleased that **all reviewers have given positive scores (7,6,5,5) on this paper**, and we are delighted that **our contributions have been recognized, including**:

- Addressing an important research problem (Reviewers gsZx, 6kCb, MDNE)
- Introducing an interesting and novel idea (Reviewers gsZx, MDNE)
- Providing reasonable motivation (Reviewers gsZx, MDNE)
- Demonstrating sound theoretical results (Reviewers gsZx, svg8, 6kCb, MDNE)
- Presenting rigorous mathematical proofs (Reviewer MDNE)
- Conducting solid and convincing experiments (Reviewers gsZx, 6kCb, MDNE)
- Delivering a well-structured and well-written paper (Reviewers svg8, MDNE)

Our paper **offers a theoretical explanation for why adversarial collaborative filtering (ACF) can outperform traditional collaborative filtering** in terms of performance and robustness. Additionally, we highlight **how to further enhance the benefits of ACF based on recommendation error bounds**. Building on these theoretical insights, we **propose PamaCF and conduct extensive experimental evaluations**. Our experimental results align well with our theoretical findings, as praised by Reviewers gsZx, 6kCb, and MDNE.

During the discussion period, our responses effectively addressed the concerns:

- Clarifying how we relaxed the assumptions and extended our theoretical insights.
- Demonstrating ACF's unique advantages, even with clean data.
- Providing experimental results confirming that PamaCF achieves state-of-the-art performance, even with non-dot-product-based loss functions or under different types of attacks.

We also clarified misunderstandings, including:

- Confirming that we use the same loss function as the cited paper.
- Clarifying that a unified base loss function is shared across all baselines in our experiments.
- Highlighting specific sections in the original paper that introduce evaluation metrics.
- Explaining that the Gaussian-based initialization and sign function do not affect our core findings.

We are encouraged by the positive feedback received from the reviewers.

We are eager and hopeful that this paper will be considered for presentation at the conference. We offer a fresh perspective on existing adversarial collaborative filtering methods. We believe this work could significantly impact the community, sparking discussions and analyses on how to improve the development of adversarial collaborative filtering algorithms.

Once again, we extend our heartfelt gratitude to all the reviewers for their constructive suggestions and for taking the time and effort to evaluate our work.

Best regards,
The Authors

---

### Decision · Program_Chairs · 2024-09-25

**Decision:**

Accept (poster)

**Comment:**

This work offers a theoretical insight into the effectiveness of adversarial collaborative filtering and introduces a practical learning approach applicable to various scenarios. Experiments conducted on three datasets show that the proposed method excels in robustness and recommendation performance. While reviewers generally acknowledge the paper's positive aspects, they have raised concerns about discrepancies between the theoretical results and the proposed methods, as well as the assumptions made in the theoretical analysis.